# TNF and type I interferon crosstalk controls the fate and function of plasmacytoid dendritic cells

Rebeca Arroyo Hornero [1,2], Raul A. Maqueda-Alfaro[3], Miguel A. Solís-Barbosa[3], Rebecca A. Leylek[1,2], Olin Medina Chavez[3], Olivia M. Martinez[2,4], Andres Gottfried-Blackmore[3,5,6] & Juliana Idoyaga [1,2,3,7] ✉

Plasmacytoid dendritic cells (pDCs) are major producers of type I interferon (IFN-I), an important antiviral cytokine, and activity of these cells must be tightly controlled to prevent harmful inflammation and autoimmunity. Evidence exists that one regulatory mechanism is a fate-switching process from an IFN-I-secreting pDC to a professional antigen-presenting conventional dendritic cell (cDC) that lacks IFN-I-secreting capacity. However, this differentiation process is controversial owing to limitations in tracking the fate of individual cells over time. Here we use single-cell omics and functional experiments to show that activated human pDCs can lose their identity as IFN-I-secreting cells and acquire the transcriptional, epigenetic and functional features of cDCs. This pDC fate-switching process is promoted by tumor necrosis factor but blocked by IFN-I. Importantly, it occurs in vivo during human skin inflammatory diseases and injury, and physiologically in elderly people. This work identifies the pDC-to-cDC reprogramming trajectory and unveils a mechanistic framework for harnessing it therapeutically.

Effective host defense against diverse threats relies on the generation of highly specialized immune cell lineages. Two closely related dendritic cell (DC) lineages collaborate to initiate immune responses[1,2]. Plasmacytoid DCs (pDCs) are secretory cells with a superior capacity to respond to viruses by quickly producing type I interferon (IFN-I), a key antiviral cytokine with detrimental effects when dysregulated. Conventional DCs (cDCs), comprising cDC1s and cDC2s, specialize in priming antigen-specific T cells. DC lineage commitment can be tracked to the hematopoietic stem cell[3–5] and occurs through a series of changes that progressively restrict pDC and cDC fates via the expression of lineage-defining transcription factors (TFs)[1,2]. Although lineage commitment is generally regarded as irreversible, a few observations suggest that pDCs can acquire antigen-presenting functions on activation. Yet, whether pDCs can fully lose their identity and acquire all the features of cDCs by rewiring their transcriptional regulation remains highly debated[6,7]. Understanding the extent and regulation of pDC plasticity could unlock new strategies to modulate immune responses during viral infections and IFN-I-driven diseases.

Hints that pDCs acquire antigen-presenting functions date back to the late 1990s, when CD40L-stimulated—but not resting—human pDCs were shown to activate allogeneic lymphocytes[8–10]. Mouse pDCs were similarly observed to gain antigen-presenting capacity after

[1]Department of Microbiology and Immunology, Stanford University School of Medicine, Stanford, CA, USA. [2]Immunology Program, Stanford University School of Medicine, Stanford, CA, USA. [3]Department of Pharmacology, University of California San Diego School of Medicine, La Jolla, CA, USA. [4]Department of Surgery, Division of Abdominal Transplantation, Stanford University School of Medicine, Stanford, CA, USA. [5]Department of Medicine, Division of Gastroenterology, University of California San Diego School of Medicine, La Jolla, CA, USA. [6]Gastroenterology Section, Veterans Affairs San Diego Healthcare System, La Jolla, CA, USA. [7]Department of Molecular Biology, University of California San Diego School of Biological Sciences, La Jolla, CA, USA. ✉e-mail: jidoyaga@health.ucsd.edu

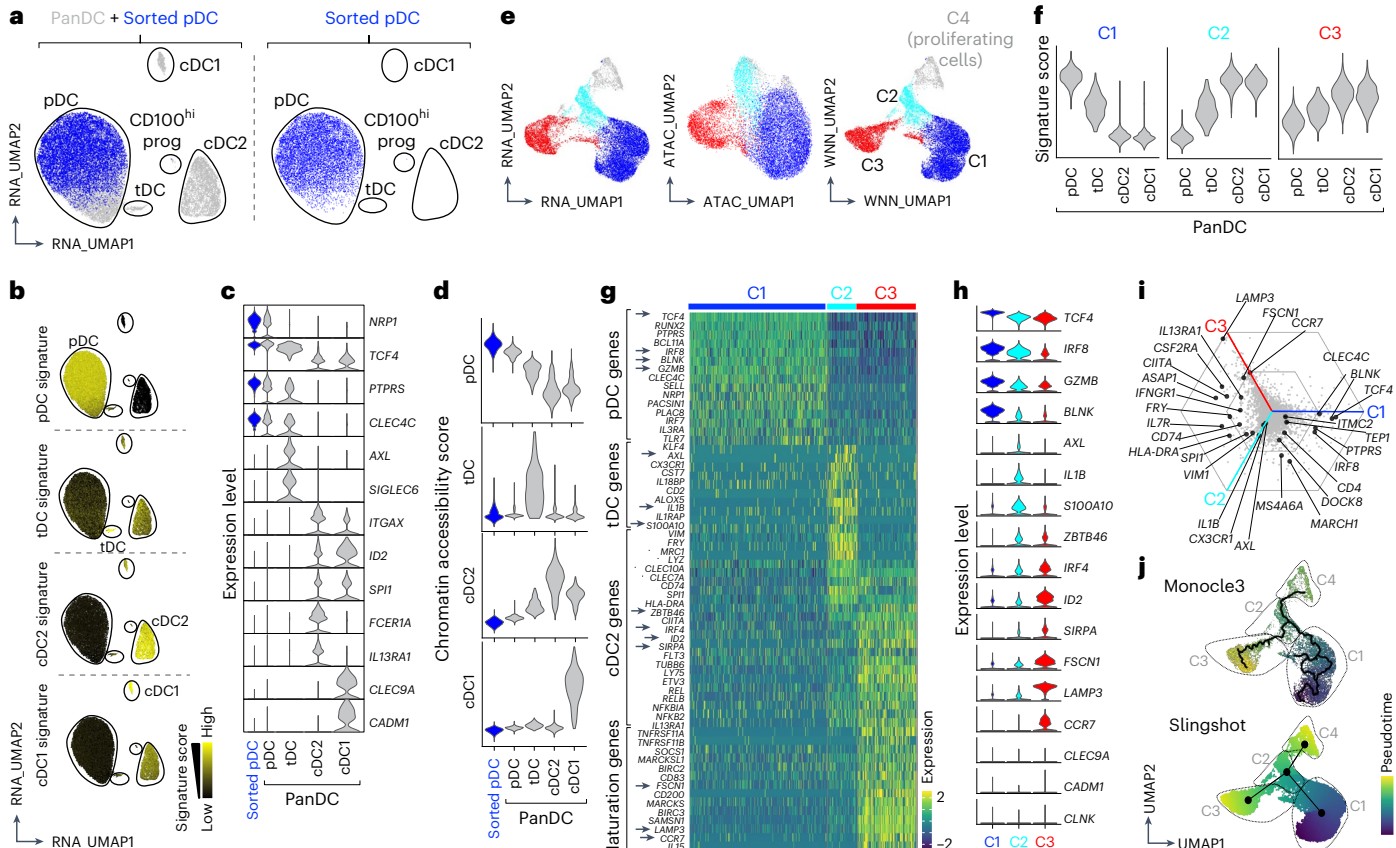

**Fig. 1 | Conversion of pDCs into icDC2s on activation. a–d,** snMultiome-seq of freshly isolated panDCs and pDCs (Extended Data Figs. 1 and 2a,b). **a,** Uniform Manifold Approximation and Projection (UMAP) of integrated panDC and pDC (left) or pDC-only (right) snRNA-seq. **b,** Signature score for DC subsets on integrated panDC and pDC UMAP (Supplementary Table 1). **c,** Expression of selected genes across DC subsets in panDC (gray) and pDC (blue) snRNA-seq. **d,** Chromatin accessibility signature score for integrated panDC (gray) and pDC (blue) snATAC-seq. **e–j,** snMultiome-seq of pDCs cultured for 4 d with CD40L (Extended Data Fig. 2c–g). **e,** UMAP of gene expression (left), chromatin accessibility (middle) and their integrated weighted nearest neighbor (WNN) profiles (right), colored by unsupervised clustering on WNN. **f,** Signature score for clusters C1–C3 (Supplementary Table 2) projected onto panDC snRNA-seq. **g–h,** Heatmap (**g**) and violin plots (**h**) of selected genes from clusters C1–C3. **i,** Barycentric plot showing relative gene expression. **j,** WNN UMAP with trajectory and pseudotime calculated by Monocle3 and Slingshot. Prog, progenitors.

activation[11,12]. These findings, however, were later scrutinized by the realization that original pDC preparations were contaminated with DC precursors or transitional DCs (tDCs)—a population with mixed pDC or cDC2 features and the natural capacity to differentiate into antigen-presenting cDC2s[2,13–18]. Most studies of pDC plasticity relied on population-level methods that could not resolve individual cell fate or in vivo models that may not distinguish fate changes from new hematopoiesis[19–21]. As a result, a key question remains: can specialized IFN-I-secreting pDCs lose their identity, rewire their transcriptional program and fully adopt the features of antigen-presenting cDCs? If so, what cues regulate this transition?

Here we use single-cell omics to dissect the pDC-to-cDC transition with high resolution, identifying tumor necrosis factor (TNF) as an important trigger and IFN-I as a brake. We also show that human pDC fate switching occurs in vivo during inflammatory skin diseases, traumatic wounding and physiological aging.

## Results

### Plasmacytoid DCs convert into cDC2s on activation

We hypothesized that single-cell multiomics would allow us to track the fate of individual pDCs during activation. To establish a reference framework, we first generated a panDC map capturing the transcriptome and chromatin landscape of all major human blood DC populations. For this reference map, we excluded inflammatory DC3s—given their similarity

to monocytes—by purifying CD14⁻ cells (Extended Data Fig. 1a–c). Unsupervised clustering resolved pDC, cDC2 and cDC1 clusters using two public datasets[13,14], with frequencies mirroring those observed by flow cytometry (Extended Data Fig. 1d–g and Supplementary Table 1). As expected, the cDC2 cluster of the panDC dataset aligned with cDC2s, but not DC3s using a public dataset[22] (Extended Data Fig. 1h). We also identified a small population of cells aligning with CD100^hi DC precursors, previously described in ref. 14 (Extended Data Fig. 1i). Finally, our map captured a population of tDCs that, as previously reported[16], aligned with AXL⁺ DCs (ASDCs)[14] and some definitions of pre-cDCs[13] (Extended Data Fig. 1i). Thus, our panDC single-nucleus (sn)Multiome sequencing approach resolves human blood DC heterogeneity and provides a high-resolution framework to track pDC fate transitions.

We next generated an snMultiome dataset of sorted pDCs and integrated it with the panDC dataset to assess whether our pDC preparations contained putative contamination with other DCs (Fig. 1a and Extended Data Fig. 2a,b). Sorted pDCs were assigned only to the pDC cluster of the panDC dataset and expressed higher levels of pDC-associated genes (*NRP1*, *TCF4* and *PTPRS*; Fig. 1a–c). Sorted pDCs shared a chromatin accessibility signature with pDCs from the panDC dataset, but not with tDCs, cDC2s and cDC1s (Fig. 1d). Thus, our pDC preparations lack contamination with other DCs.

We then queried the fate of activated pDCs. To replicate conditions suggested to promote fate switching[8,9], we stimulated sorted pDCs

with CD40L, always in the presence of the survival cytokine interleukin (IL)-3 (Extended Data Fig. 2c). Approximately 50% of live pDCs were recovered at days 2 and 4 and ~35% at day 6 (Extended Data Fig. 2d). snMultiome–seq of live pDCs sorted from day 4 cultures resolved four clusters of cells with distinct gene expression and chromatin profiles (Fig. 1e). Cluster C4, identified as proliferating cells (Extended Data Fig. 2e), was excluded from further analysis because it likely reflects a cell-cycling state rather than a distinct functional population. The identity of other clusters was inferred by aligning their transcriptomic signature with our panDC dataset (Fig. 1f). Only cluster C1 aligned with pDCs, whereas clusters C2 and C3 aligned with cDCs. All clusters had some association with tDCs, according to their transitional pDC-to-cDC2 gene signature[14–17,21]. C1 expressed higher levels of pDC genes (*TCF4* and *GZMB*; Fig. 1g,h). C3 expressed higher levels of cDC2 transcripts (*SIRPA*, *IRF4* and *ZBTB46)* and DC maturation genes (*LAMP3*, *CCR7* and *FSCN1*). None of the clusters expressed cDC1 genes (*CLEC9A*, *CADM1* and *CLNK*; Fig. 1h and Extended Data Fig. 2f). C2 expressed some cDC2 (*CLEC10A*, *MRC1* and *CLEC7A*) and tDC (*AXL*, *IL1B* and *S100A10* (ref. 14)) genes, suggesting that it represents an intermediate pDC-to-cDC2 cell stage. To explore cluster similarities, we analyzed gene expression on a barycentric plot (Fig. 1i). Clusters C1 and C2 shared the expression of pDC genes (*IRF8* and *PTPRS*), whereas clusters C2 and C3 shared expression of some cDC2 genes (*IL13RA1*, *HLA-DRA* and *CIITA*). To infer a temporal relationship between the clusters, we performed trajectory analysis (Fig. 1j and Extended Data Fig. 2g). Monocle3 and Slingshot predicted cluster C2 in the middle of the trajectory between clusters C1 and C3, emphasizing their transitional features.

Altogether, our data show that pDC activation results in the emergence of three clusters, that is, C1 which associates with pDCs, an intermediate C2 that resembles tDCs and C3 which resembles cDC2s or mature DCs. For simplicity, we refer to these pDC-derived cells as induced tDCs (itDCs) and induced cDC2s (icDC2s).

## The icDC2s are a stable endpoint of pDC differentiation

To develop a gating strategy for sorting pDCs, itDCs and icDC2s, we screened surface markers by mass cytometry (CyTOF), enabling analysis of >45 proteins at the single-cell resolution. The pDCs, itDCs and icDC2s were identified unbiasedly using Scaffold, which assigns identity based on phenotypic similarities to a reference map (Fig. 2a). Marker enrichment modeling identified CD11c and CD33 as the optimal combination to discriminate pDCs, itDCs and icDC2s, which was corroborated by flow cytometry over time (Fig. 2b–d). The pDC frequencies declined, icDC2s increased and itDCs remained relatively stable across the culture period (Extended Data Fig. 3a).

To link CD11c/CD33 expression with snMultiome transcriptional signatures (clusters C1–C3), we used a plate-based cellular indexing approach coupled with SMART–seq2 (Fig. 2e). We analyzed day 2 cultures, hypothesizing that they would capture a wider spectrum of itDCs transitioning from pDCs to icDC2s. Freshly isolated pDCs and tDCs were also

analyzed to rule out contamination. Unsupervised analysis identified six clusters consistent across two donors (Fig. 2f and Extended Data Fig. 3b). Clusters C4 and C5 corresponded to freshly isolated pDCs and tDCs based on index sorting. The remaining four clusters corresponded to the day 2 cultures, including one cluster of proliferating cells that was not analyzed further (Extended Data Fig. 3c). Cluster C1–C3 identity was assigned by aligning to the panDC signature. Cluster C1 was pDCs, whereas clusters C2 and C3 were enriched in tDC and cDC2 signatures (Fig. 2g), matching our snMultiome analysis. Indeed, SMART–seq2 clusters C1–C3 showed strong correlation with snMultiome clusters C1–C3 (Fig. 2h and Supplementary Table 2). Cluster C1 expressed pDC genes, C2 expressed tDC and cDC2 genes and C3 expressed cDC2 or maturation genes (Fig. 2i and Supplementary Table 3). Correlation between SMART–seq2 clusters and index sorting showed that CD11c or CD33 protein expression identified pDCs, itDCs and icDC2s with >65% accuracy (Fig. 2j). Accordingly, flow cytometry confirmed differential expression of several genes at the protein level (Fig. 2k and Extended Data Fig. 3d). Thus, pDCs, itDCs and icDC2s can be reliably identified by CD11c or CD33 expression.

The high correlation between SMART–seq2 (day 2) and snMultiome (day 4) profiles indicated that pDCs, itDCs and icDC2s represent transcriptionally distinct states. To assess the stability and directionality of these states, we sorted each population from day 2 or day 4 cultures and re-cultured them for 2–4 d before reassessing identity (Fig. 2l,m and Extended Data Fig. 4a–c). The icDC2s preserved their phenotype, consistent with a stable, terminally differentiated state. A small fraction of pDCs (~30%) transitioned into itDCs or icDC2s, but most retained their identity. In contrast, ~50% of itDCs became icDC2s, whereas reversion to pDCs was rare, supporting a unidirectional trajectory. This directionality persisted without CD40L (Fig. 2m and Extended Data Fig. 4d,e), suggesting that the process is not readily reversible. To rule out survival bias, we assessed apoptosis. Day 1 icDC2s showed slightly elevated—but modest—apoptosis, likely due to the challenges in quantifying rare icDC2s at this time point (Extended Data Fig. 4f). In contrast, day 2 and day 4 icDC2s showed no significant apoptosis, supporting their stability.

Previous studies describe limited proliferation as an intermediate step in fate switching[23]. To test this, we tracked: (1) cellular state; (2) Ki67 expression; and (3) CellTrace Violet (CTV) dilution as a readout of cell division (Fig. 2n). At day 2, ~40% of the cells were itDCs or icDC2s, but Ki67+ or CTV^low cells were rare, suggesting that proliferation is not required for fate switching. By day 4, ~60% of the cells were itDCs or icDC2s. Few pDCs or icDC2s expressed Ki67, whereas ~45% of itDCs were Ki67+. CTV^low itDCs increased from 50% at day 4 to 90% by day 6, indicating active proliferation. Accordingly, trajectory analysis linked itDCs (C2) with proliferating cells (C4) (Fig. 1j). Although few icDC2s expressed Ki67, 50–90% were CTV^low at days 4–6, consistent with rederivation from itDCs. Notably, only one to two cell divisions occurred by day 4 (Fig. 2o), distinguishing pDCs, itDCs and icDC2s from highly proliferative progenitors[14].

**Fig. 2 | The icDC2s are a stable endpoint of pDC differentiation. a**, Scaffold map of CyTOF data from pDCs cultured 6 d with CD40L (one of three donors, three experimental). **b**, Marker enrichment modeling scores for cells falling within pDC, tDC and cDC2 clusters from **a**. **c**, CyTOF gating strategy for pDCs, itDCs and icDC2s based on CD33 and CD11c. **d**, Flow cytometry expression of CD33 and CD11c in pDCs cultured with CD40L for 0, 2, 4, or 6 d. The middle and bottom plots show staining controls (five donors, five experimental). **e**, SMART–seq2 experimental setup: pDCs from two blood donors cultured 2 d with CD40L and single-cell sorted (scFACS) as pDCs, itDCs or icDC2s (gating in **d**). Fresh pDCs and tDCs were also scFACS and sequenced. **f**, SMART–seq2 UMAP colored by unsupervised clustering. **g,h**, Signature scores of panDC snMultiome–seq populations (**g**) and day 4 culture clusters C1–C3 (**h**) projected onto SMART–seq2 clusters. **i**, Heatmap of top 100 DEGs between SMART–seq2 clusters C1–C3. **j**, Correlation between sorted cell types and SMART–seq2 clusters C1–C3. **k**, Flow cytometry protein expression in day 4 cultures with numbers indicating

geometric mean fluorescence intensity (gMFI) or percentage of positive cells (1 of 9 (BDCA2), 1 of 4 (BDCA4), 1 of 5 (LILRA4, CD62L), 1 of 7 (CD172a, CLEC10A) and 1 of 14 (HLA-DR) donors; 4–14 experimental; Extended Data Fig. 3d). **l**, FACS purification strategy (day 4; left); CD33 and CD11c profiles post-sort (day 4) and after re-culture with CD40L (right, day 6) (one of five donors, five experimental). **m**, As in **l**, but percentage of pDCs, itDCs and icDC2s after re-culturing with CD40L (n = 5 donors, 5 experimental) or medium (IL-3) alone (n = 4 donors, 4 experimental) (mean + s.d.; Extended Data Fig. 4a–e). **n**, Pie charts of pDC, itDC and icDC2 frequencies over time (top), with Ki67+ (middle) and CTV^low (bottom) fractions (n = 3 donors, 3 experimental). **o**, As in **n**, but representative CTV histogram for a donor. **p**, Single-cell differentiation assay: sorted CTV-labeled pDCs were plated at 1–5,000 cells per well with CFSE-labeled filler (F) pDCs and cultured for 4 d. The bar graphs show the percentage of pDCs, itDCs and icDC2s among CTV+ cells; the numbers on top indicate recovered cells (n = 4 donors, 4 experimental).

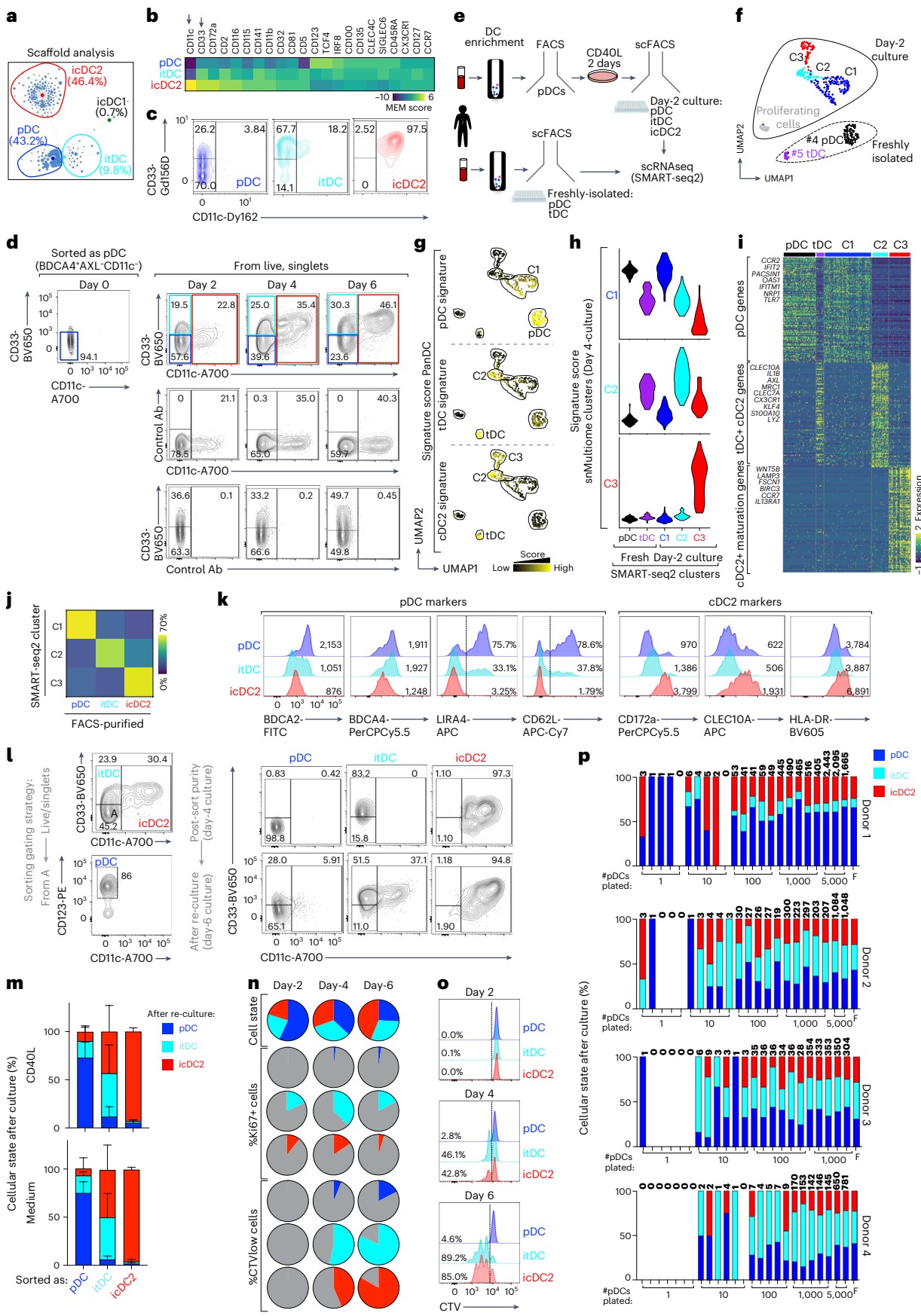

We next investigated whether individual pDCs could generate itDCs and icDC2s using a clonal differentiation assay. We cultured 1, 10, 100, 1,000 or 5,000 CTV-labeled pDCs with 5,000 (carboxyfluorescein succinimidyl ester) CFSE-labeled 'filler' pDCs from the same donor. After 4 d, we assessed the fate of CTV$^+$ cells by flow cytometry (Fig. 2p and Extended Data Fig. 4g). Single pDCs gave rise to itDCs and icDC2s in multiple wells, confirming their intrinsic differentiation capacity (Fig. 2p). Wells seeded with 1–10 pDCs showed greater output variability, underscoring limited proliferation and intrinsic cell-to-cell variability.

Together, these results show that individual pDCs intrinsically generate icDC2s via a proliferative itDC intermediate. Transcriptomic and protein expression (CD5$^+$, CD163$^-$ and CD14$^-$) indicate that icDC2s most closely resemble cDC2s, not DC3s, suggesting a conserved differentiation path shared with tDCs (Extended Data Fig. 4h,i)[17].

## The pDCs convert into functional cDCs

The pDC round-to-ovoid morphology is adapted for cytokine secretion, whereas the cDC dendritic morphology increases surface contact with T cells[24]. The icDC2s lost the pDC round morphology and instead displayed a stellate morphology with pseudopods and dendrites, resembling freshly isolated cDC2s (Fig. 3a). Quantification confirmed decreased circularity and increased area index in icDC2s compared with pDCs (Fig. 3b and Extended Data Fig. 5a). The itDCs showed intermediate morphology. These findings indicate that fate switching includes acquisition of cDC-like morphology.

Gene ontology (GO) analysis revealed enrichment for 'response to IFN-I' and 'regulation of secretory pathway' genes in pDCs, including *IRF7* and *MYD88* expression, which drive IFN-I secretion downstream of toll-like receptors (TLRs)[25] (Fig. 3c,d). Accordingly, day 4-sorted pDCs, but not icDC2s, secreted IFNα in response to the TLR9 agonist CpG-A (Fig. 3e) (low itDC yield precluded testing).

The cDC2s excel at activating CD4$^+$ T cells through major histocompatibility complex class II (MHC-II)-mediated antigen presentation. Similarly, icDC2s showed enrichment of 'MHC protein complex' and 'antigen assembly' pathways (Fig. 3c). The icDC2s upregulated *CIITA*, the master MHC-II regulator, correlating with increased *HLA-DR* gene and protein expression (Figs. 2k and 3d and Extended Data Figs. 3d and 5b). *CIITA* induction was accompanied by increased chromatin accessibility at promoter I and predicted enhancers (Extended Data Fig. 6a), a known cDC hallmark[26,27]. In addition, icDC2s downregulated *MARCH1* (Fig. 3d), a ligase that promotes MHC-II internalization and degradation, suggesting stabilization of peptide–MHC-II complexes at the membrane—another hallmark of cDCs[27,28]. *MARCH1* downregulation may occur post-transcriptionally[29] and was accompanied by reduced chromatin accessibility at predicted enhancers (Extended Data Fig. 6b). The icDC2s expressed higher levels of CD86—consistent with *MARCH1* downregulation[30]—as well as CD80 and CD40, costimulatory molecules that synergize with MHC-II–T cell receptor (TCR) signaling (Fig. 3f and Extended Data Fig. 3d). Altogether, icDC2s upregulate the gene programs and surface molecules required for effective antigen presentation.

We next analyzed antigen capture, the first step in MHC-II antigen presentation. Day 4-culture cells were incubated with fluorescently labeled live or apoptotic autologous peripheral blood mononuclear cells (PBMCs), xenogeneic mouse splenocytes or *Staphylococcus aureus*. Controls included cytochalasin D treatment and 4 °C incubation to block actin polymerization and intracellular movement. Minimal capture was observed in these controls or with live autologous cells, which controlled for the active recognition of dying or foreign particulate antigens (Extended Data Fig. 5c–e). In the other cases, icDC2s captured particulate antigen more efficiently than pDCs, resembling cDC2s (Fig. 3g,h and Extended Data Figs. 5c–e and 6c,d). The itDCs showed intermediate capacity. We also evaluated antigen processing using DQ-ovalbumin (DQ-OVA), a fluorogenic substrate for

lysosomal proteases. The itDCs and icDC2s processed antigen more efficiently than pDCs, again resembling freshly isolated cells (Fig. 3g and Extended Data Figs. 5f and 6d).

To test antigen presentation, we measured the ability of icDC2s to activate autologous naive CD4$^+$ T cells using xenogeneic cells as antigen (Fig. 3i,j and Extended Data Fig. 5g,h). After antigen loading, sorted pDCs and icDC2s were cocultured with autologous naive T cells (itDCs were excluded due to low yield). Antigen-loaded icDC2s were superior to pDCs at inducing naive T cell proliferation and activation and closely resembled freshly isolated cDC2s (Fig. 3k), confirming robust antigen-presenting function.

The cDC2s traffic to T cell-rich lymphoid areas via CCR7-dependent migration[31]. The icDC2s expressed higher levels of *CCR7* gene and protein than the pDCs (Fig. 3l and Extended Data Fig. 3d). In chemotaxis assays toward CCR7 ligands CCL19 or CCL21, icDC2s had superior migration compared with pDCs (Fig. 3m). Conversely, pDCs migrated more efficiently toward CCL2, a CCR2 ligand associated with lymphoid organ homing[32].

Collectively, pDCs undergo functional reprogramming to acquire the hallmarks of cDC2s.

## Fate switching of pDCs is triggered by TNF

Only day 4 pDCs retained TF activity seen in freshly isolated pDCs, whereas itDCs and icDC2s shared TF activity with tDCs and cDC2s (Fig. 4a and Extended Data Fig. 7a). Specifically, *TCF4* and *RUNX2*—two pDC lineage-defining TFs[32–34]—were active in pDCs but silenced in itDCs and icDC2s (Fig. 4b). In contrast, icDC2s gained *IRF4* activity, a cDC2-associated TF[1], whereas itDCs exhibited *KLF4* activity, like tDCs[17,18]. These data indicate that fate switching involves silencing of pDC TFs and acquisition of cDC2-defining programs.

*TCF4* promotes pDC lineage-restricted genes while repressing cDC lineage genes[34]. Its activity declined during conversion, correlating with decreased TCF4 protein and upregulation of *ID2*, a known TCF4 antagonist[33,34] (Fig. 4c–e and Extended Data Fig. 7b). TCF4-activated genes were enriched in cultured pDCs, whereas TCF4-repressed genes were upregulated in itDCs and icDC2s[35] (Fig. 4f,g and Extended Data Fig. 7c). Notably, itDCs also showed activity for pioneer TFs *CEBPA* and *SPI1* (Fig. 4c), which open compacted chromatin[36] and likely enable access to cDC2 programs. SPI1 regulates pDC versus cDC identity[37]; accordingly, SPI1-repressed genes were enriched in pDCs, whereas SPI1-activated genes were enriched in icDC2s, including cDC-specific genes (Fig. 4h,i and Extended Data Fig. 7d).

Although no tDC or precursor contamination was detected (Fig. 1e), only a fraction of pDCs converted, suggesting that extrinsic cues or cell-to-cell variations influence fate. Gene set enrichment analysis (GSEA) revealed that pDCs expressed 'IFNα response' genes (for example, *ISG20*, *MX1* and *OAS1*), whereas icDC2s expressed 'TNF signaling via NF-κB' genes (for example, *TNFAIP2*, *TNFAIP3* and *TRAF1*) (Fig. 4j,k). The icDC2s showed enhanced NF-κB (nuclear factor κ-light-chain-enhancer of activated B cells) activity (Fig. 4l and Extended Data Fig. 7e), implicated in cDC2 development[38]. These patterns suggested that TNF and IFN-I—cytokines with known antagonism[39]—may regulate pDC fate. Indeed, TNF alone induced fate switching, whereas IFNα blocked CD40L- and TNF-induced conversion in a dose-dependent manner (Fig. 4m and Extended Data Fig. 8a), demonstrating opposing roles.

We further examined TNF's role in fate switching. CD40L stimulation triggered TNF secretion, peaking at days 1–2 and declining by day 4 (Fig. 4n and Extended Data Fig. 8b). IL-8, a TNF-driven cytokine[40], mirrored TNF. Blocking TNF receptor 1 (TNFR1) or 2 (TNFR2) impaired CD40L-driven icDC2 generation at days 2 and 4, with stronger effects from TNFR1 blockade (Fig. 4o and Extended Data Fig. 8c). We next asked whether TNF promotes fate switching by downregulating TCF4 (ref. 41). Both TNF and CD40L reduced TCF4 expression in cultured pDCs (Fig. 4p), correlating with decreased IRF8 (IFN regulatory factor 8),

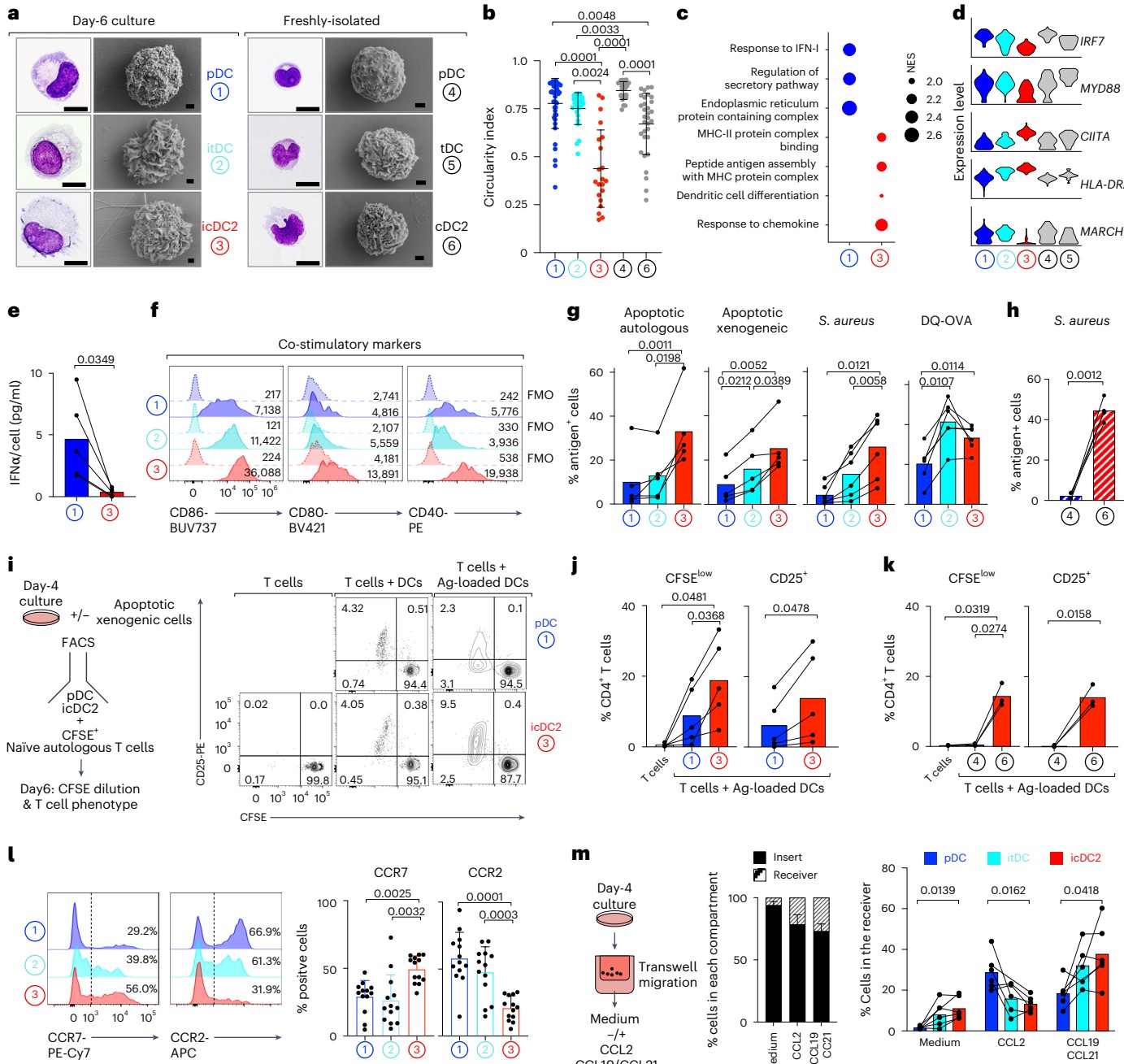

**Fig. 3 | Conversion of pDCs into functional cDCs. a**, Cytology (scale, 10 µm) and scanning electron microscopy (scale, 1 µm) of freshly isolated and day 6 culture cells, FACS purified (*n* = 1 of 2 donors; 2 experimental). **b**, Circularity index of freshly isolated pDCs (*n* = 25 cells) and cDC2s (*n* = 32 cells) and day 6-culture pDCs (*n* = 43 cells), itDCs (*n* = 26 cells) and icDC2s (*n* = 21 cells). Each dot represents a single cell (one of two donors). **c**, GSEA of GO pathways between SMART–seq2 clusters C1 and C3. **d**, Expression of selected genes from **c**. **e**, IFNα secretion measured after 24-h re-stimulation of pDCs and icDC2s (FACS purified at day 4) with CpG-A, normalized to the cell number per condition (*n* = 5 donors, 5 experimental). **f**, Flow cytometry protein expression in day 4 cultures (gMFI; one of six (CD80, CD40) and one of seven (CD86) donors; six to seven experimental; Extended Data Fig. 3d). **g**, Uptake of apoptotic autologous (*n* = 5 donors, 5 experimental), apoptotic xenogeneic (*n* = 5 donors, 5 experimental), *S. aureus* (*n* = 6 donors, 6 experimental) and processing of DQ-OVA (*n* = 5 donors, 5 experimental) by day 4 cultures relative to 4 °C controls (Extended Data Fig. 5c–f). **h**, As in **g**, but using freshly isolated DCs (*n* = 4 donors, 4 experimental;

Extended Data Fig. 6d). **i**, Experimental setup for autologous naive T cell priming (left) and a representative plot (right). **j**, Percentage of CFSE^low CD4+ T cells and CD25+CD4+ T cells from **i** (*n* = 5 donors, 5 experimental). **k**, As in **j**, but using freshly isolated DC subsets (*n* = 3 donors, 3 experimental; Extended Data Fig. 6c). **l**, Flow cytometry protein expression in day 4 cultures (*n* = 13 donors, 13 experimental; Extended Data Fig. 3d). **m**, Migration of day 4 cultures in transwell assay with CCL2, CCL19 + CCL21 or medium alone (IL-3). Left, experimental setup. Middle, percentage of total cells that migrated (receiver) or remained (insert). Right, migrating relative to total cells for each subset (*n* = 6 donors, 6 experimental). Statistical tests show mean ± s.d. throughout.: Kruskal–Wallis test with Dunn's test (**b**); two-sided, paired Student's *t*-test (**e**,**h**,**j**, CD25+; **k**, CD25+); paired one-way analysis of variance (ANOVA) with Tukey's test (**g**,**j**, CFSE^low; **k**, %CFSE^low; **l**); and mixed-effects model with the Geisser–Greenhouse correction and Tukey's multiple-comparison test (**m**). Ag, antigen; NES, normalized enrichment score; GSEA, Gene Set Enrichment Analysis; GO, Gene Ontology.

a TCF4-regulated TF[21,33]. TNFR1 or -2 blockade abrogated TCF4 and IRF8 downregulation, confirming a TNF–TCF4 regulatory axis (Fig. 4p). We also tested whether TNF boosts CD40L responsiveness. Although CD40 was barely detectable in freshly isolated pDCs, it was rapidly upregulated by CD40L stimulation (Extended Data Fig. 8d). TNFR1 or -2 blockade reduced CD40 induction (Extended Data Fig. 8e), suggesting that TNF promotes a feedforward loop enhancing pDC responsiveness. Altogether, these data position TNF as a central regulator of the pDC-to-icDC2 switch by downregulating TCF4 and dismantling the pDC gene program.

Next, we asked whether insufficient TNF signaling limits conversion to only a fraction of pDCs. Conversion plateaued at ~25% on day 2, regardless of TNF dose (Fig. 4q) and rose to ~35–40% by day 4 (Fig. 4r)—likely reflecting a single cell division (Fig. 2o). This plateau was not due to TNF exhaustion or TNF-mediated cell death, because additional TNF had no effect and cell recovery remained unchanged (Fig. 4r and Extended Data Fig. 8f). Instead, only ~10–30% of pDCs expressed TNFR1 and ~60–80% expressed TNFR2, suggesting that variation in receptor abundance limits conversion (Extended Data Fig. 8g). Indeed, interindividual variation in TNFR1 (but not TNFR2) expression correlated with icDC2s frequencies across donors (Fig. 4s).

IFN-I was undetectable by ELISA in CD40L-stimulated cultures and blocking IFN-I signaling had no effect on icDC2 frequency (Extended Data Fig. 8h), indicating that CD40L does not induce IFN-I. IFN-stimulated gene expression in cultured pDCs likely reflects prior in vivo exposure, because these genes were detected in freshly isolated cells (Extended Data Fig. 8i). We next tested how, exogenously, IFN-I blocks fate switching. Addition of IFN-I to icDC2s did not reverse their fate, emphasizing the stability of this population and suggesting that IFN-I acts upstream of fate commitment (Fig. 4t and Extended Data Fig. 4d). Although exogenous IFN-I reduced IL-8 (ref. 40), it did not suppress TNF secretion (Extended Data Fig. 8j). Instead, it downregulated TNFR1 or TNFR2 (Fig. 4u), limiting pDC access to the fate-switching cue. Consistently, IFN-I prevented TNF-induced TCF4 downregulation (Fig. 4v), explaining reduced icDC2 frequencies.

Together, our findings support a model in which CD40L-induced TNF drives pDC fate switching by downregulating TCF4, whereas IFN-I counteracts this transition by blocking TNF signaling, preserving TCF4 and reinforcing the pDC program (Fig. 4w). These results suggest that fate switching may not occur in IFN-I-rich environments. Accordingly, previously described pDC states induced by viral mimics and marked by CD80 and PD-L1 expression[20] did not align with pDCs, itDCs or icDC2s (Extended Data Fig. 8k), suggesting that these markers may reflect early activation rather than an identity shift.

## Fate switching of pDCs occurs during wounding

We assessed whether pDC-derived icDC2s aligned with pDCs in vivo or, instead, resembled cDCs. We focused on skin, a tissue devoid of pDCs at steady state but known to recruit them during inflammatory diseases[42]. We analyzed myeloid cell clusters from a publicly available cellular indexing of transcriptomes and epitopes by sequencing (CITE-seq) dataset of healthy and inflamed human skin[43] (Extended Data Fig. 9a–c). Unbiased transcriptomic analysis identified a distinct pDC cluster in patients with psoriasis, atopic dermatitis and other rashes, but not in healthy skin (Extended Data Fig. 9d). This cluster aligned with pDCs from our panDC dataset and day 4 cultures and expressed canonical pDC genes (for example, *TCF4*, *IL3RA* and *SELL*) (Extended Data Fig. 9e,f). In contrast, the icDC2 gene signature mapped to cells expressing cDC2 and maturation markers (Extended Data Fig. 9e,f), confirming that icDC2s lose pDC identity and cluster with cDC2s in unbiased analyses.

As icDC2 and cDC2 signatures are largely indistinguishable, we next leveraged a different skin model to evaluate whether TNF-triggered pDC fate switching occurs in vivo. Skin suction blisters, a model of traumatic wounding, are infiltrated by CD123[hi] pDCs and a distinct CD123[int] DC population of unclear origin[44]. Although CD123[int] DCs share features with pDCs—including BDCA2 expression—they also exhibit hallmark cDC functions: antigen presentation, T cell activation and lymph node migration. We hypothesized that CD123[int] DCs may represent the in vivo counterpart of pDC-derived icDC2s. We aligned the transcriptomes of CD123[hi] pDCs and CD123[int] DCs from blisters[44] with our day 2-culture signatures. CD123[hi] pDCs aligned with day 2 pDCs and expressed hallmark genes (*TCF4*, *IRF7*, *CCR2* and *TLR7*) (Fig. 5a,b and Supplementary Table 4). In contrast, CD123[int] DCs aligned with itDCs and icDC2s and expressed *ID2*, *CCR7*, *LAMP3*, *FSCN1* and *SIRPA*. Like icDC2s, CD123[int] DCs were enriched for the 'TNF signaling via NF-κB' pathway (Fig. 5a–c). Although GSEA did not reveal global enrichment of the IFNα pathway (Extended Data Fig. 9g), CD123[hi] pDCs expressed several IFN-I-related genes, including *TLR7*, *MYD88*, *IRF7* and *MX1* (Fig. 5c). Notably, CD123[hi] pDCs and CD123[int] DCs also aligned with freshly isolated pDCs and cDC2s, respectively (Extended Data Fig. 9h). Together, these data support the model that CD123[int] DCs arise from pDCs in vivo, although we cannot completely rule out the contribution of circulating cDC2s.

We next examined how these signatures changed when blisters were challenged with house dust mite (HDM), a model of sterile skin inflammation[44]. CD123[int] DCs maintained alignment with itDCs and icDC2s in both saline- and HDM-treated blisters (Fig. 5d). In contrast, a subgroup of CD123[hi] pDCs shifted toward an icDC2-like profile, suggesting intermediate states. To resolve this, we stratified CD123[hi]

**Fig. 4 | Fate switching of pDCs is triggered by TNF and blocked by IFN-I.**
**a**, Venn diagrams showing overlap of active TFs between freshly isolated DCs and day 4 cultures, analyzed by snMultiome–seq (Extended Data Fig. 7a). **b**, TF activity score from **a**. The dot size corresponds to average chromVAR motif accessibility and gene expression scores. **c**, Motif accessibility score onto the UMAP of Fig. 1e. **d**, Motif accessibility score against pseudotime for clusters C1–C3 (snMultiome–seq). **e**, *ID2* expression on day 2 cultures (SMART–seq2). **f,g**, Signature score of TCF4-regulated genes (**f**) and heatmap of selected TCF4-regulated genes (**g**), by SMART–seq2 (Extended Data Fig. 7c). **h,i**, Signature score (**h**) and heatmap (**i**) of SPI1-regulated genes (SMART–seq2) (Extended Data Fig. 7d). **j**, GSEA of selected pathways between pDCs and icDC2s (SMART–seq2). **k**, Expression of selected genes from **j**. **l**, Motif accessibility score versus pseudotime for clusters C1–C3 on snMultiome–seq (Extended Data Fig. 7e). **m**, Percentage of icDC2s in day 2 cultures with indicated cytokines (n = 7 donors, 7 experimental). **n**, TNF and IL-8 secretion in day 2 cultures (n = 11 donors, 11 experimental). **o**, Percentage of icDC2 in day 2 cultures with CD40L plus control antibodies (n = 6, 6 experimental), anti-TNFR1 and TNFR2 antibodies (n = 6 donors, 6 experimental), anti-TNFR1 antibodies (n = 4, 4 experimental) and anti-TNFR2 antibodies (n = 4, 4 experimental). **p**, Percentage

TCF4[hi] and IRF8[hi] cells in day 2 cultures (n = 3 donors, 3 experimental), with representative flow cytometry plot (one of three donors). **q**, Percentage icDC2s in day 2 cultures with increasing TNF concentrations (2, 20, 200 and 2,000 ng ml[−1]) (n = 4 donors, 4 experimental). **r**, Percentage of icDC2 in day 4 cultures with increasing TNF concentrations (200 or 2,000 ng ml[−1]; n = 7 donors, 7 experimental; left) and with a second TNF dose added at day 2 (200 or 2,000 ng ml[−1], respectively; n = 5 donors, 5 experimental; right). **s**, Correlation between TNFR1 (n = 8 donors, 8 experimental) or TNFR2 (n = 5 donors, 5 experimental) expression and icDC2 frequencies in day 2 cultures. **t**, Day 4-sorted pDCs (n = 4 donors, 4 experimental), itDCs (n = 3 donors, 3 experimental) and icDC2s (n = 4 donors, 4 experimental) re-cultured with IFNα for 2 additional days (analyzed on day 6). **u**, TNFR1[+] (n = 6 donors, 6 experimental) and TNFR2[+] (n = 5 donors, 5 experimental) cells in day 2 cultures, with or without IFNα. **v**, Percentage TCF4[hi] and IRF8[hi] cells in day 4 cultures (n = 4 donors, 4 experimental). **w**, Schematic of proposed TNF or IFN-I regulation of pDC fate. Statistical tests give the mean + s.d. throughout and are as follows: two-sided, Wilcoxon's matched-pairs sign-rank test (**m**: groups 5–6; **n**); two-sided, paired Student's *t*-test (**o,u,v**); paired one-way ANOVA with Tukey's test (**m**: groups 1–4, **p,r**); paired one-way ANOVA with Dunnett's test (**q**); and Pearson's correlation (**s**).

pDCs—originally sorted as a homogeneous CD123[hi]BDCA2[+] pDC population—into 'high', 'intermediate' and 'low' pDC signature subgroups (Extended Data Fig. 9i). The low pDC subgroup aligned with icDC2s and showed loss of pDC-defining genes alongside upregulation of icDC2 genes, similar to CD123[int] DCs (Fig. 5e,f). They also expressed

reduced *IL3RA* (encoding CD123), demonstrating that transcriptional downregulation precedes protein loss. Moreover, this subgroup was enriched for TCF4-repressed, SPI1-activated and 'TNF signaling via NF-κB' pathway genes, at levels comparable to CD123[int] DCs (Fig. 5e–g).

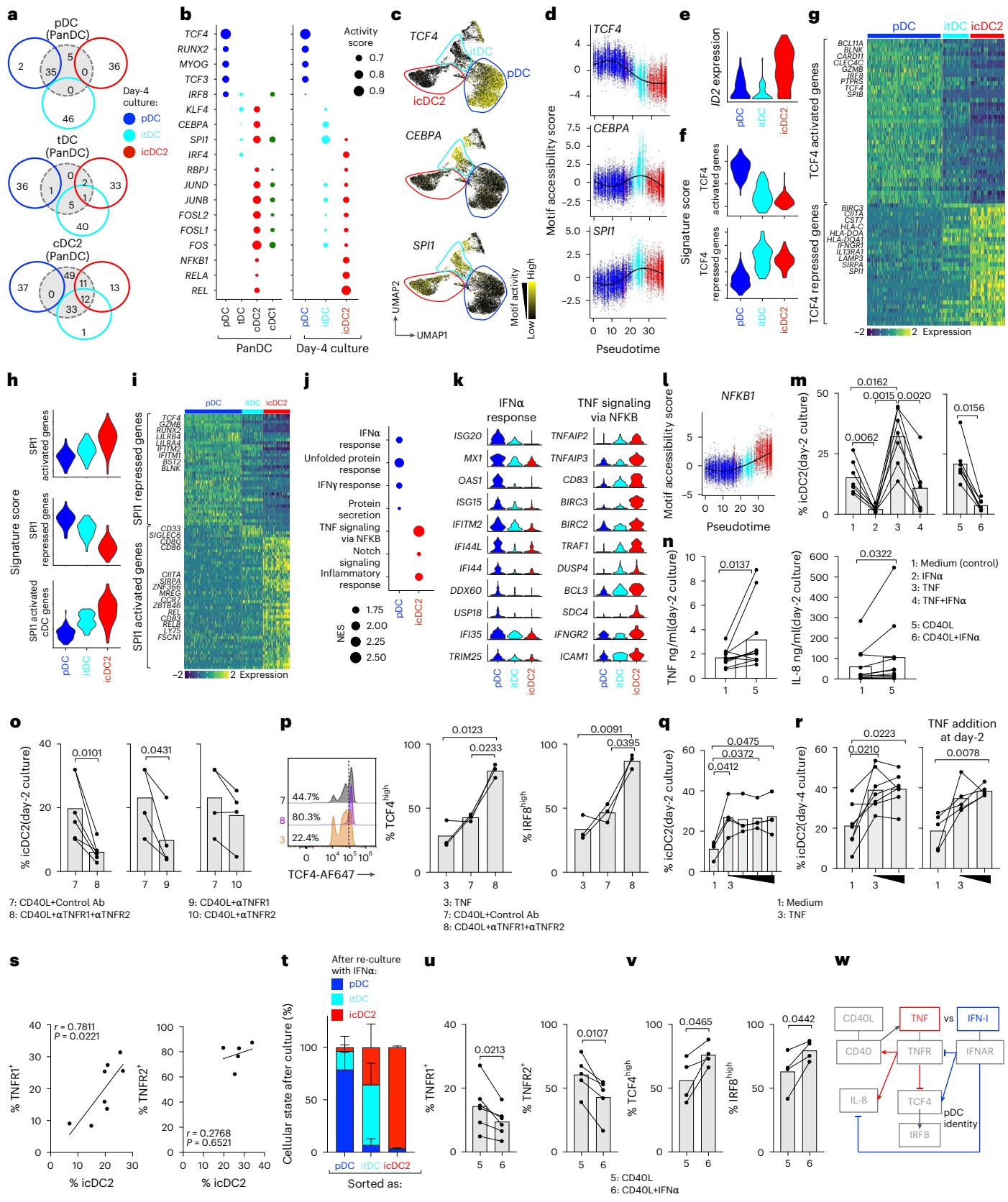

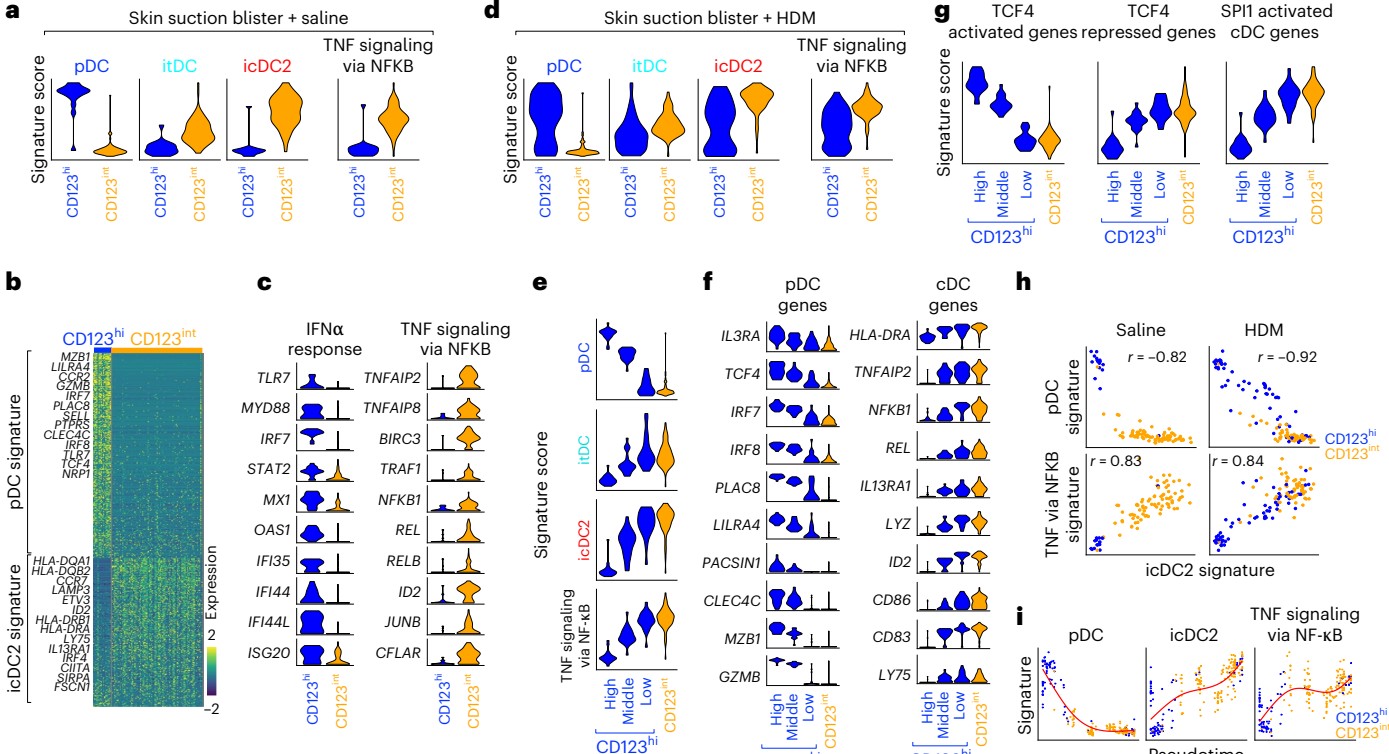

**Fig. 5 | Fate switching of pDCs occurs during wounding.** Analysis of a public SMART–seq2 dataset of CD123hi pDCs (blue) and CD123int DCs (orange) isolated from human skin blister fluid. Blisters were challenged with saline or HDM. **a**, Signature score of day 2-culture pDCs, itDCs and icDC2s (SMART–seq2) and the 'TNF signaling via NF-κB' MSigDB gene set projected onto the saline blister dataset. **b,c**, Expression of selected genes from **a** depicted as a heatmap (**b**) or violin plots (**c**). **d**, As in **a**, but signature scores onto the HDM-challenged blister dataset. **e**, Within the HDM blisters, CD123hi pDCs were split into 'high', 'middle' and 'low' groups based on the panDC-pDC signature score (Extended Data

Fig. 9i). Signature score of day 2-culture pDCs, itDCs and icDC2s (SMART–seq2) and the 'TNF signaling via NF-κB' MSigDB gene set. **f**, Expression of selected genes from **e**. **g**, Signature score of TCF4-regulated or SPI1-regulated cDC genes projected onto the HDM-challenged blister dataset. **h**, Correlation between the day 2-culture icDC2 transcriptional signature and the pDC or 'TNF signaling via NF-κB' signatures in saline and HDM blisters (Pearson's correlation). **i**, Pseudotime trajectory analysis (Monocle3): expression of indicated gene signatures along pseudotime (saline and HDM datasets combined). HDM, house dust mite.

These data suggest an in vivo pDC-to-icDC2 conversion trajectory, progressing through transcriptional intermediates (high, intermediate and low). These states become more prominent during HDM-induced inflammation. Correlation analysis confirmed that loss of the pDC program was tightly associated with acquisition of the icDC2 gene signature and 'TNF signaling via NF-κB' pathway upregulation (Fig. 5h). Consistently, pseudotime trajectory analysis predicted a transition from CD123hi to CD123int cells, supporting the presence of an in vivo differentiation continuum (Fig. 5i).

Altogether, these findings show that pDCs infiltrate sites of traumatic skin injury and undergo a transcriptional fate switching in vivo, mirroring the TNF-driven transition to icDC2s observed in vitro.

### Loss of identity of pDCs during aging

We next asked whether pDC fate switching occurs physiologically in healthy individuals. Aging is associated with elevated systemic TNF levels[45], reduced pDC numbers and impaired IFN-I production[46–48]. We hypothesized that these changes reflect a loss of pDC identity and acquisition of cDC2-like features. To test this, we performed bulk RNA sequencing (RNA-seq) and assay for transposase-accessible chromatin with sequencing (ATAC–seq) on purified pDCs from adult and elderly healthy donors (Extended Data Fig. 10a). Bulk profiling was chosen to detect subtle transcriptional and epigenetic changes between phenotypically similar cells. Strikingly, adult pDCs aligned with cultured pDCs, whereas elderly pDCs more closely resembled icDC2s (Fig. 6a). Pathway analysis revealed enrichment of 'IFNα response' genes in adult pDCs

and 'TNF signaling via NF-κB' ones in elderly pDCs (Fig. 6b), consistent with increased NF-κB TF activity (Fig. 6c and Extended Data Fig. 10b).

To evaluate these changes at the single-cell level, we performed CyTOF on an independent cohort. DC subsets were identified using unsupervised clustering and our established gating strategy (Fig. 6d–f and Extended Data Fig. 10c). Both adult and elderly pDCs clustered within the canonical pDC gate (Fig. 6g). However, direct comparison revealed that elderly pDCs expressed slightly lower levels of pDC markers and higher levels of cDC2-associated proteins (Fig. 6g,h). Notably, elderly pDCs exhibited reduced IRF8 expression (Fig. 6h), a TF regulated by TCF4 (refs. 21,33). Although CyTOF did not reveal differences in TCF4—likely due to batch effects—we quantified both TCF4 and IRF8 in two separate donor cohorts by flow cytometry, which enabled direct signal comparison. These analyses confirmed reduced TCF4 and IRF8 expression in elderly pDCs (Fig. 6i and Extended Data Fig. 10d). As previously reported[46], we also observed a significant reduction in pDC numbers in elderly donors (Fig. 6j).

Finally, we assessed the capacity of adult and elderly pDCs to undergo fate switching in response to CD40L. A higher proportion of elderly pDCs acquired an icDC2 phenotype in day 4 cultures and showed enhanced particulate antigen capture (Fig. 6k,l) (low elderly pDC yield precluded additional functional analyses).

Together, these findings demonstrate that pDCs from elderly individuals progressively lose lineage-defining features and become more prone to acquiring cDC2-like characteristics, supporting the physiological relevance of pDC fate switching during aging.

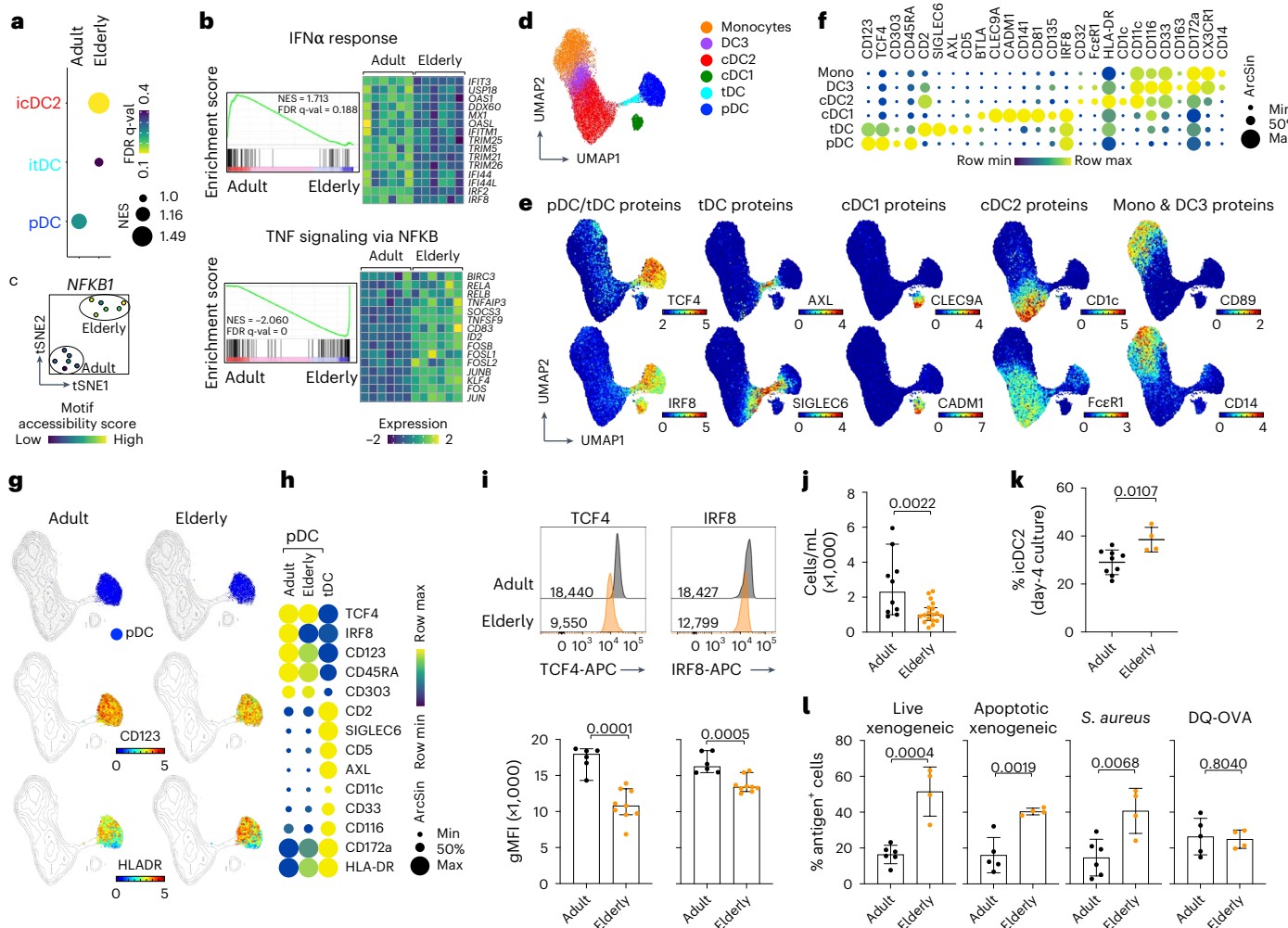

**Fig. 6 | Loss of pDC identity during aging. a–c,** Bulk RNA-seq and ATAC-seq were performed on sorted pDCs from six adult (age 24–30 years) and six elderly (age 73–89 years) donors (Extended Data Fig. 10a). **a,** Signature score of day 2 SMART-seq2 clusters projected onto bulk RNA-seq of adult and elderly donors. **b,** GSEA of the top two pathways differentially active between adult and elderly pDCs and heatmap of selected genes. **c,** Motif accessibility score of NF-κB1 computed from bulk ATAC-seq (Extended Data Fig. 10b). **d–h,** Circulating pDCs from adult (n = 6 donors, age 25–41 years) and elderly (n = 6 donors, age 62–78 years) donors analyzed by CyTOF. **d,** UMAP of DC populations clustered by FlowSOM. **e,** UMAPs colored by relative protein expression (ArcSin). **f,** Bubble plot of key protein markers distinguishing clusters in **d**. **g,** UMAPs highlighting pDCs (top), CD123 (middle) or human leukocyte antigen (HLA)-DR (bottom) expression in adult and elderly donors. **h,** Bubble plot of protein expression in pDCs from adult or elderly donors (tDCs plotted as controls). **i,** Geometric MFI of TCF4 and IRF8 in pDCs from adult (n = 6 donors) and elderly (n = 9 donors; 1 cohort out of 2; Extended

Data Fig. 10d) donors, by flow cytometry. **j,** Circulating pDCs from adult (n = 10 donors, 10 experimental) and elderly (n = 18 donors, 18 experimental) donors by flow cytometry. **k,l,** FACS-purified pDCs from adult (age 27–40 years; n = 21 experimental) and elderly (age >75 years; n = 4 experimental) donors cultured with CD40L for 4 d. **k,** Percentage of icDC2s from adult (n = 9 donors, 9 experimental) and elderly (n = 4 donors, 4 experimental) donors. **l,** Uptake of live xenogeneic cells (n = 6 adult donors in 6 experimental and 4 elderly donors in 4 experimental), apoptotic xenogeneic cells (n = 5 adult donors in 5 experimental and 4 elderly donors in 4 experimental), *S. aureus* (n = 6 adult donors in 6 experimental and 4 elderly donors in 4 experimental) and processing of DQ-OVA (n = 5 adult donors in 5 experimental and 4 elderly donors in 4 experimental) in day 4 cultures relative to controls. The statistical tests show mean ± s.d. throughout; two-sided, unpaired Student's t-test (**i,j,k,l**). t-SNE, t-distributed stochastic neighbor embedding.

## Discussion

We demonstrated that fully differentiated pDCs can undergo fate switching on activation, rewiring their transcriptional programs to acquire cDC2-like identity and function. Stringent purification and single-cell multiomic profiling ruled out contamination by tDC or cDC precursors. Instead, TNF initiated a reprogramming process that erases pDC identity—a phenomenon observed in vivo during inflammation, wounding and aging.

The relationship between pDCs and cDCs has long been debated, with recent controversy centered on pDC ontogeny and proposals to reclassify pDCs as innate lymphocytes[6,7]. Our findings support retaining their classification as DCs. We show that pDCs naturally remodel their chromatin to become icDC2s, acquiring the migratory

and T cell-priming functions characteristic of cDC2s. In inflamed skin, pDC-derived icDC2s align transcriptionally with cDC2s—not pDCs—a pattern also observed during wounding and aging. This capacity to undergo fate switching highlights the developmental proximity between pDCs and cDCs[2,27].

Previous observations suggested that TNF and IFN-I regulate pDC function. TNF induces antigen-presenting pDCs[20,39,49] that accumulate in the TNF-rich synovium of patients with rheumatoid arthritis[50]. Conversely, TNF blockade triggers IFN-I overexpression in pDCs and lupus-like symptoms[39,51]. Our findings provide a mechanistic framework for these observations by identifying a fate decision bifurcation based on cytokine sensing. TNF promotes fate switching by down-regulating TCF4, whereas IFN-I blocks it by suppressing TNFR1 or

-2 and sustaining TCF4 expression. These observations raise a key question: can pDCs convert into icDC2s during viral infections, where IFN-I is abundant? Based on our data, we propose that the balance and timing between IFN-I and TNF may be critical variables enabling this process. Notably, although pDCs gain some cDC-like traits during viral infection, they often retain core pDC identity[19]—likely due to sustained IFN-I signaling.

Compared with previous studies of pDC plasticity[20], our work presents a distinct conceptual framework for understanding pDC fate switching. We employed stringent pDC purification and clonal single-cell assays, ensuring that observed transitions were lineage intrinsic. In contrast to short-term activation (24–48 h), we performed longitudinal single-cell multiomics across 2–6 d, revealing a dynamic trajectory from pDCs to cDC2-like cells via a proliferative intermediate. Notably, fate switching was minimal at 24 h and previously reported activation markers (CD80 and PD-L1 (ref. 20)) did not map to pDCs, itDCs or icDC2s (Extended Data Fig. 8k). Rather than supporting a diversification model[20], our data demonstrate a unidirectional differentiation program involving transcriptional and chromatin remodeling. We show that icDC2s acquire bona fide antigen presentation capacity—including antigen uptake, processing and autologous naive T cell activation—functional assays that go beyond surrogate readouts like mixed leukocyte reactions, which reflect MHC-II expression, but not antigen uptake or processing. Importantly, we identified TNF–IFN-I crosstalk that governs fate switching, providing mechanistic insight into how pDCs integrate environmental cues. In sum, although previous studies focused on early transcriptional responses and potential diversification during viral infection[20], we defined a distinct, cytokine-driven, fate-switching program that culminates in stable acquisition of cDC2-like identity and function.

Loss-of-function mouse models show that DCs can change their fate on TF perturbation[1,2]. TCF4 is required for pDC development[33] and its deletion in mature pDCs leads to identity loss and acquisition of cDC traits[34]. SPI1 is necessary for DC development[52] and its expression in committed cDCs maintains identity while silencing the pDC program[37]. Thus, DC identity can be re-specified through manipulation of key TFs. However, it remains unclear under which physiological conditions fate switching occurs. Here we have shown that pDC reprogramming can occur physiologically, without genetic perturbation—through cytokine exposure. Fate switching progresses in a stepwise manner, through a proliferative itDC state, consistent with observations that dividing cells are more permissive to epigenetic changes[23]. The itDCs exhibit activity of pioneer TFs, for example, C/EBPα (CCAAT/enhancer-binding protein α), which may destabilize pDC lineage commitment and synergize with other cDC-specific TFs, a mechanism previously described in B cell-to-macrophage transdifferentiation[53]. Among C/EBPα-regulated TFs is SPI1 (refs. 53,54), a nonclassic pioneer TF. C/EBPα and SPI1 may cooperate to establish enhancers that facilitate the NF-κB-mediated response to TNF[55]. In sum, C/EBPα, SPI1, TCF4 and NF-κB form a transcriptional network that may orchestrate pDC-to-icDC2 fate switching. Dissecting this regulatory circuitry will be essential for understanding how pDC identity is maintained or rewired in different contexts.

Our alignment with in vivo datasets confirms the physiological relevance of pDC fate switching. In skin suction blister samples[44], CD123[int] DCs align transcriptionally with icDC2s, whereas CD123[hi] pDCs showed intermediate states, delineating the conversion trajectory. Notably, TNF sensing appeared critical for the conversion of infiltrating pDCs during sterile skin inflammation. This observation calls for a re-evaluation of TNF-driven skin diseases where activated DCs have been described. Importantly, we also show that pDC fate switching occurs during healthy aging: pDCs from elderly donors express lower levels of TCF4 and IRF8 and convert more readily into functional icDC2s, linking immune aging to the physiological erosion of pDC identity.

Altogether, our findings reveal that pDCs are not fixed, but versatile, capable of switching identity in response to environmental cues. This plasticity is governed by a tunable transcriptional network integrating TNF and IFN-I signals. The growing repository of pDC-specific, lineage-tracing mouse models[56] will be invaluable for dissecting these mechanisms in mouse models and for testing therapeutic strategies that harness pDC fate switching in disease.

## Online content

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

## Methods

### Human PBMCs

De-identified blood (collected using EDTA-coated tubes; BD Biosciences) and buffy coats from healthy adults (age 20–40 years) and elderly donors (age >65 years; Supplementary Table 6) were obtained through local lab-led blood donation efforts following Stanford University and University of California San Diego (UCSD) guidelines, or from the Stanford Blood Center. Donors provided informed consent under protocols approved by the institutional review boards of Stanford University and UCSD and did not receive compensation. The number of human donors is indicated in each figure legend. When provided, information on donor age and sex is reported in Methods (sections: 'snMultiome-seq sample preparation', 'SMART-seq2 dataset generation' and 'Bulk RNA-seq and ATAC-seq dataset generation') or in the supporting data (Supplementary Table 6). Samples were not selected based on sex, age (beyond the range described above), race or other individual characteristics. Blood was processed within 2 h of procurement. PBMCs were isolated by density gradient centrifugation using Ficoll-Paque PLUS (GE Healthcare), according to the manufacturer's instructions. Unless otherwise specified in the figure legends, cells were used immediately for experimentation.

### DC isolation

DCs were isolated from fresh PBMCs by negative magnetic enrichment followed by FACS sorting. PBMCs were treated with human γ-globulin (Invitrogen) for 15 min on ice to block nonspecific binding, then incubated with antibodies against CD3, CD19, CD335 and CD66b, followed by anti-mouse magnetic Dynabeads (Thermo Fisher Scientific), as described[57]. PanDCs and pDCs were enriched from PBMCs using the EasySep Human PanDC or Plasmacytoid DC Isolation Kits (STEMCELL Technologies) according to the manufacturer's instructions. For FACS purification, enriched cells were stained with an antibody cocktail for 30 min on ice and sorted using FACSAria II or Fusion (BD Biosciences).

### In vitro culture of pDCs

Sorted pDCs, 10,000, were cultured in 96-well U-bottomed plates in 200 µl of R10 complete medium (Roswell Park Memorial Institute (RPMI) medium (Corning) with 10% heat-inactivated fetal bovine serum (FBS; Gibco), 2 mM L-glutamine, 100 IU ml⁻¹ of penicillin, 100 µg ml⁻¹ of streptomycin, 10 mM Hepes, 1 mM sodium pyruvate, 1X MEM Non-Essential Amino Acids (all Corning) and 55 µM 2-mercaptoethanol (Gibco)) at 37 °C. All cultures contained 10 ng ml⁻¹ of recombinant human IL-3 (R&D Systems; carrier free) for pDC survival. Activation stimuli included 100–200 ng ml⁻¹ of CD40L (R&D Systems; carrier free), 2–2,000 ng ml⁻¹ of TNF (BioLegend; carrier free), 10–1,000 U ml⁻¹ of IFNα (PBL Assay Science) or 5 µg ml⁻¹ of CpG-A (Invivogen, cat. no. ODN 2216). For IFN-I blockade, 1,000 ng ml⁻¹ of B18R (R&D Systems) was added. For TNF blockade, pDCs were pre-incubated for 1 h with 10 µg ml⁻¹ of anti-TNFR1 (R&D Systems, clone 16805), anti-TNFR2 (R&D Systems, clone 22210) or isotype control before CD40L stimulation. Secreted TNF and IL-8 were measured in day 1, 2 or 4 supernatants by cytometric bead array Human Enhanced Sensitivity kit (BD Biosciences). For IFN-I detection, sorted DCs were cultured with 150 µl of R10 complete medium + IL-3 + 5 µg ml⁻¹ of CpG-A for 24 h. Supernatants were frozen at −80 °C and analyzed with VeriKine Human IFN Alpha Multi-Subtype ELISA Kit (PBL Assay Science).

### Flow cytometry

Antibodies for flow cytometry were purchased from BioLegend, R&D Systems, BD Biosciences and Thermo Fisher Scientific (Supplementary Table 5). Cells were incubated with human γ-globulin for 15 min on ice to block nonspecific binding, then stained for 20 min at 4 °C with surface markers diluted in FACS buffer (2 mM EDTA and 2% donor equine serum in phosphate-buffered saline (PBS)). CCR2 and CCR7 were stained separately at 37 °C for 45 min in PBS. Cells were acquired on a 5-laser

LSRFortessa X-20 (BD Biosciences) using FACS Diva software (v.8.01) or a Cytek Aurora (Cytek) using Cytek SpectroFlo (v.3.3.0). Compensation was set up using beads (BD Biosciences) and dead cells excluded by DAPI or LIVE/DEAD Fixable Blue (Thermo Fisher Scientific) staining. For Ki67, TCF4 and IRF8, cells were surface stained, fixed with Foxp3 Fix/Perm Buffer (Thermo Fisher Scientific) for 1 h and intracellularly stained for 45 min in 1× Permwash buffer (Thermo Fisher Scientific). Positive gates were defined using an isotype and fluorescent minus one controls. For proliferation assays, PBMCs were labeled with 2.5 µM CTV (Thermo Fisher Scientific) for 10 min at 37 °C before enrichment and sorting. Apoptosis was measured with Apotracker Green (BioLegend). Data were analyzed using FlowJo software (Tree Star) and plotted using GraphPad Prism 10 (GraphPad Software).

### Single-cell differentiation

Half of the enriched pDCs were labeled with 1.7 nM CFSE (Sigma-Aldrich) and half with 2.5 µM CellTrace Violet (CTV, Thermo Fisher Scientific) for 10 min at 37 °C. Carboxyfluorescein succinimidyl ester (CFSE)-labeled pDCs ('filler' cells) were plated at 5,000 cells per well in 96-well plates. CTV-labeled pDCs were stained with antibodies for 30 min on ice and FACS purified into wells at 1, 10, 100, 1,000 or 5,000 cells per well. Conversion was assessed on CFSE⁺CTV⁻ (filler) and CFSE⁻CTV⁺ (experimental) cells after 4 d of culture (Extended Data Fig. 4g).

### CyTOF

CyTOF was performed as previously described[15,16,21,57]. Fresh PBMCs and day 6-culture pDCs were pooled with mouse splenocytes ('cell bed'), stained with 0.25 mM cisplatin (Fluidigim), surface stained with heavy-metal-labeled antibodies, fixed with Foxp3 Fix/Perm Buffer (Thermo Fisher Scientific) and stained intracellularly. Cells were incubated overnight with 2% paraformaldehyde (Electron) in PBS with 125 nM iridium intercalator (Fluidigm), washed, filtered and acquired in a CyTOF2 (Fluidigm) at the Shared FACS Facility at Stanford University. Flow cytometry standard files were normalized using the Premessa R package. A reference map of PBMC pDCs, tDCs, cDC1s and cDC2s (gated in FlowJo) was generated using Scaffold (v.0.1) (https://github.com/nolanlab/scaffold)[58]. Cells clustering within pDC, itDC or icDC2 nodes were input into marker enrichment modeling analysis[59], using arcsinh transformation (cofactor = 15). Heatmaps were generated with GraphPad Prism 10. For the aging dataset, PBMCs were thawed in batches and CD45 barcoded to allow simultaneous processing of adult, elderly and internal standard samples (for batch correction). Normalized flow cytometry standard files were gated (Extended Data Fig. 10c), followed by UMAP and FlowSOM analyses in FlowJo. Protein expression heatmaps (arcsinh transformed) were visualized in Morpheus (Broad Institute) and other data visualization was done using CYT (SightOf) with MATLAB (MathWorks).

### Antigen uptake and processing

Freshly isolated DCs (enriched with EasySep Human PanDC kit) or day 4-culture cells were assessed for uptake of live or apoptotic autologous cells, xenogeneic cells and bacteria. For autologous uptake, PBMCs were labeled with 2 µM PKH26 (Sigma-Aldrich) at 20 × 10⁶ cells ml⁻¹ for 5 min at room temperature. Apoptosis was induced by 60 Gy of X-ray irradiation. For xenogeneic uptake, C57BL/6 mouse splenocytes were labeled with PKH26 and apoptosed by osmotic shock: hypertonic medium (0.5 M sucrose (Sigma-Aldrich), 10% w/v poly(ethylene glycol) 1000 (Alfa Aesar) and 10 mM Hepes (Corning) in RPMI medium) for 10 min at 37 °C, followed by hypotonic medium (40% endotoxin-free water (Cytiva) and 60% RPMI medium) for 2 min at 37 °C. For bacteria uptake, cells were incubated with 50–100 µg ml⁻¹ of pHrodo Red *S. aureus* (Thermo Fisher Scientific). In all cases, DCs were incubated with particulate antigen for 3 h at 37 °C and uptake measured by flow cytometry as %PKH26⁺ or pHrodo⁺ cells. For antigen processing, DCs were incubated with 0.05–0.50 µg ml⁻¹ of DQ-OVA (Invitrogen) for

3 h at 37 °C. Negative controls included pretreatment with 5 µg ml⁻¹ of cytochalasin D (Sigma-Aldrich) for 30 min at 37 °C, followed by 3 h incubation at 4 °C.

## T cell priming

For naive T cell preparation, frozen PBMCs were thawed, washed twice with PBS, labeled with 1.7 nM CFSE or 2.5 µM CTV at 37 °C for 10 min and washed with R10 medium. $CD3^+CD45RA^+CD45RO^-$ naive T cells were isolated (>98%) using EasySep Human Naïve Pan T cell isolation Kit (STEMCELL Technologies) per the manufacturer's instructions (Extended Data Fig. 5h). Day 4-culture DCs were incubated ± apoptotic mouse splenocytes for 3–18 h (1 DC to 300 splenocytes) and FACS purified (Extended Data Fig. 5g). Sorted pDCs and icDC2s were cocultured with CFSE- or CTV-labeled autologous naive T cells (1:20 ratio) for 6 d in the presence of IL-3. For homeostatic proliferation controls, T cells were cultured alone in IL-3. For assays using freshly isolated DCs, PBMCs enriched from DCs were FACS purified (Extended Data Fig. 6c) and cocultured with CFSE-labeled autologous naive T cells (1:20) for 6 d in the presence of IL-3 and apoptotic mouse splenocytes (1 DC to 300 splenocytes).

## Migration assay

Day 4-culture cells were harvested, counted and plated in the upper well of 5-µm pore size, 96-well transwells (Corning) with IL-3 (10 ng ml⁻¹). Lower wells contained IL-3 alone (control), IL-3 + 50 ng ml⁻¹ of CCL2 or 50 ng ml⁻¹ of CCL19 and CCL21 (all chemokines from Thermo Fisher Scientific). Cells were incubated for 3 h at 37 °C, then harvested from the upper and lower wells, stained and counted using CountBright beads by flow cytometry. Migration was calculated as the percentage of each cell state migrating relative to the total cell number.

## Cytology and scanning electron microscopy

Fresh DC subsets and day 6-culture cells were FACS purified for cytology and scanning electron microscopy analysis. For cytology, sorted cells were cytospun onto poly(lysine)-coated glass slides (Shandon Polysine, Thermo Fisher Scientific) and stained using the Three-Step Stain Set (Thermo Fisher Scientific) per the manufacturer's instructions. Images were acquired with a BZ-X800 microscope (Keyence) at ×60 and cell size and circularity were quantified using ImageJ (National Institutes of Health (NIH)). For scanning electron microscopy, sorted cells were fixed overnight at 4 °C in Karnovsky's fixative (2% glutaraldehyde and 4% paraformaldehyde in 0.1 M sodium cacodylate, pH 7.4 buffer (all from EMS)), washed 3× and treated with 1% osmium tetroxide (EMS) for 1 h at room temperature. Cells were washed, dehydrated through graded ethanol (50%, 70%, 95% and 100%, 2×), and resuspended in hexamethyldisilazane (EMS). Air-dried cells were mounted onto poly(lysine)-coated 12-mm coverslips, sputter coated with 4-nm gold at 10° using a Leica ACE600 and imaged with a Zeiss Sigma FESEM at 3 kV at the Stanford Cell Sciences Imaging Facility.

## snMultiome–seq sample preparation

snMultiome–seq was performed on freshly isolated DCs (1 male donor, age 32 years), pDCs (1 male donor, age 32 years) and day 4-culture pDCs (1 male donor, age 29 years) (Extended Data Figs. 1a and 2a,c). Nuclei were isolated from FACS-sorted cells using the 10x Genomics protocol (no. CG000365) adapted for low input. Briefly, 100,000 cells were lysed with chilled multiome lysis buffer (10 mM Tris-HCl, pH 7.4 (TEKnova), 10 mM NaCl (Thermo Fisher Scientific), 3 mM $MgCl_2$ (Sigma-Adrich), 0.1% Tween-20 (Roche), 0.1% IGEPAL (Sigma-Aldrich), 0.01% digitonin (Promega), 1% bovine serum albumin (BSA) (Sigma-Aldrich), 1 mM dithiothreitol (Thermo Fisher Scientific) and 1 U µl⁻¹ of RNase inhibitor (Sigma-Aldrich) in nuclease-free water) for 3 min on ice. Nuclei were washed twice with multiome wash buffer (identical but without IGEPAL and digitonin) and resuspended in multiome nuclei buffer (1× nuclei (10x Genomics), 1 mM dithiothreitol and 1 U µl⁻¹ of RNase inhibitor prepared in nuclease-free water).

## snMultiome–seq library preparation and data analysis

Single-cell libraries were prepared at the Stanford Genomics Core using the Chromium Next GEM Single Cell Multiome ATAC + Gene Expression kit (10x Genomics) and sequenced on an Illumina NovaSeq 6000. Data were processed with Cell Ranger ARC (v.2.0.0 and v.2.0.2) using the GRCh38 reference genome. Downstream analyses were done with Seurat (v.5.0.2) and Signac (v.1.12.9004). snRNA-seq data were normalized ('LogNormalize', scale factor 10,000) and scaled ('ScaleData') before calculating the UMAP. snATAC–seq data were peak called with MACS2 (v.2.2.9.1), filtered to remove nonstandard chromosomes and blacklist regions, normalized with RunTFIDF, reduced by RunSVD (latent semantic indexing) and visualized by UMAP. The weighted nearest neighbor (WNN) graphs were constructed using FindMultiModalNeighbors. PanDC and pDC snRNA-seq datasets were batch corrected with IntegrateLayers (HarmnoyIntegration method) in Seurat (Fig. 1a–c). snATAC–seq datasets were integrated with FindIntegrationAnchors and IntegrateEmbeddings in Signac (Fig. 1d). Cells with abnormal RNA-seq or ATAC-seq quality control scores were excluded during quality control.

## SMART–seq2 dataset generation

Freshly isolated pDCs, tDCs and day 2-culture cells from two donors (1 male donor, age 36 years and 1 female donor, age 28 years) were profiled using SMART–seq2, which performs better for detecting low-expression genes[60]. Mononuclear phagocytes were enriched from PBMCs using antibodies against CD3, CD19, CD335 and CD66b, followed by anti-mouse magnetic Dynabeads (Thermo Fisher Scientific). Cells were stained and single-cell sorted into 96-well plates as pDCs (DAPI⁻CD3⁻CD19⁻CD20⁻CD335⁻CD66b⁻CD14⁻CD16⁻HLA-DR⁺CD123⁺AXL⁻CD33⁻BDCA4⁺CD11c⁻) and tDCs (DAPI⁻CD3⁻CD19⁻CD20⁻CD335⁻CD66b⁻CD14⁻CD16⁻HLA-DR⁺CD123⁺AXL⁺CD33⁺). Sorted pDCs from the same donor were cultured for 2 d and sorted as pDCs (DAPI⁻HLA-DR⁺CD33⁻CD11c⁻), itDCs (DAPI⁻HLA-DR⁺CD33⁺CD11c⁻) and icDC2s (DAPI⁻HLA-DR⁺CD11c⁺) (Fig. 2e). Reverse transcription, complementary DNA synthesis and amplification were performed using the Takara Smart-Seq Single Cell Kit. The cDNA was cleaned with Ampure XP beads (Beckman Coulter) and quantified with Quant-iT PicoGreen dsDNA Assay (Thermo Fisher Scientific). Libraries were prepared using the Illumina Nextera XT DNA Library Prep and Unique Dual Index Kits (Illumina) and sequenced paired end (75 bp) on a Hi-seq4000 (Illumina). Reads were aligned to the UCSC hg19 transcriptome using STAR and gene counts were determined using featureCounts. A total of 383 cells from healthy donors were sequenced; after quality control filtering, 377 high-quality cells remained. Data were normalized and scaled ('LogNormalize', scale factor 10,000; 'ScaleData') using Seurat (v.5.0.1) before UMAP generation.

## Bulk RNA-seq and ATAC–seq dataset generation

The pDCs were enriched from freshly isolated PBMCs (Extended Data Fig. 10a). Then, 10,000 pDCs from 6 adult (age 24–30 years) and 6 elderly (age 73–89 years) donors were sorted directly into QIAzol lysis buffer (QIAGEN) and frozen at −80 °C for RNA-seq. RNA extraction and library preparation were performed by the Stanford Functional Genomics Core. For ATAC–seq, 10,000 pDCs were sorted into R10 and DNA was extracted and libraries prepared using the Omni-ATAC protocol at the Stanford Functional Genomic Core. DNA was stored at −20 °C after transposition until all the samples had been collected. Amplification and quantitative PCR were performed simultaneously across samples. Library quality was assessed by Bioanalyzer. The 24 samples were barcoded, pooled and sequenced on a NovaSeq 6000 at the Stanford Functional Genomics Core.

## DEG analysis

For single-cell sequencing datasets, differentially expressed genes (DEGs) between clusters were identified using Wilcoxon's rank-sum test in Seurat's FindMarkers function. DEGs (P < 0.05, min.pct = 0.25,

log$_2$(fold-change) (log$_2$(FC)) > 0) from the panDC dataset were used to generate transcriptional signatures for each DC subset (Supplementary Table 1). DEGs (*P* < 0.05, min.pct = 0.25, log$_2$(FC) > 1) from clusters C1–C3 in the day 4 snMultiome–seq dataset were used to define cellular states (pDCs, itDCs and icDCs; Supplementary Table 2). Additional signatures were derived from published datasets (Extended Data Figs. 1g–i, 4h and 7c,d): accession nos. GSE94820 (ref. 14), GSE35457 (Supplementary Table 2)[13], GSE132566 (Supplementary Table 2 and Fig. 6; CD5$^+$CD163$^-$ cDC2 versus CD14$^-$CD163$^+$ DC3)[22], GSE75650 (ref. 35) (TCF4 silencing) and GSE121446 (ref. 37) (*Spi1$^{-/-}$*). Murine genes were converted to human orthologs using g:Profiler[61]. Gene signature scores were calculated with AddModuleScore or AddModuleScore_UCell functions in Seurat. Triwise (v.0.99.5) was used to analyze gene expression changes during pDC conversion (Fig. 1i). Normalized gene expression was used to calculate barycentric coordinates with transformBarycentric and data were plotted on a hexagonal polygon[62]. Cell cycle scores were calculated using CellCycleScoring with Seurat using default S and G2/M phase genes (Extended Data Figs. 2e and 3c).

### Differentially open chromatin regions and TF activity

For single-cell datasets, differentially accessible chromatin regions between DC subsets were identified using FindMarkers (Signac) (*P* < 0.05, min.pct = 0.1, log$_2$(FC) > 0). These regions were used to compute chromatin accessibility scores with AddModuleScore (Seurat; Fig. 1d). *CIITA* enhancers were visualized based on peak-gene linkages (LinkPeaks default settings; Signac). *MARCH1* enhancers were selected based on the GeneHancer database (v.4.4)[63] and overlapping open peaks. Motif accessibility was computed using chromVAR (v.1.22.1) with the UCSC hg38 genome (BSgenome.Hsapiens.UCSC.hg38) and the JASPAR2020 motif collection via RunChromVAR (Signac). Presto (Wilcoxon's rank-sum test) was used to rank TF activity by integrating gene expression and motif accessibility, with activity ranked by the average Presto area-under-the-curve statistic. TF activity signatures were created by filtering TFs with average area under the curve >0.6 for: (1) each DC subset in the panDC dataset and (2) pDCs, itDCs and icDC2s in the day 4-culture dataset (Extended Data Fig. 7a).

### Pseudotemporal trajectory analysis

Monocle3 (v.2.28.0) and Slingshot (v.2.8.0) were used to study pseudotemporal trajectories in the day 4-culture snMultiome–seq dataset. Processed expression data and WNN UMAP coordinates from Seurat were loaded into Slingshot or Monocle3. For Slingshot, trajectories were learned using getLineages, assigning cluster C1 (pDCs) as the starting point. For Monocle3, trajectories were learned using learn_graph and pseudotime assigned with order_cells, setting cluster C1 as the root. To study motif accessibility over pseudotime (Fig. 5i), Monocle3 output was imported into Seurat, and trends were visualized using FeatureScatter and ggplot2 with geom_smooth(method = 'lm', formula = *y* ~ splines::ns(*x*, df = 3)).

### GSEA

DEGs were identified from the day 2-culture SMART–seq2 dataset by comparing cluster C1 (pDCs) versus C3 (icDC2s) using Seurat's FindMarkers function (min.pct = 0, logfc.threshold = 0) to extract all genes. Genes were ranked by log$_2$(FC) and analyzed with GSEA Pre-ranked (GSEA v.4.2.3) using 1,000 permutations. GO, Hallmark and canonical pathways from MSigDB were used as input gene sets. For the aging bulk RNA-seq dataset, DEGs between adult and elderly pDCs were identified using DESeq2, ranked by log$_2$(FC) and analyzed with GSEA Pre-ranked for Hallmark and canonical pathways. To compare transcriptional changes in DC subsets with aging, a separate GSEA Pre-ranked analysis was performed using the adult versus elderly DEG list, with day 2 SMART–seq2 DC transcriptional signatures (Supplementary Table 2) used as input gene sets.

### Bulk RNA-seq and ATAC–seq

For bulk RNA-seq analysis, reads were aligned to hg19 using STAR on Stanford's Sherlock computing cluster, and counts were generated with featureCounts. Differential gene expression was performed using DESeq2. For bulk ATAC–seq analysis, reads were quality controlled, trimmed, aligned, filtered and normalized using nfatac (https://nf-co.re/atacseq/2.1.2) on Sherlock. Reads were aligned to hg19 and narrow peak calling and consensus peak generation were performed with nfatac. Differential peak analysis was conducted with DESeq2 and TF activity was inferred using ChromVAR (https://greenleaflab.github.io/chromVAR/).

### Transcriptomics of skin inflammation and skin wounding

Publicly available CITE–seq (EGAS00001005271)[43] (Extended Data Fig. 9) and SMART–seq2 (E-MTAB-8498)[44] (Fig. 5) datasets were downloaded and analyzed in R using Seurat. Pre-processed data from EGAS00001005271 were downloaded (https://explore.data.humancellatlas.org/projects/5bd01deb-01ee-4611-8efd-cf0ec-5f56ac4/project-matrices) and subsetted to include myeloid cell clusters based on *HLA-DRA$^+$* and *MS4A1$^-$* gene expression (Extended Data Fig. 9a). Donor skin conditions included healthy controls and those with atopic dermatitis, psoriasis vulgaris, bullous pemphigoid, lichen planus and indeterminate rashes. Gene signatures from panDC and SMART–seq2 day 2-culture datasets were scored using Seurat's AddModuleScore function. The E-MTAB-8498 dataset was downloaded (https://explore.data.humancellatlas.org/projects/67a3de09-45b9-49c3-a068-ff4665daa50e/project-metadata) and converted into a Seurat object. Quality control and data preprocessing followed the original publications. Data were normalized ('LogNormalize', scale factor 10,000), scaled ('ScaleData') and UMAPs generated. The dataset was subsetted to BDCA2$^+$ pDCs and gene signatures analyzed using AddModuleScore. Gene sets for 'IFNα response' and 'TNF signaling via NF-κB' were downloaded from MSigDB (v.2024.1.Hs). Pearson's correlations (Fig. 5h) were calculated using Seurat's FeatureScatter. Monocle3 was used to reconstruct pseudotemporal trajectories in the skin blister dataset.

### Statistical analysis

Blood samples were randomly allocated across experiments. Statistical details, including *n* value, number of experiments, *P* values and the type of statistical tests used for each experiment can be found in the figure legends. No technical replicates were used; in all cases, each data point represents a distinct donor (biological replicate). In experiments where a paired experimental design was used (that is, cells from the same donor were subjected to different conditions), this is visually indicated in the figures by lines connecting paired data points. No statistical methods were used to predetermine sample sizes, but the sample sizes were consistent with those used in previous publications[15,16,20,21,39]. Statistical analyses were performed with three or more biological replicates using GraphPad Prism 10 (GraphPad Software) or R (v.4.3.1). Data distribution was assessed using the Shapiro–Wilk normality test. Data collection and analysis were not performed blind to the conditions of the experiments and no data points were excluded. For single-cell RNA-seq, pre-established exclusion criteria—low number of unique genes, abnormally high read count or high mitochondrial gene content—were applied. All graphs with error bars show mean ± or + s.d. values. Violin plots, UMAPs and heatmaps were generated in R (v.4.3.1) using Seurat (v.5.0.2), Signac (v.1.12.9004), ggplot2 (v.3.5.0) and viridis (v.0.6.5) packages or with GraphPad Prism 10 (GraphPad Software).

### Reporting summary

Further information on research design is available in the Nature Portfolio Reporting Summary linked to this article.

## Data availability

Data generated in this study have been deposited in the National Center for Biotechnology Information's Genome Expression Omnibus under the following accession nos.: GSE267100 (GRCh38 reference genome), GSE267099 (GRCh38), GSE267174 (GRCh38), GSE266889 (hg19), GSE269411 (hg19) and GSE279911 (hg19). Publicly available CITE–seq data (EGAS00001005271)[43] were download from https://explore.data.humancellatlas.org/projects/5bd01deb-01ee-4611-8efd-cf0ec-5f56ac4/project-matrices. Publicly available SMART–seq2 data (E-MTAB-8498)[44] were downloaded from https://explore.data.human-cellatlas.org/projects/67a3de09-45b9-49c3-a068-ff4665daa50e/project-metadata. All other data generated in this study are included in the Article and Supplementary Information. Source data are provided with this paper.

## Code availability

This study did not generate unique codes.

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

## Acknowledgements

Work was supported by the NIH (grant nos. AI158808 and AI163775 to J.I.). R.A.H. was supported by the Dean's fellowship (Stanford), R.M.-A. and M.A.S.-B. by SECTEI (CDMX, Mexico) and A.G.-B. by a KL2 Mentored Career Development Award (no. KL2TR003143). CyTOF and sequencing were performed using NIH-funded instruments (nos. S10OD016318-01, S10OD025212 and 1S10OD021763). We thank D. Seong (Stanford University) and M. Alcántara-Hernández (Stanford University) for their assistance with aging-related data collection and initial analysis, members of J.I.'s and O.M.M.'s lab for technical support and discussions and C. Murre (UCSD) for discussions.

## Author contributions

Conceptualization: R.A.H., R.A.L. and J.I. Methodology: R.A.H., R.M.-A., M.A.S.-B., R.A.L., O.M.C., A.G.-B. and J.I. Formal analysis: R.A.H., R.M.-A., M.A.S.-B., A.G.-B. and J.I. Investigation: R.A.H. Resources: O.M.M. Visualization: R.A.H. and J.I. Writing: R.A.H. and J.I. Funding acquisition: J.I. Supervision: J.I. Project administration: J.I.

## Competing interests

J.I. serves on the scientific advisory board of Immunitas Therapeutics; this affiliation is unrelated to the present work. The other authors declare no competing interests.

## Additional information

**Extended data** is available for this paper at https://doi.org/10.1038/s41590-025-02234-3.

**Correspondence and requests for materials** should be addressed to Juliana Idoyaga.

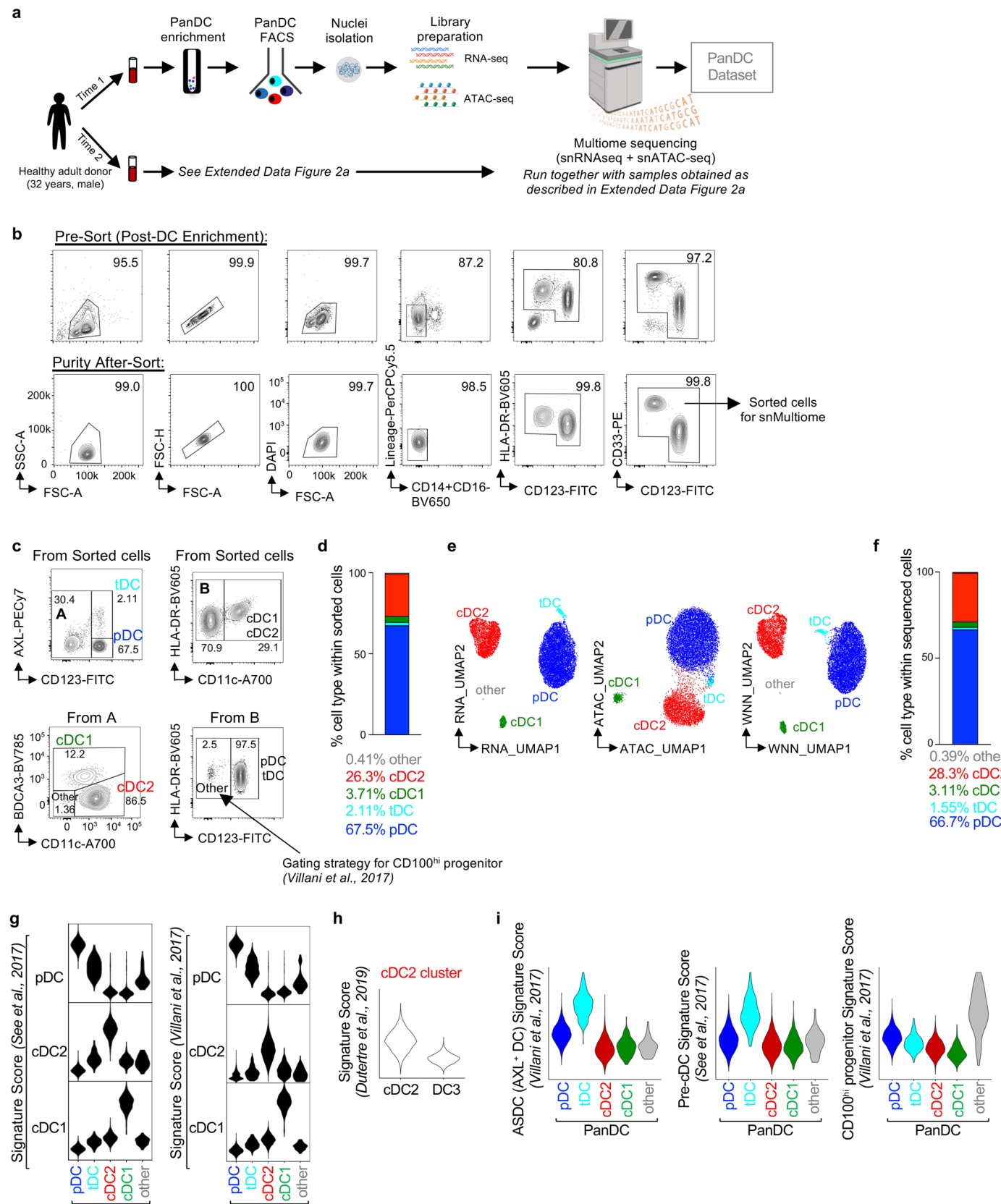

**Extended Data Fig. 1 | See next page for caption.**

**Extended Data Fig. 1 | Generation of a PanDC snMultiome-seq dataset of human dendritic cells. a**, Experimental design for generating panDC and sorted pDC snMultiome-seq datasets. DCs were magnetically enriched from fresh PBMCs, followed by FACS purification and nuclei isolation. snRNA-seq and snATAC-seq libraries were generated from nuclei and sequenced. **b**, PanDC purification strategy. Magnetically enriched DCs were stained and FACS-purified. Lineage-positive cells (CD3[+], CD19[+], CD20[+], CD335[+], CD66b[+]), CD14[+] and CD16[+] were excluded from live singlets, as were HLA-DR[−] and CD33[−]/CD123[−] cells. Representative plots before (top) and after (bottom) sorting are shown. **c-d**, Sorted panDCs were analyzed by flow cytometry to determine the frequency of each DC subset: pDCs (CD123[+]AXL[−]), tDCs (CD123[+]AXL[+]), cDC2s (CD123[−]CD11c[+]BDCA3[−]), cDC1s (CD123[−]BDCA3[+]) and "other" cells (CD123[−]CD11c[−]BDCA3[−]). Flow cytometry plots (**c**) and corresponding

quantification (**d**) are shown. Previously described CD100[hi] progenitors[14] are contained within the "other" population (HLA-DR[+]CD11c[−]CD123[−]). **e**, UMAP of panDCs based on snRNA-seq (left), snATAC-seq (middle), and their integrated profiles using weighted nearest neighbor (WNN) analysis (right). Clusters are colored by unsupervised WNN clustering. **f**, Bar graph showing the percentage of DC subset within clusters from (**e**). **g**, Violin plots showing DC subset gene signature scores based on two publicly available datasets[13,14], calculated using Seurat AddModuleScore function. **h**, DC2 (CD163[−]CD5[+] cells) and DC3 (CD163[+]CD14[+] cells) gene signatures[22] were applied to the cDC2 cluster of panDC dataset using the AddModuleScore_UCell function. **i**, Signature score of AXL[+] DCs and progenitor populations[13,14] applied to the panDC dataset using the AddModuleScore function. Graphics in **a** created using BioRender.com.

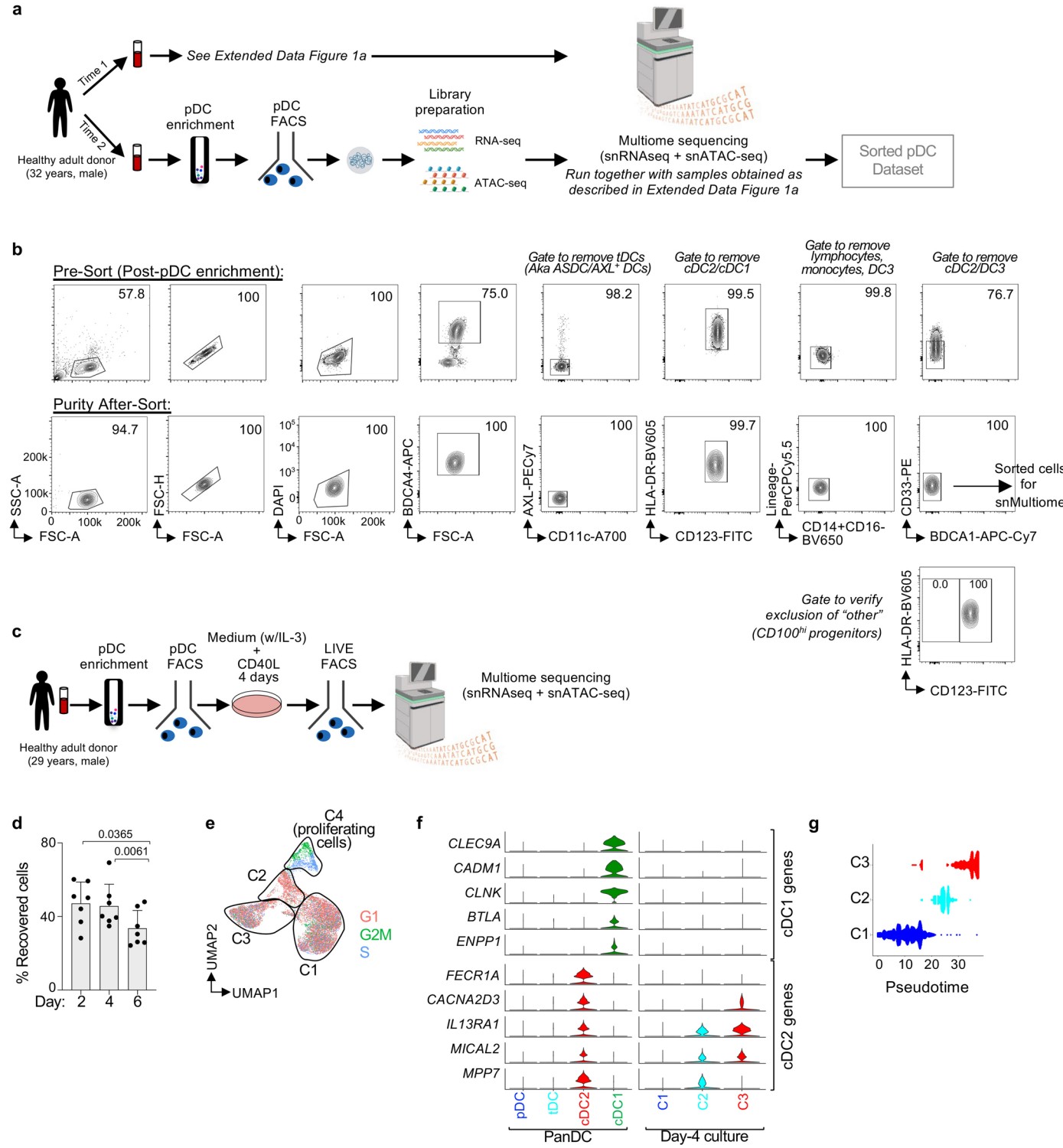

**Extended Data Fig. 2 | Generation of a snMultiome-seq dataset of human pDCs before and after CD40L stimulation. a**, As in Extended Data Fig. 1a, but showing the purification of pDCs from the same donor at a different time point. Briefly, pDCs were magnetically enriched from PBMCs, stained and purified by FACS. **b**, pDC purification strategy. Lineage-positive cells (CD3⁺, CD19⁺, CD20⁺, CD335⁺, CD66b⁺), CD14⁺ and CD16⁺ were excluded from live singlets. Cells were then gated as BDCA4⁺AXL⁻CD11c⁻HLA-DR⁺CD123⁺BDCA1⁻CD33ˡᵒʷ. Representative plots before (upper panels) and after (bottom) sorting are shown. Previously described CD100ʰⁱ progenitors[14] were not present in sorted pDCs. **c**, Experimental design for generating a snMultiome-seq dataset from CD40L-stimulated pDCs.

Purified pDCs were cultured with CD40L (always in the presence of IL-3) for 4 days, live cells were sorted and subjected to snRNA-seq and snATAC-seq library preparation and sequencing. **d**, Bar graph showing cell recovery after 2, 4, or 6 days of culture with CD40L (always in the presence of IL-3) (mean + SD; n = 7 donors in 7 exp.; paired one-way ANOVA with Tukey's multiple comparison test). **e**, WNN UMAP of day-4 culture snMultiome-seq dataset with cells colored by cell cycle stage based on Seurat analysis. **f**, Violin plots showing expression of selected cDC1 and cDC2 genes in panDC dataset (left) and day-4 culture dataset (right). **g**, Cells from 4 day-culture snMultiome dataset plotted by clusters and pseudotime trajectory. Graphics in **a** and **c** created using BioRender.com.

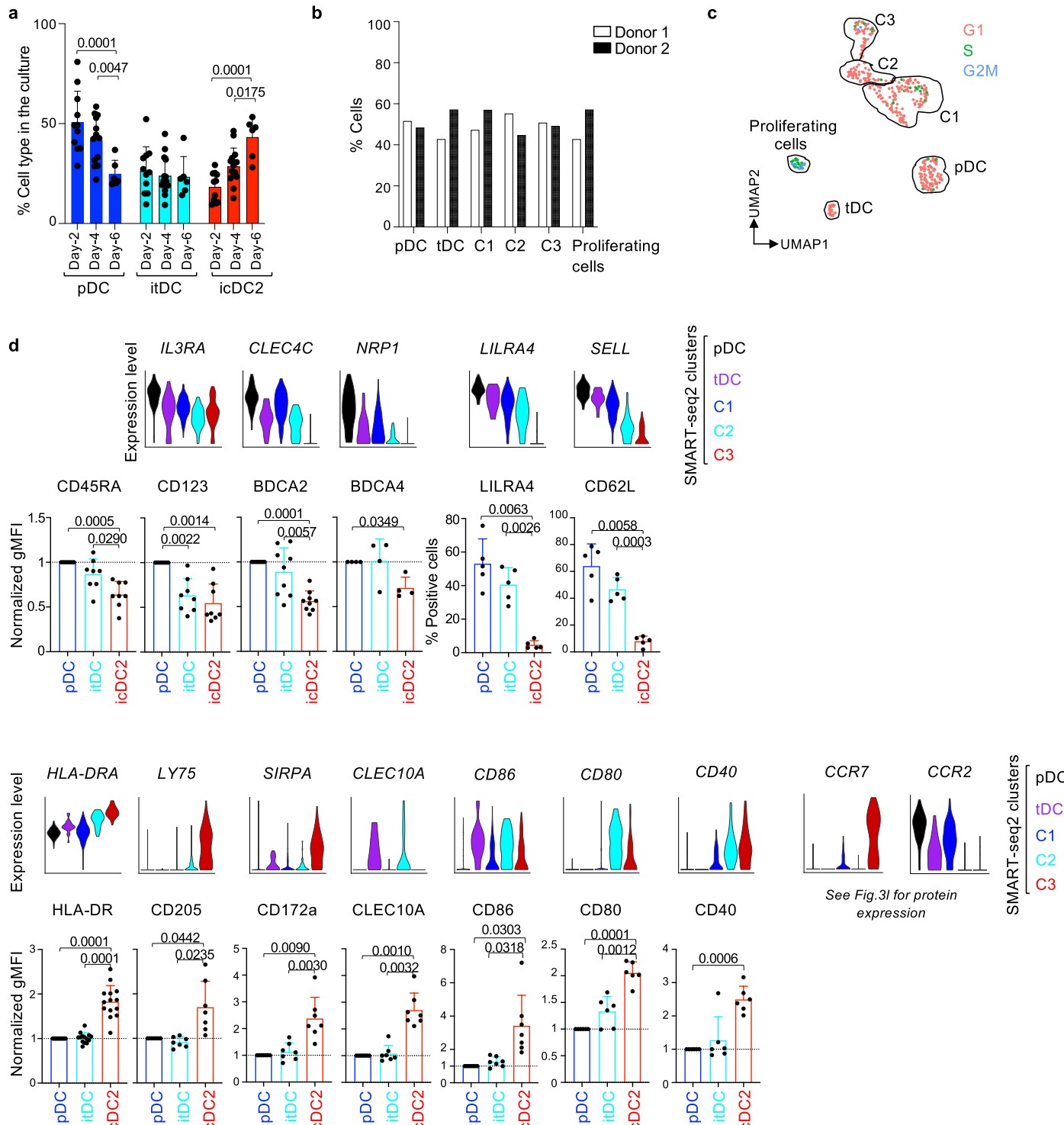

**Extended Data Fig. 3 | Flow cytometry analysis of CD40L stimulated pDCs.**
**a**, Purified pDCs were cultured with CD40L (always in the presence of IL-3), and the percentage of pDCs, itDCs and icDC2s was analyzed at day 2 (n = 11 donors, 11 exp.), day 4 (n = 14 donors, 14 exp.) and day 6 (n = 6 donors, 6 exp.) (mean+SD; unpaired two-way ANOVA with Tukey's multiple comparison test). **b**, Percentage of cells in each SMART-seq2 cluster, colored by blood donor. **c**, UMAP of the SMART-seq2 dataset, colored by cell cycle phase according to Seurat analysis.
**d**, Violin plots of selected gene expression in SMART-seq2. Bar graphs show

protein expression in day-4 culture pDCs, itDCs and icDC2s. Data are shown as percentage of positive cells or gMFI relative to pDCs for the following markers: CD45RA, CD123 (n = 8 donors, 8 exp.); BDCA2 (n = 9 donors, 9 exp.); BDCA4 (n = 4 donors, 4 exp.); HLA-DR (n = 14 donors, 14 exp.); CD205, CD172a, CLEC10A, CD86 (n = 7 donors, 7 exp.); CD80, CD40 (n = 6 donors, 6 exp.); and percentages of LILRA4+ and CD62L+ cells (n = 5 donors, 5 exp.). (mean+SD; paired one-way ANOVA with Tukey's multiple comparisons test).

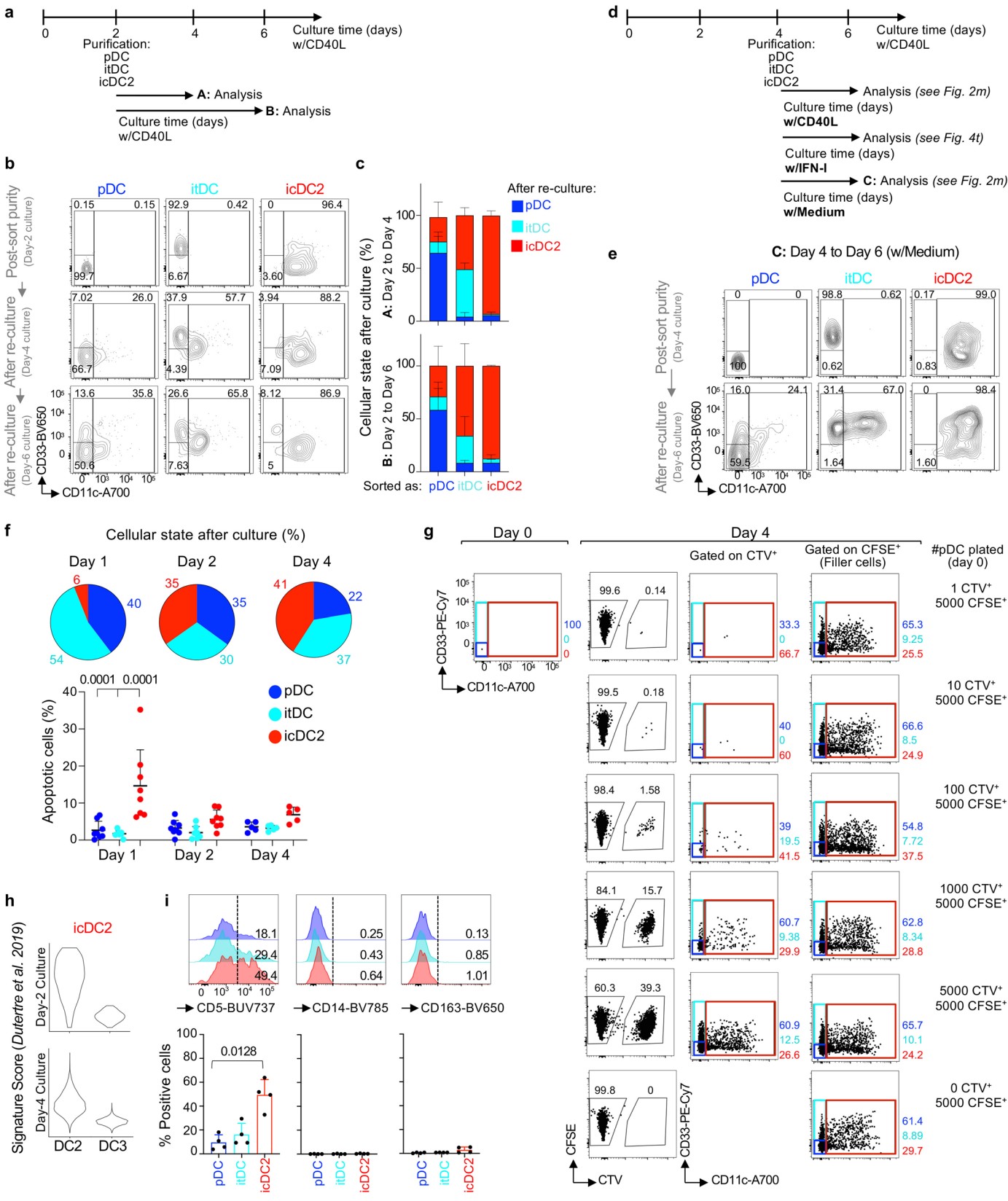

**Extended Data Fig. 4 | See next page for caption.**

**Extended Data Fig. 4 | pDC fate during culture. a-c**, FACS-purified pDCs were stimulated with CD40L for 2 days, and then sorted into pDCs, itDCs and icDC2s and re-cultured with CD40L for an additional 2-4 days. **a**, Experimental set up. **b**, CD33 and CD11c expression at day 2 (top), day 4 (middle) and day 6 (bottom) for one representative donor. **c**, Quantification of (b) at day 4 (n = 4 donors, 4 exp.) and day 6 (n = 3 donors, 3 exp.) (mean+SD). **d**, Experimental set up: FACS-purified pDCs were stimulated with CD40L for 4 days, sorted into pDCs, itDCs and icDC2s, and re-cultured for 2 more days with either CD40L, IFNα, or control media (all conditions contain IL-3; Fig. 2m, Fig. 4t). **e**, CD33 and CD11c expression at day 4 (top) and day 6 (bottom) (1 donor out of 4). **f**, Pie charts showing distribution of pDCs, itDCs and icDC2s (top), and the percentage of apoptotic cells within each population, measured by Apotracker Green staining (bottom; mean±SD; unpaired two-way ANOVA with Tukey's multiple comparison test), following CD40L stimulation. Data were collected by flow cytometry at day 1 (n = 8 donors, 8 exp.), day 2 (n = 8 donors, 8 exp.), and day 4 (n = 5 donors, 5 exp.). **g**, CTV-labelled pDCs were FACS-purified at 1, 10, 100, 1,000 or 5,000 cells/well onto a CFSE-labelled pDCs "filler" bed, and cultured for 4 days with CD40L. Representative flow cytometry plots from 1 of 4 donors (Fig. 2p). **h**, cDC2 (CD163⁻CD5⁺ cells) and DC3 (CD163⁺CD14⁺ cells) gene signature[22] was applied to the icDC2 cluster from day-4 snMultiome-seq (C3 in Fig. 1e) or day-2 SMART-seq 2 (cluster C3 in Fig. 2f) datasets using Seurat AddModuleScore_UCell function. **i**, Flow cytometry expression of cDC2 marker (CD5) and DC3 markers (CD14 and CD163) in day-4 cultures (mean+SD; n = 4 donors, 4 exp.; paired one-way ANOVA with Tukey's multiple comparisons test).

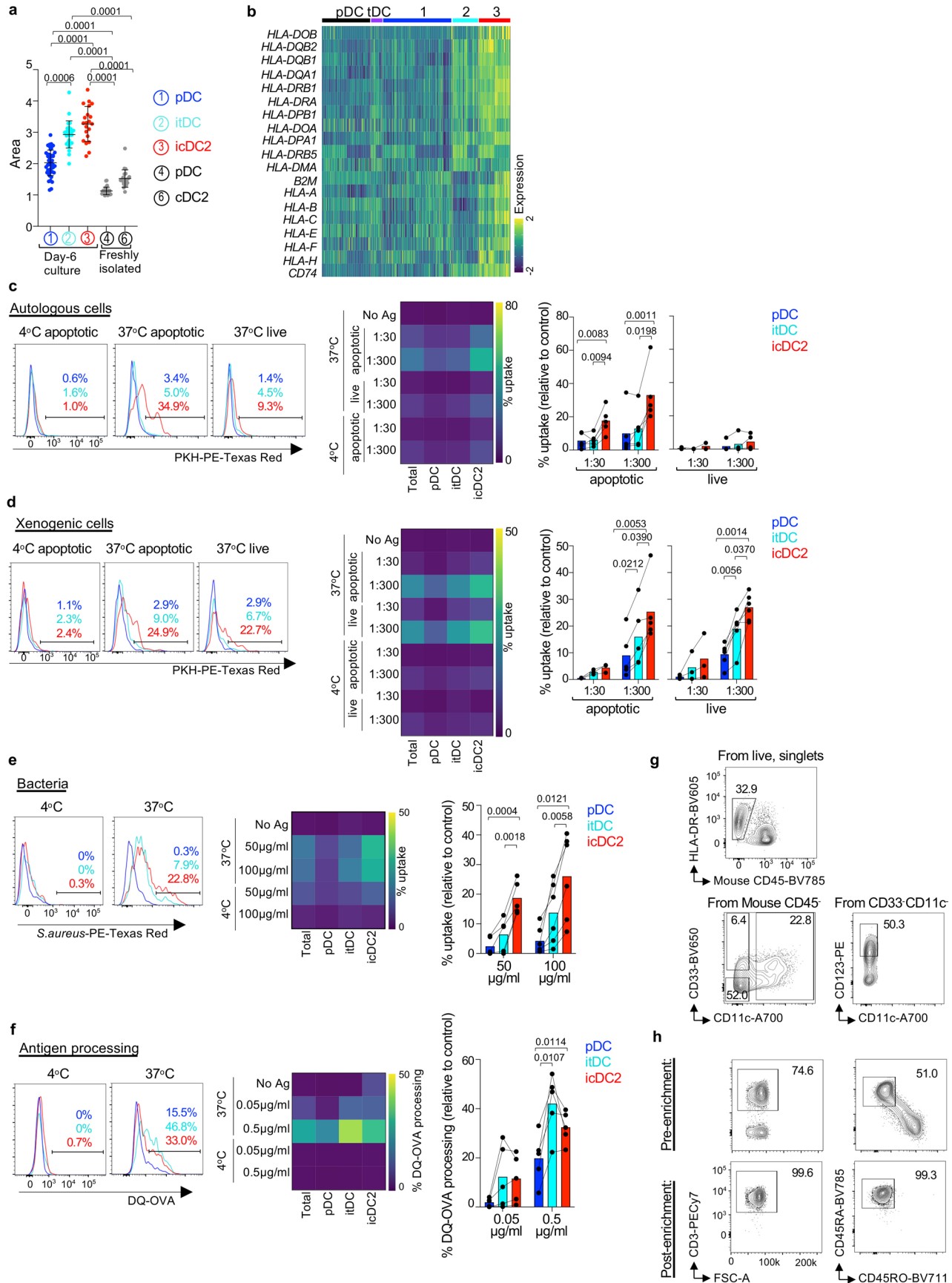

**Extended Data Fig. 5 | See next page for caption.**

**Extended Data Fig. 5 | icDC2 are functional cDCs. a**, Cell area of freshly isolated pDCs (n = 28 cells) and cDC2 (n = 22 cells), and day-6 culture pDCs (n = 43 cells), itDCs (n = 26 cells) and icDC2s (n = 21 cells) from Fig. 3a. Each dot represents a single cell (1 of 2 donors; mean±SD; Kruskal-Wallis test with Dunn's multiple comparisons). **b**, Heatmap of selected genes from the MHCII-related pathways shown in Fig. 3c. **c-e**, Purified pDCs were cultured with CD40L (in the presence of IL-3) for 4 days, then incubated for 3 h at 37 °C with various fluorescently labelled particulate antigens. As negative control, cells were treated with Cytochalasin D and incubated with antigens at 4 °C. Antigen uptake was measured by flow cytometry. Representative histograms (single donor), heatmaps, and bar graphs (relative to 4 °C control) are shown (paired two-way ANOVA with Tukey's multiple comparisons test). **c**, Uptake of PKH-labelled autologous cells, apoptotic apoptotic (1:30 and 1:300, n = 5 donors, 5 exp.) and live (1:30, n = 2, 2 exp.; 1:300, n = 4 donors, 4 exp.). **d**, Uptake of PKH-labelled xenogeneic cells (mouse splenocytes), apoptotic (1:30, n = 3 donors in 3 exp.; 1:300, n = 5 in 5 exp.) and live (1:30, n = 3 donors in 3 exp.; 1:300, n = 6 donors in 6 exp.). **e**, Uptake of pHrodo-labelled *S. aureus*, 50 µg/ml (n = 5 donors, 5 exp.) and or 100 µg/ml (n = 6 donors, 6 exp.). **f**, Cells were incubated with DQ-OVA for 3 h to assess antigen processing (n = 5 donors, 5 exp.). Representative histogram, heatmap (mean), and bar graphs (relative to 4 °C control) are shown (paired two-way ANOVA with Tukey's multiple comparisons test). **g**, Day-4 culture cells were loaded with apoptotic xenogeneic splenocytes for 3-18 hours. pDCs and icDC2s were stained and FACS-purified using the indicated gating strategy. **h**, Naive T cells were enriched to >98%. Representative plots before (top) and after (bottom) enrichment are shown.

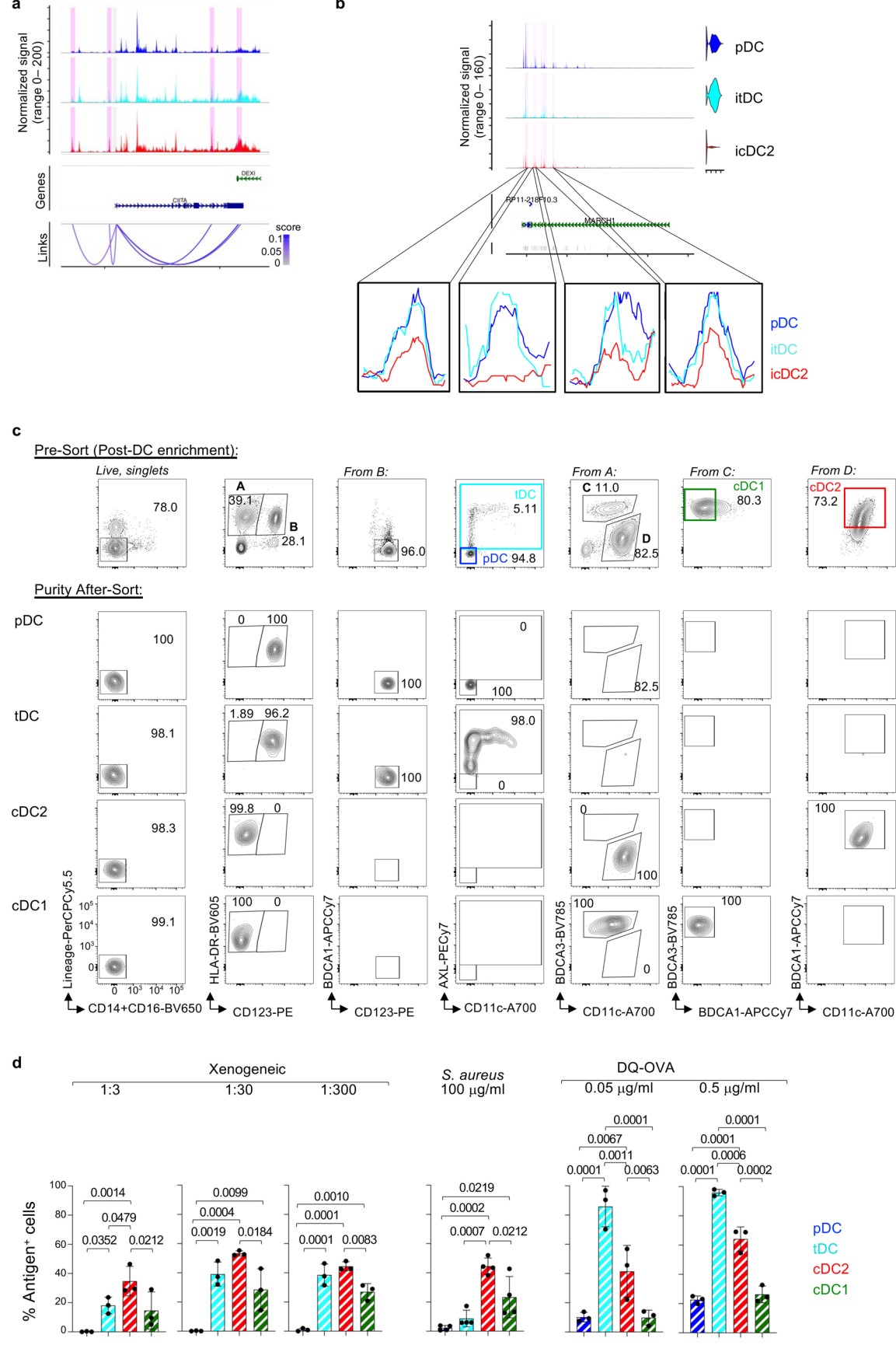

**Extended Data Fig. 6 | See next page for caption.**

**Extended Data Fig. 6 | Epigenetic analysis and antigen presentation capacity of fresh DCs. a**, Epigenetic track showing chromatin accessibility at *CIITA* locus and peak-to-gene links. Promoter 1 is highlighted in grey; putative enhancers are highlighted in purple. **b**, Epigenetic tracks showing chromatin accessibility at the *MARCH1* locus. Putative enhancers overlapping open peaks are highlighted in grey. **c**, Purification strategy for DC subsets. DCs were magnetically enriched from PBMCs, stained, and FACS-purified. Lineage-positive cells (CD3[+], CD19[+], CD20[+], CD335[+], CD66b[+], CD14[+], CD16[+]) were excluded from live singlets. Cells were gated as follows: pDCs (HLA-DR[+]CD123[+]BDCA1[-]AXL[-]CD11c[-]); tDCs (HLA-DR[+]CD123[+]BDCA1[-]AXL[+]); cDC2s (HLA-DR[+]CD123[-]BDCA3[-]CD11c[+]BDCA1[+]); and cDC1s (HLA-DR[+]CD123[-]BDCA3[+]CD11c[+]BDCA1[-]). Representative plots before (top) and after sorting (bottom) are shown (see also Fig. 3k). **d**, Magnetically enriched DCs were incubated with the indicated fluorescently labelled antigens for 3 h at 37°C. Uptake of xenogeneic live cells (n = 3 donors, 3 exp.) and *S. aureus* (100 µg/ml; n = 4 donors, 4 exp.), or processing of DQ-OVA (n = 3 donors, 3 exp.) were measured by flow cytometry and reported as percentage relative to 4°C controls (mean±SD; paired one-way ANOVA with Tukey's multiple comparisons test).

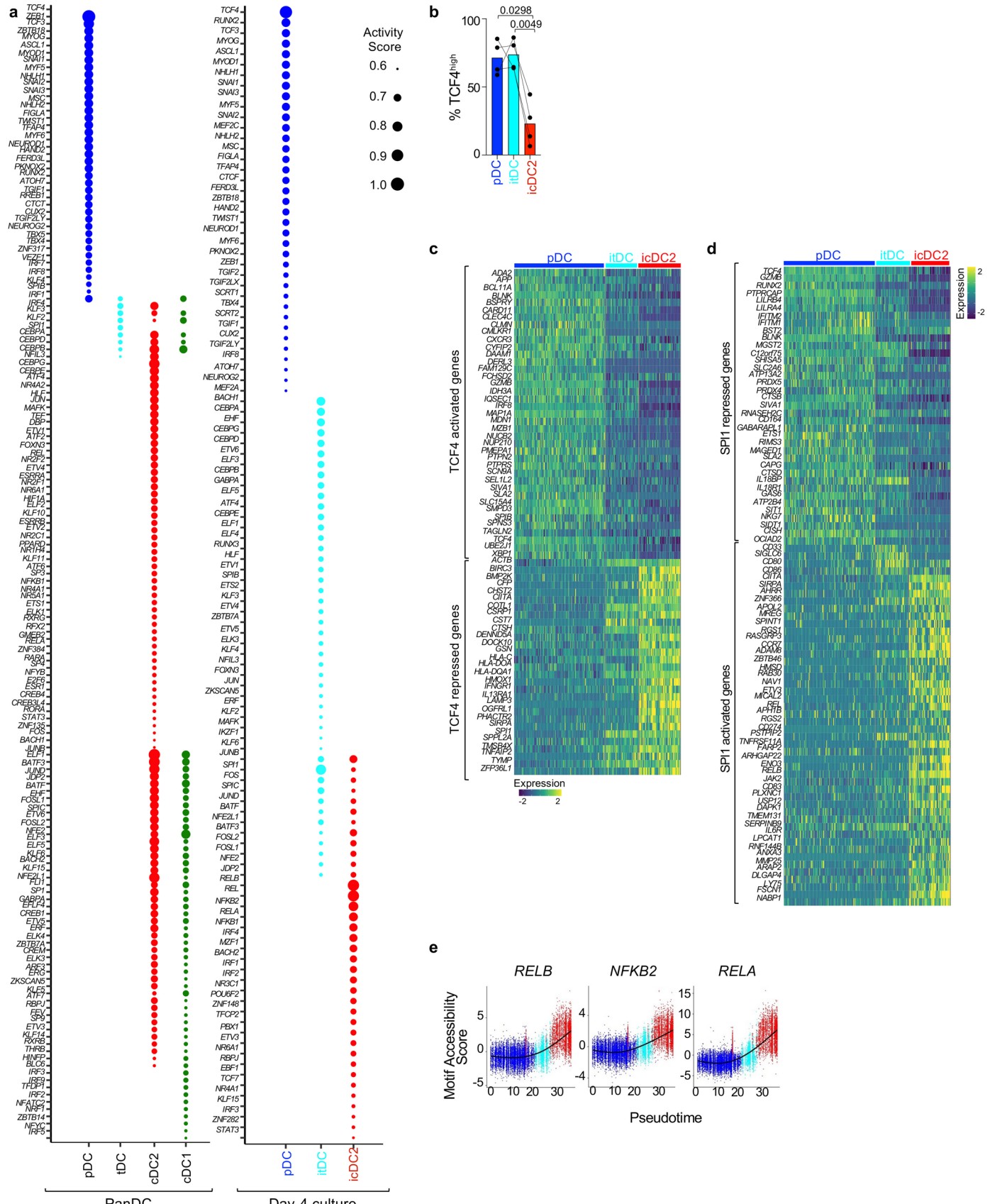

**Extended Data Fig. 7 | Fate decision process is dictated by TNF signaling via NF-κB. a**, TF activity scores (average chromVAR motif accessibility and gene expression) were calculated in the panDC (left) and day-4 culture (right) snMultiome-seq datasets. TFs with an activity score >0.6 are shown. **b**, Percentage TCF4[hi] cells in day-4 cultures (n = 4 donors, 4 exp.;

paired one-way ANOVA with Tukey's multiple comparisons test). **c**, Heatmap of all TCF4-regulated genes analyzed in Fig. 4g. **d**, Heatmap of all SPI1-regulated genes analyzed in Fig. 4i. **e**, Scatter plots showing motif accessibility vs pseudotime for the indicated TFs across clusters C1-C3 in the day-4 culture snMultiome-seq dataset (Fig. 4l).

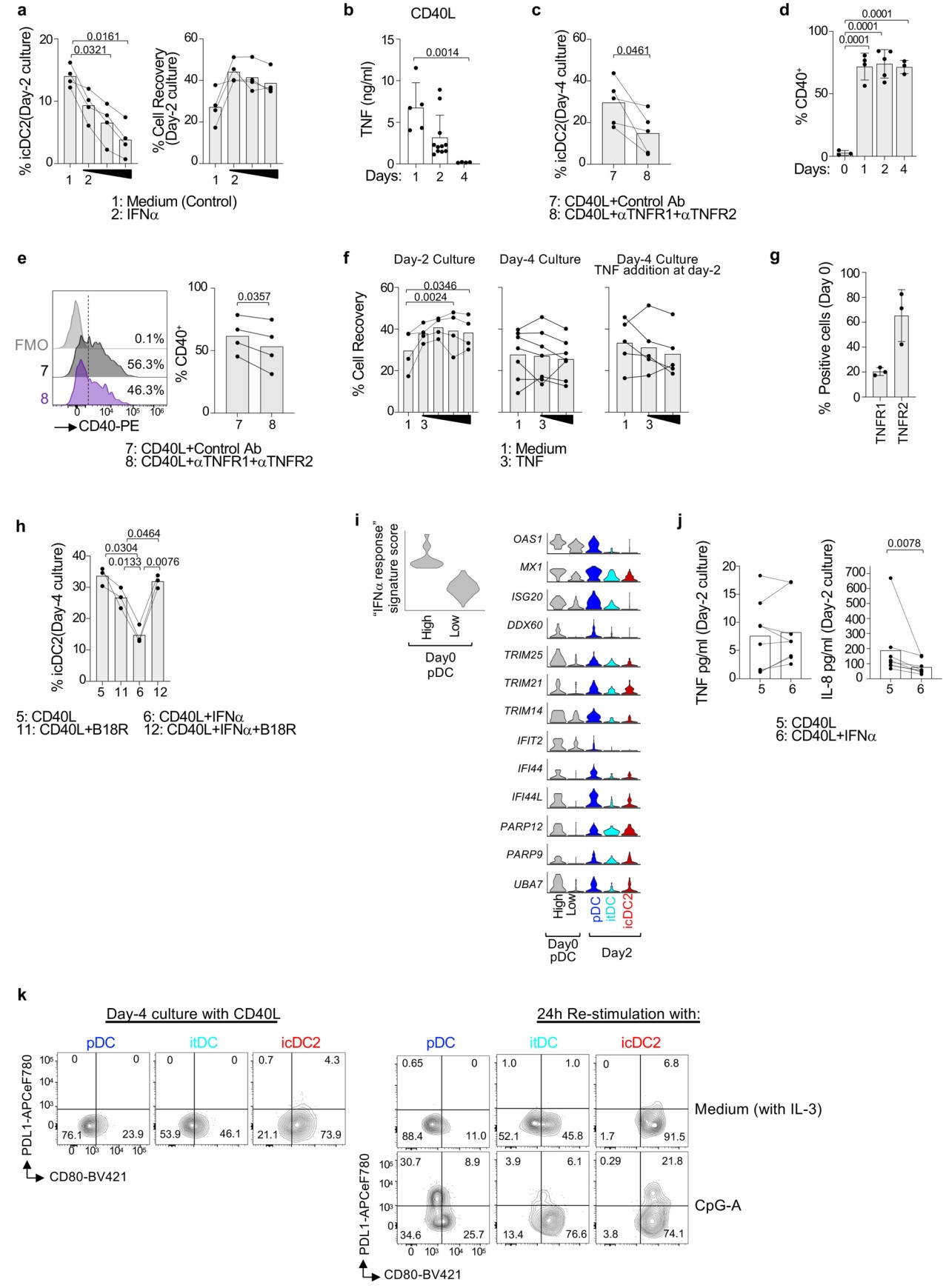

Extended Data Fig. 8 | See next page for caption.

**Extended Data Fig. 8 | IFN-I blocks pDC fate switching. a**, Percentage of icDC2s and cell recovery after 2-days of culture with control media alone (IL-3) or increasing concentration of IFNα (10, 100, 1000 U/ml) (n = 4 donors, 4 exp.; paired one-way ANOVA with Dunnett's multiple comparisons test). **b**, TNF secretion in cultures stimulated with CD40L, measured in supernatant by cytometric bead array (CBA) at day 1 (n = 5 donors, 5 exp.), day 2 (n = 11 donors, 11 exp.) and day 4 (n = 4 donors, 4 exp.) (mean+SD; two-sided Kruskal-Wallis test with Dunn's multiple comparisons test). **c**, Percentage of icDC2s in day-4 cultures with CD40L and either TNFR1/2 blocking antibodies or isotype controls (all conditions contained IL-3) (n = 5 donors, 5 exp.; two-sided paired t test). **d**. Percentage of CD40+ cells before (day 0; n = 3 donors, 3 exp.) or after 1 day (n = 4 donors, 4 exp.), 2 days (n = 5 donors, 5 exp.), and 4 days (n = 3 donors, 3 exp.) culture with CD40L (mean±SD; unpaired one-way ANOVA with Tukey's multiple comparisons test). **e**, Percentage of CD40+ cells after 4 days culture with CD40L and either TNFR1/2 blocking antibodies or isotype controls (all conditions contained IL-3). Representative flow cytometry plots and quantification shown (n = 4 donors, 4 exp.; two-sided paired t test). **f**, Cell recovery after pDC culture

with increasing TNF concentrations (2, 20, 200, 2000 ng/ml) (all with IL-3), measured at day 2 (left; n = 4 donors, 4 exp.), day-4 (middle; n = 7 donors, 7 exp.), or day-4 with a second TNF dose (200, 2000 ng/ml) added on day 2 (right; n = 5 donors, 5 exp.) (paired one-way ANOVA with Tukey or Dunnett's multiple comparison test; see Fig. 4q-r). **g**, Percentage of TNFR1+ and TNFR2+ cells in freshly isolated pDCs (n = 3 donors, 3 exp.; mean±SD). **h**, Percentage of icDC2 in day-4 cultures under indicated conditions (all with IL-3) (n = 3 donors, 3 exp.; paired one-way ANOVA with Tukey's multiple comparisons test). **i**, The fresh pDC cluster in the SMART-seq2 dataset (pDC cluster in Fig. 2f) was split onto 'High' and 'Low' groups based on IFNα response signature (MSigDB). Violin plots show expression of selected IFN-stimulated genes. **j**, TNF and IL-8 secretion in day-2 cultures, measured in supernatant by CBA (n = 8 donors, 8 exp.; two-sided Wilcoxon matched-pairs signed rank test or paired t test). **k**, Flow cytometry plots of PDL1 and CD80 expression on pDCs, itDCs and icDC2s from day-4 cultures (CD40L) (left), and after 1-day re-stimulation with medium (IL-3) or CpG-A (1 of 4 donors in 4 exp.).

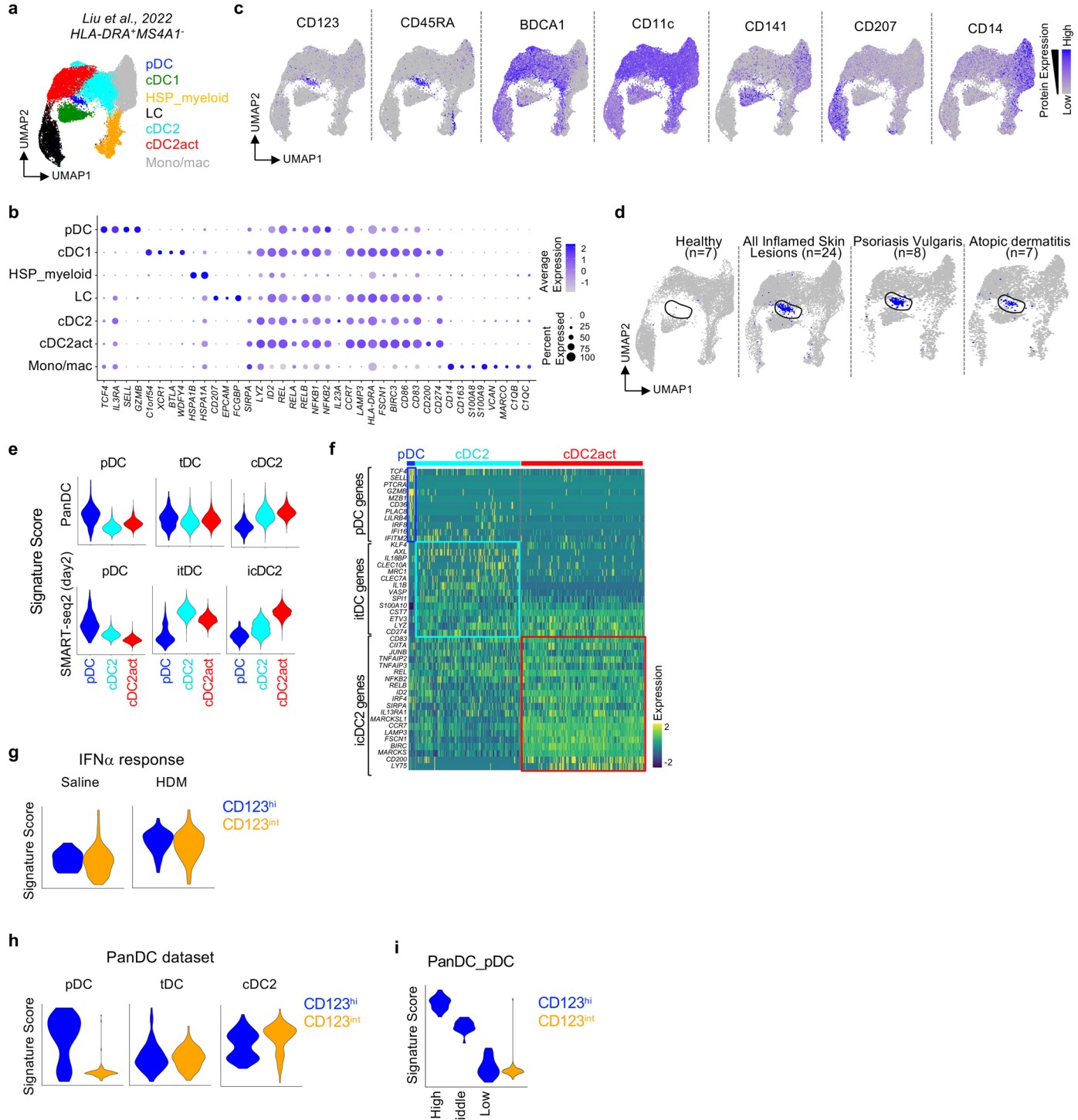

**Extended Data Fig. 9 | pDC fate-switching during skin inflammation. a-f,** CITE-seq analysis of healthy and inflamed human skin samples from a public dataset[43]. **a,** RNA UMAP of myeloid cell clusters (subsetted based on positive *HLA-DR*+ and *MS4A1*− gene expression). 'HSP-myeloid' denotes a myeloid cell cluster with high heat-shock protein expression, 'LC' denotes Langerhans cells. 'cDC2act' denotes an activated cDC2 cluster. **b-c,** Expression of key critical transcriptional (**b**) or protein (**c**) markers distinguishing cellular clusters in (**a**). **d,** RNA UMAPs from (**a**), showing cellular distribution in: healthy controls, all inflamed lesions (atopic dermatitis, psoriasis vulgaris, bullous pemphigoid, lichen planus and patients with clinicopathologically indeterminate rash), atopic dermatitis alone, and psoriasis vulgaris alone. Numbers of donors is indicated in brackets.

The pDC cluster is highlighted in blue. **e,** Gene signature score of DC subsets from the panDC snMultiome (top) and SMART-seq2 (bottom) datasets projected onto pDC, cDC2 and cDC2act clusters from (**a**). **f,** Heatmap of selected genes shown in (**e**). **g,** pDC conversion was analyzed in human skin blisters challenged with saline or house dust mite (HDM)[44]. Shown is the "IFNα response" signature MSigDB score in CD123hi pDCs (blue) and CD123int DCs (orange) from saline and HDM blisters (Fig. 5a-d). **h,** Gene signature score of DC subsets from the panDC snMultiome dataset projected onto the saline-treated blister dataset. **i,** Within the HDM-treated blister dataset, CD123hi cells were stratified onto 'High', 'Middle' and 'Low' groups based on panDC-pDC signature score.

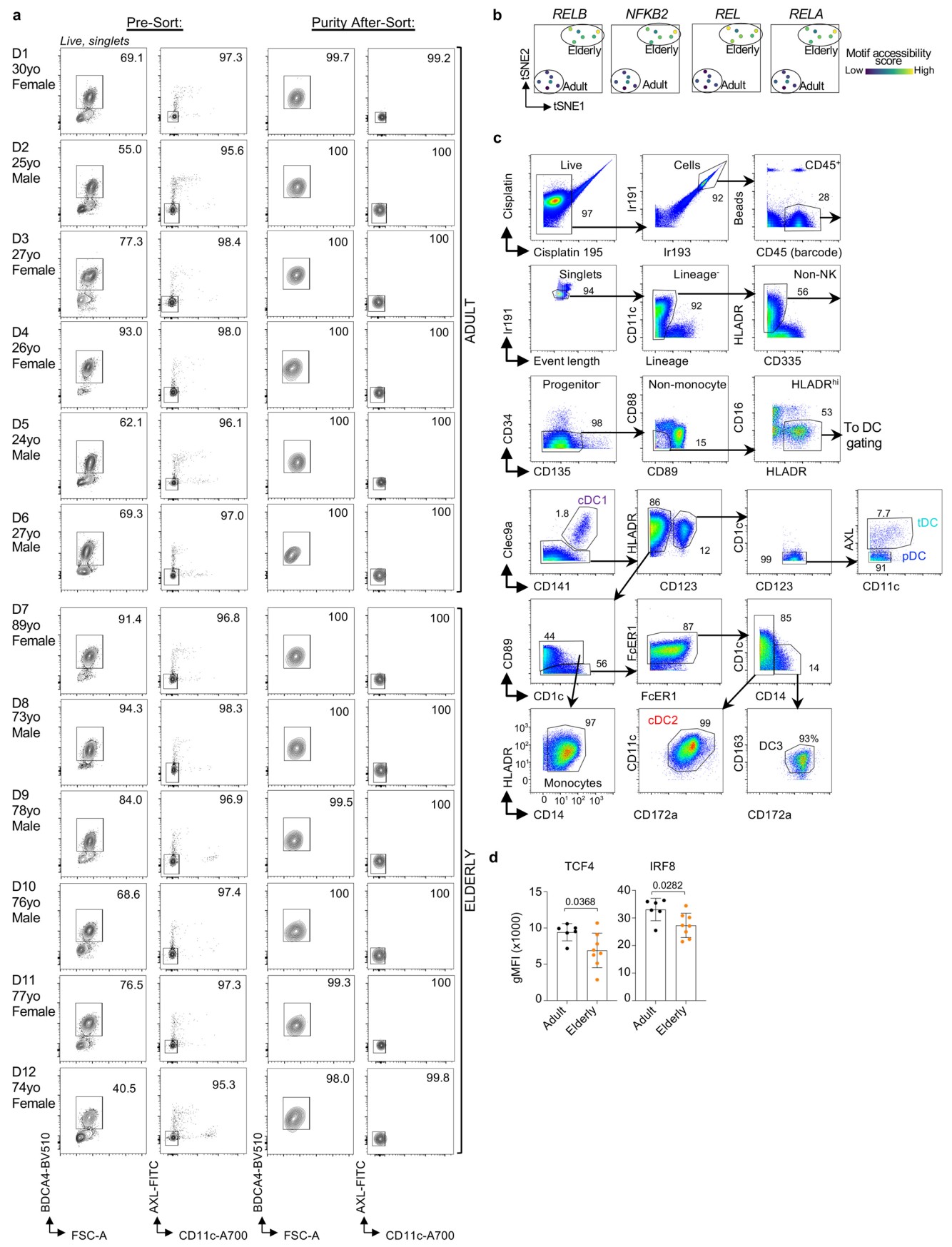

**Extended Data Fig. 10 | See next page for caption.**

**Extended Data Fig. 10 | pDC gating strategy in adult and elderly donors.**
**a**, pDCs were magnetically enriched from PBMCs of adult and elderly donors, stained, and purified by FACS. Cells were gated as BDCA4$^+$AXL$^-$CD11c$^-$. Representative flow cytometry plots are shown before (left panels) and after sorting (right panels). **b**, Motif accessibility scores of selected TFs computed from bulk ATAC-seq of pDCs from adult and elderly donors (Fig. 6c). **c**, CyTOF gating strategy used for the experiments shown in Fig. 6d-h). **d**, gMFI of TCF4 and IRF8 in adult (n = 6 donors) and elderly (n = 8 donors) donors. PBMCs were cryopreserved, and cells from all donors in this cohort were stained and analyzed together in a single flow cytometry run (mean±SD; two-sided unpaired t test; one of two independent cohorts—see also Fig. 6i for the second cohort).

# Reporting Summary

## Statistics

For all statistical analyses, confirm that the following items are present in the figure legend, table legend, main text, or Methods section.

| n/a | Confirmed | |
|---|---|---|
| ☐ | ☒ | The exact sample size (*n*) for each experimental group/condition, given as a discrete number and unit of measurement |
| ☐ | ☒ | A statement on whether measurements were taken from distinct samples or whether the same sample was measured repeatedly |
| ☐ | ☒ | The statistical test(s) used AND whether they are one- or two-sided<br>*Only common tests should be described solely by name; describe more complex techniques in the Methods section.* |
| ☒ | ☐ | A description of all covariates tested |
| ☐ | ☒ | A description of any assumptions or corrections, such as tests of normality and adjustment for multiple comparisons |
| ☐ | ☒ | A full description of the statistical parameters including central tendency (e.g. means) or other basic estimates (e.g. regression coefficient) AND variation (e.g. standard deviation) or associated estimates of uncertainty (e.g. confidence intervals) |
| ☐ | ☒ | For null hypothesis testing, the test statistic (e.g. *F*, *t*, *r*) with confidence intervals, effect sizes, degrees of freedom and *P* value noted<br>*Give P values as exact values whenever suitable.* |
| ☒ | ☐ | For Bayesian analysis, information on the choice of priors and Markov chain Monte Carlo settings |
| ☒ | ☐ | For hierarchical and complex designs, identification of the appropriate level for tests and full reporting of outcomes |
| ☐ | ☒ | Estimates of effect sizes (e.g. Cohen's *d*, Pearson's *r*), indicating how they were calculated |

*Our web collection on statistics for biologists contains articles on many of the points above.*

## Software and code

Policy information about availability of computer code

| | |
|---|---|
| Data collection | Flow cytometry data was acquired using BD FACS Diva software (v8.01) or CYTEK SpectroFlo (v3.3.0). RNA Sequencing data were obtained using Illumina novaseq6000 platform. |
| Data analysis | Flow cytometry data was analyzed using FlowJo software (v.10.10.0). Sequencing data was analyzed using the following programs: Seurat (v.5.0.2), Signac (v.1.12.9004), R (v.4.2.3), GSEA (v.4.2.3), chromVAR (v.1.22.1), Monocle3 (v.2.28.0), Slingshot (v.2.8.0), Harmony (v.1.2.0), viridis (v.0.6.5), ggplot2 (v.3.5.0), scCustomize (v.2.1.2), Triwise (v.0.99.5), MACS2 (v.2.2.9.1), DESEq2 (v.3.14), Pheatmap (v. 1. 0.12). Bulk RNAseq data was aligned using STAR (2.7.3a), and bulk ATACseq data was processed using (nfatachttps://nf-co.re/atacseq/2.1.2). CyTOF data was analyzed using FlowJo (v10.10.1), R (v.4.3.1), Scaffold (v0.1) (https://github.com/nolanlab/scaffold.git), Premessa (v.0.3.4), Marker Enrichment Modeling (v.0.1.1) and MATLAB (v2023b). Graphing and Statistical analysis were performed on GraphPad Prism 10 (GraphPad Software, Inc.). |

For manuscripts utilizing custom algorithms or software that are central to the research but not yet described in published literature, software must be made available to editors and reviewers. We strongly encourage code deposition in a community repository (e.g. GitHub). See the Nature Portfolio guidelines for submitting code & software for further information.

# Data

Policy information about availability of data

All manuscripts must include a data availability statement. This statement should provide the following information, where applicable:
- Accession codes, unique identifiers, or web links for publicly available datasets
- A description of any restrictions on data availability
- For clinical datasets or third party data, please ensure that the statement adheres to our policy

Data generated in this study have been deposited in NCBI GEO under accession numbers: GSE267100 (GRCh38 reference genome); GSE267099 (GRCh38 reference genome); GSE267174 (GRCh38 reference genome); GSE266889 (hg19 reference genome), GSE269411 (hg19 reference genome); and GSE279911 (hg19 reference genome). Publicly available CITE-seq (EGAS00001005271) was download from (https://explore.data.humancellatlas.org/projects/5bd01deb-01ee-4611-8efd-cf0ec5f56ac4/project-matrices). Publicly available SMART-seq2 (E-MTAB-8498) dataset was downloaded from (https://explore.data.humancellatlas.org/projects/67a3de09-45b9-49c3-a068-ff4665daa50e/project-metadata). Source data are provided with this paper.

# Research involving human participants, their data, or biological material

Policy information about studies with human participants or human data. See also policy information about sex, gender (identity/presentation), and sexual orientation and race, ethnicity and racism.

| | |
|---|---|
| Reporting on sex and gender | When provided, information on donor age and sex is reported in the Methods (see below) or in the supporting data (Supplementary Table 6). Samples were not selected based on sex, age (beyond the range described above), race, or other individual characteristics. |
| Reporting on race, ethnicity, or other socially relevant groupings | This manuscript did not consider race and ethnicity during data analyses. |
| Population characteristics | Healthy donors (buffy coats and blood drawn) between 20-40 and over 65 years old, without signs of infection for at least 2 weeks. |
| Recruitment | De-identified blood (collected using EDTA-coated tubes; BD Biosciences) and buffy coats from healthy adults (20–40 years) and elderly donors (>65 years; Supplementary Table 6) were obtained through local lab-led blood donation efforts following Stanford University and UCSD guidelines, or from the Stanford Blood Center. Donors provided informed consent under protocols approved by the Institutional Review Boards (IRBs) of Stanford University and UC San Diego, and did not receive compensation. |
| Ethics oversight | De-identified blood and buffy coats were obtained following Stanford University and UCSD guidelines. Donors provided informed consent under protocols approved by the Institutional Review Boards (IRBs) of Stanford University and UC San Diego. |

Note that full information on the approval of the study protocol must also be provided in the manuscript.

# Field-specific reporting

Please select the one below that is the best fit for your research. If you are not sure, read the appropriate sections before making your selection.

☒ Life sciences   ☐ Behavioural & social sciences   ☐ Ecological, evolutionary & environmental sciences

For a reference copy of the document with all sections, see nature.com/documents/nr-reporting-summary-flat.pdf

# Life sciences study design

All studies must disclose on these points even when the disclosure is negative.

| | |
|---|---|
| Sample size | No sample-size calculations were performed, but sample sizes are similar to those reported in previous calculations (Leylek, et al. Cell Reports 2019; 2020; Alculumbre et al., Nat. Immunol 2018; Palucka et al., PNAS, 2005; Alcantara-Hernandez et al., Immunity 2017). |
| Data exclusions | No data were excluded. For sequencing data, pre-established criteria for single-cell exclusion, i.e., low number of unique genes, abnormally high read count, and high mitochondrial gene content, was used. |
| Replication | All assays were repeated at least twice with multiple blood donors. All attempts of replication were successful. |
| Randomization | Blood samples were randomly allocated across experiments. |
| Blinding | Investigators were not blinded because no assay where it would be necessary was performed. We performed quantitative measurements and unbiased computational analyzes. |

# Reporting for specific materials, systems and methods

We require information from authors about some types of materials, experimental systems and methods used in many studies. Here, indicate whether each material, system or method listed is relevant to your study. If you are not sure if a list item applies to your research, read the appropriate section before selecting a response.

## Materials & experimental systems

| n/a | Involved in the study |
|---|---|
| ☐ | ☒ Antibodies |
| ☒ | ☐ Eukaryotic cell lines |
| ☒ | ☐ Palaeontology and archaeology |
| ☒ | ☐ Animals and other organisms |
| ☒ | ☐ Clinical data |
| ☒ | ☐ Dual use research of concern |
| ☒ | ☐ Plants |

## Methods

| n/a | Involved in the study |
|---|---|
| ☒ | ☐ ChIP-seq |
| ☐ | ☒ Flow cytometry |
| ☒ | ☐ MRI-based neuroimaging |

## Antibodies

Antibodies used

Species reactivity Target Color Clone Catolog number Company Identifier Dilution
Anti-human AXL (clone 108724) AF488 R&D Systems Cat# FAB154G; RRID: AB_2714170; 1:50
Anti-human AXL (clone DS7HAXL) PE-Cy7 Thermo Fisher Cat# 25-1087-42; RRID:AB_2723959; 1:100
Anti-human BDCA1 (clone L161) APC-Cy7 Biolegend Cat# 331520; RRID:AB_10644008; 1:100
Anti-human BDCA2 (clone 201A) FITC Biolegend Cat# 354208; RRID:AB_2561364; 1:50
Anti-human BDCA3 (clone M80) BV785 Biolegend Cat# 344116; RRID:AB_2572194; 1:100
Anti-human BDCA4 (clone 12C2) PerCPCy5.5 Biolegend Cat# 354510; RRID:AB_2561558; 1:100
Anti-human BDCA4 (clone 12C2) APC Biolegend Cat# 354506; RRID:AB_11219600; 1:400
Anti-human BDCA4 (clone 12C2) BV510 Biolegend Cat# 354515; RRID:AB_25630741:200
Anti-human BDCA4 (clone M80) BV785 Biolegend Cat# 344116; RRID:AB_2572194; 1:100
Anti-human CCR2 (clone K036C2) APC Biolegend Cat# 357207; RRID:AB_2562239; 1:50
Anti-human CCR7 (clone G043H7) PE-Cy7 Biolegend Cat# 353226; RRID:AB_11125576; 1:50
Anti-human CD11c (clone Bu15) Alexa700 Biolegend Cat# 337220; RRID:AB_2561502; 1:200
Anti-human CD123 (clone 7G3) BUV395 BD Biosciences Cat# 564195; RRID:AB_2714171; 1:100
Anti-human CD123 (clone 6H6) FITC Biolegend Cat# 306014; RRID:AB_2124259; 1:200
Anti-human CD123 (clone 6H6) PE Biolegend Cat# 306006; RRID:AB_314580; 1:400
Anti-human CD123 (clone 6H6) PE-Cy7 Biolegend Cat# 306009; RRID:AB_493576; 1:200
Anti-human CD123 (clone 6H6) PE/Dazzle 594 Biolegend Cat# 306034; RRID:AB_2566450; 1:200
Anti-human CD14 (clone M5E2) BV650 Biolegend Cat# 301836; RRID:AB_11204241; 1:100
Anti-human CD14 (clone M5E2) BV785 Biolegend Cat# 301840; RRID:AB_2563425; 1:200
Anti-human CD16 (clone 3G8) BV650 Biolegend Cat# 302042; RRID:AB_11125578; 1:200
Anti-human CD19 (clone HIB19) PerCP Cy5.5 Biolegend Cat# 302230; RRID:AB_2275547; 1:200
Anti-human CD19 (clone HIB19) PB Biolegend Cat# 302232; RRID: AB_2073118; 1:50
Anti-human CD20 (clone 2H7) PerCP Cy5.5 Biolegend Cat# 302326; RRID:AB_893285; 1:200
Anti-human CD20 (clone 2H7) PB Biolegend Cat# 302328; RRID AB_1595435; 1:200
Anti-human CD25 (clone BC96) PE Biolgend Cat# 302606; RRID:AB_314275; 1:200
Anti-human CD3 (clone UCTH1) PECy7 Biolegend Cat# 300419; RRID:AB_439781; 1:200
Anti-human CD3 (clone UCTH1) PerCP Cy5.5 Biolegend Cat# 300430; RRID:AB_893299; 1:200
Anti-human CD3 (clone UCTH1) PB Biolegend Cat # 300431; RRID: AB_1595437; 1:100
Anti-human CD33 (clone WM53) BV650 Biolegend Cat# 303430; RRID:AB_2650933; 1:25
Anti-human CD33 (clone WM53) PE Biolegend Cat# 303403; RRID:AB_314348; 1:100
Anti-human CD33 (clone WM53) PE-Cy7 Biolegend Cat# 303433; RRID:AB_2734264; 1:100
Anti-human CD335 (NKp46) (clone 9 E2) PerCP Cy5.5 Biolegend Cat# 331920; RRID:AB_2561665; 1:50
Anti-human CD335 (NKp46) (clone 9 E2) PB Biolegend Cat# 331912; RRID: AB_2149280; 1:200
Anti-human CD40 (clone 5C3) BV421 Biolegend Cat# 334331; RRID:AB_AB_2564210; 1:50
Anti-human CD40 (clone 5C3) PE Biolegend Cat# 334308; RRID:AB_1186038; 1:50
Anti-human CD45 (clone HI30) PE Biolegend Cat# 304008; RRID:AB_2564156; 1:50
Anti-human CD45RA (clone HI100) BV 605 Biolegend Cat# 304134; RRID:AB_2563814; 1:400
Anti-human CD45RA (clone HI100) BV785 Biolegend Cat# 304139; RRID:AB_2563816; 1:100
Anti-human CD45RO (clone UCHL1) BV711 Biolegend Cat# 304235; RRID:AB_2562107; 1:25
Anti-human CD62L (clone DREG-56 ) APCCy7 biolegend Cat# 304813; RRID:AB_493583; 1:100
Anti-human CD66B (clone G10F5) PerCP Cy5.5 Biolegend Cat# 305108; RRID:AB_2077856; 1:50
Anti-human CD66B (clone G10F5) PB Biolegend Cat# 305112; RRID: AB_2563294; 1:200
Anti-human CD8 (clone RPA-T8) APCCy7 Biolegend Cat# 301016; RRID:AB_314133; 1:200
Anti-human CD80 (clone 2D10) BV421 Biolegend Cat# 305222; RRID:AB_2564407; 1:50
Anti-human LILRA4 (CD85g) (clone 17G10.2) Alexa647 biolegend Cat# 326410; RRID:AB_2265747; 1:25
Anti-human CD86 (clone 2331, FUN-1) BUV737 BD Biosciences Cat# 564428; RRID:AB_2738804; 1:50
Anti-human CLEC10A (clone H037G3) APC Biolegend Cat# 354705; RRID:AB_11218803; 1:50
Anti-human CELC10A (clone H037G3) PE Biolegend Cat# 354703; RRID:AB_11219202; 1:50
Anti-human DEC205 (clone 3G9) Alexa647 Obtained from Celldex Therapeutics, and labeled in-house; 1:2000

Anti-human HLA-DR (clone L243) APC Biolegend Cat# 307610; RRID:AB_314687; 1:100
Anti-human HLA-DR (clone L243) BV605 Biolegend Cat# 307640; RRID:AB_11219187; 1:100
Anti-human IFNAR1 (clone 85228) PE Thermo Fisher Cat# MA5-23630; RRID:AB_2609327; 1:50
Anti-human IFNAR2 (clone 122) FITC Thermo Fisher Cat# MA5-40953; RRID:AB_2898714; 1:50
Anti-human IFNAR2 (clone 122) APC Thermo Fisher Cat# MA5-40952; RRID:AB_2898713; 1:50
Anti-human IRF8 (clone V3GYWCH) PerCP-eFluor710 Thermo Fisher Cat#46-9852-82; RRID:AB_2573904; 1:400
Anti-human Ki67 (clone Ki-67) PerCP Cy5.5 Biolegend Cat# 350520; RRID:AB_2562295; 1:50
Anti-human SIRPa (CD172a) (clone SE5A5) PerCP Cy5.5 Biolegend Cat# 323811; RRID:AB_11219000; 1:100
Anti-human/mouse TCF4 (clone NCI-R159-6) Alexa647; Abcam Cat# ab246763; RRID:AB_2714172; 1:2000
Anti-human TNFR2 (clone 3G7A02) PE-Cy7 Biolegend Cat # 358411; RRID: AB_2564396; 1:50
Anti-human TNFR1 (clone W15099A) APC Biolegend Cat # 369905; RRID: AB_2650764 ; 1:50
Anti-mouse CD45 (clone 30-F11) BV785 Biolegend Cat# 103149; RRID:AB_2564590; 1:400
Anti-mouse CD45 (clone 30-F11) FITC Biolegend Cat# 103108; RRID:AB_312972; 1:200
Anti-mouse CD45 (clone 30-F11) PE Biolegend Cat# 103106; RRID:AB_312971; 1:800

| Validation | All antibodies used were commercially available, well-validated clones routinely quality-controlled by their respective manufacturers. Prior to experimental use, each antibody was titrated to determine the optimal staining concentration under our specific assay conditions, including the different flow cytometry platforms used. Validation in our hands included confirming expected positive and negative populations based on well-characterized expression patterns. When used for the first time, fluorescence-minus-one (FMO) and isotype controls were included to define gating boundaries. Technical specifications and validation details for each antibody clone are available on the vendors' websites by referencing the catalog numbers listed above. |
|---|---|

# Plants

| Seed stocks | *Report on the source of all seed stocks or other plant material used. If applicable, state the seed stock centre and catalogue number. If plant specimens were collected from the field, describe the collection location, date and sampling procedures.* |
|---|---|
| Novel plant genotypes | *Describe the methods by which all novel plant genotypes were produced. This includes those generated by transgenic approaches, gene editing, chemical/radiation-based mutagenesis and hybridization. For transgenic lines, describe the transformation method, the number of independent lines analyzed and the generation upon which experiments were performed. For gene-edited lines, describe the editor used, the endogenous sequence targeted for editing, the targeting guide RNA sequence (if applicable) and how the editor was applied.* |
| Authentication | *Describe any authentication procedures for each seed stock used or novel genotype generated. Describe any experiments used to assess the effect of a mutation and, where applicable, how potential secondary effects (e.g. second site T-DNA insertions, mosiacism, off-target gene editing) were examined.* |

# Flow Cytometry

## Plots

Confirm that:

☒ The axis labels state the marker and fluorochrome used (e.g. CD4-FITC).

☒ The axis scales are clearly visible. Include numbers along axes only for bottom left plot of group (a 'group' is an analysis of identical markers).

☒ All plots are contour plots with outliers or pseudocolor plots.

☒ A numerical value for number of cells or percentage (with statistics) is provided.

## Methodology

| Sample preparation | Peripheral blood mononuclear cells (PBMCs) were isolated by density gradient centrifugation from buffy coats or blood of healthy donors using Ficoll-Paque PLUS (GE Healthcare), following manufacture instructions. Dendritic cells were isolated from fresh PBMCs by negative magnetic-bead enrichment followed by FACS-sorting. To enrich all mononuclear phagocytes, PBMCs were treated with human gamma-globulin (Invitrogen) for 15 minutes on ice to block non-specific binding and incubated with antibodies against CD3, CD19, CD335 and CD66b, followed by anti-mouse magnetic Dynabeads (ThermoFisher). Alternatively, PanDCs and pDCs were enriched using the EasySep Human PanDC pre-enrichment kit or EasySep Human Plasmacytoid DC Isolation Kit (STEMCELL), following manufacture instructions. For FACS purification, enriched cells were stained with antibody cocktail for 30 minutes on ice and sorted using FACSAria II or FACSAria Fuision (BD Biosciences). For naïve T cell preparation, frozen PBMCs were thawed, washed twice with PBS, incubated with 1.7 nM CFSE (Sigma-Aldrich) or 2.5 uM CellTrace Violet (ThermoFisher) at 37C in a water bath for 10 min, following by washing with R10 complete media. CD3+CD45RA+CD45RO- naïve T cells were isolated to >98% using EasySep Human Naïve Pan T cell isolation Kit (STEMCELL) according to manufacturer's instructions. |
|---|---|
| | For in vitro culture, 10,000 sorted pDCs were cultured in 96-well U-bottom plates in 200ul R10 complete media [RPMI (Corning) with 10% heat-inactivated FBS (GIBCO), 2mM L-glutamine, 100IU/mL Penicillin, 100ug/mL Streptomycin, 10mM HEPES, 1mM Sodium Pyruvate, 1X MEM Nonessential Amino Acids (all Corning), and 55uM 2-Mercaptoethanol (GIBCO)] at 37C. All cultures contained 10ng/mL recombinant human IL-3 (R&D Systems; carrier-free) for pDC survival. Activation stimuli included 100-200ng/mL CD40L (R&D Systems; carrier-free), 2-2000ng/mL TNF (Biolegend; carrier-free), 10-1000U/mL IFNa (PBL Assay Science), or 5ug/mL CpG-A (ODN 2216, Invivogen). For IFN-I blockade, 1000ng/mL B18R (R&D) was added. For TNF blockade, pDCs were pre-incubated for 1 h with 10ug/mL anti-TNFR1 (clone 16805, R&D), anti-TNFR2 (clone 22210, R&D), or isotype control before CD40L stimulation. Secreted TNF and IL-8 were measured in day-1, 2 or 4 supernatants by Cytometric |

Bead Array (CBA) Human Enhanced Sensitivity kit (BD Biosciences). For IFN-I detection, sorted DCs were cultured with 150uL R10 + IL-3 + 5ug/mL CpG-A for 24 h. Supernatants were frozen at -80C and analyzed with VeriKine Human IFN Alpha Multi-Subtype ELISA Kit (PBL Assay Science).

For flow cytometry, cells were then stained for 20 minutes at 4C with a cocktail of Abs against surface markers diluted in FACS buffer (2mM EDTA, 2% Donor equine serum in PBS), except the chemokine receptors CCR2 and CCR7 that were stained at 37C for 45 minutes in PBS. For Ki67, TCF4 and IRF8 staining, cells were stained with LIVE/DEAD Fixable Blue (ThermoFisher) in PBS for detection of dead cells, stained for surface markers, then fixed using Foxp3 Transcription Factor Fix/Perm Buffer (ThermoFisher) for 1 hr and stained intracellularly for 45 minutes in 1X Permwash buffer (ThermoFisher).

For CyTOF, fresh PBMCs and day-6 cultured pDCs were pooled with mouse splenocytes ("cell bed"), stained with 0.25mM cisplatin (Fluidigim), surface-stained with heavy-metal-labeled antibodies, fixed with Foxp3 Fix/Perm Buffer (ThermoFisher), and stained intracellularly. Cells were incubated overnight with 2% paraformaldehyde (Electron) in PBS with 125nM Iridium intercalator (Fluidigm), washed, filtered, and acquired in a CyTOF2 (Fluidigm) at the Shared FACS Facility at Stanford University.

For multiome sequencing data, DCs were sorted as live cells before nuclei isolation. Nuclei were extracted following the demonstrated protocol CG000365 adapted for low cell input nuclei from 10X Genomics. In brief, 100,000 FACS-sorted cells were washed with 0.04% BSA and treated with chilled multiome lysis buffer (10 mM Tris-HCl pH 7.4 (TEKnova), 10mM NaCl (Fisher Scientific), 3 mM MgCl2 (Sigma Adrich), 0.1% Tween-20 (Roche), 0.1% IGEPAL CA-630 (Sigma Aldrich), 0.01% Digitonin (Promega), 1% BSA (Sigma Aldrich), 1 mM DTT (ThermoFisher), 1 U/ul RNase inhibitor (Sigma Aldrich) prepared in nuclease-free water (Cytiva)) on ice for 3 min to obtain nuclei. The isolated nuclei were washed twice with multiome wash buffer (10 mM Tris-HCl pH 7.4, 10mM NaCl, 3 mM MgCl2, 0.1% Tween-20, 1% BSA, 1 mM DTT, 1 U/ul RNase inhibitor prepared in nuclease-free water), and then resuspended with chilled multiome nuclei buffer (1X Nuclei Buffer (20X) (10X Genomics), 1 mM DTT, 1 U/ul RNase inhibitor prepared in nuclease free water). Immediately after nuclei preparation, single-cell libraries were prepared at the Genomics core facility of Stanford University using Chromium Next GEM Single-Cell Multiome ATAC + Gene Expression kit (10X Genomics), and following the manufacturer's protocol. Libraries were sequenced on an Illumina NovaSeq instrument.

For SMARTseq2 data, cells were single cell FACS-sorted into lysis buffer. The Takara Smart-Seq Single Cell kit was used for reverse transcription, cDNA synthesis and amplification (Takara). For bulk sequencing, cells were sorted into lysis buffer for RNAseq or into media for ATACseq. RNA/DNA extraction and library preparation was performed at Stanford Functional Genomic Core.

For bulk RNA-seq, 10,000 pDCs from six adult (24-30 years) and six elderly (73-89 years) donors were sorted directly into Qiazol lysis buffer (Qiagen) and frozen at -80C for RNA sequencing. RNA extraction and library preparation were performed by the Stanford Functional Genomics Core. For ATAC-seq, 10,000 pDCs were sorted into R10, and DNA was extracted and libraries prepared using the Omni-ATAC protocol at the Stanford Functional Genomic Core. DNA was stored at -20C after transposition until all samples were collected. Amplification and qPCR were performed simultaneously across samples. Library quality was assessed by Bioanalyzer. The 24 samples were barcoded, pooled and sequenced on a NovaSeq 6000 at the Stanford Functional Genomic Core.

| | |
|---|---|
| Instrument | Samples were analyzed on an LSR Fortessa X20 (BD Biosciences) or CYTEK Aurora (CYTEK). Sorting was performed on FACSAria II or Fusion (BD Biosciences). |
| Software | Data collection was performed with BD FACSDiva software (v8.01) or SpectroFlo software (v3.3.0). Data analysis was performed with FlowJo software v10.10.0 (Tree Star, Inc). |
| Cell population abundance | From each donor, approximately 50,000-100,000 cells were sorted from 50 mL of blood. Cell sorting gating strategies are shown in Extended Data Fig.1b, Extended Data Fig.2b, Extended Data Fig.5g, Extended Data Fig.6c, Extended Data Fig.10a., and Fig.2l Representative purification data is shown in Extended Data Fig.1b, Extended Data Fig.2b, Extended Data Fig.4b, Extended Data Fig.4e, Extended Data Fig.5h, Extended Data Fig.6c, Extended Data Fig.10a, and Fig.2l. Purity was >95% as determined by flow cytometry of postsort populations. |
| Gating strategy | Gating strategies for the analysis of DCs and other immune cells are shown in Extended Data Fig.1c, Extended Data Fig.4b, Extended Data Fig.4e, Extended Data Fig.4g, Extended Data Fig.5g-h, Extended Data Fig.6c, Extended Data Fig.8k, Extended Data Fig.10a, Extended Data Fig.10c, and Fig. 2c, Fig. 2d, Fig.2l, Fig.3i. Briefly, cells were gated on FSC/SSC, singlets, and live, prior to gate each population of interest. Positive gates were determined after titrating antibodies, using FMO or an isotype control antibody. |

☒ Tick this box to confirm that a figure exemplifying the gating strategy is provided in the Supplementary Information.

