## [Peer Review File · Nature Immunology]

TNF AND TYPE I INTERFERON CROSSTALK CONTROLS THE FATE AND FUNCTION OF PLASMACYTOID DENDRITIC CELLS

Corresponding Author: Dr Juliana Idoyaga

Version 0:

Decision Letter:

18th Jun 2024

Dear Dr Idoyaga

Thank you for submitting your manuscript entitled "CELL-FATE-SWITCHING OF PLASMACYTOID DENDRITIC CELLS TO CONVENTIONAL DENDRITIC CELLS", for consideration. I regret that we are unable to peer review your paper for Nature Immunology.

As you may know, we decline a substantial proportion of manuscripts without sending them to referees, so that they may be sent elsewhere without delay. These editorial judgments are based on such considerations as the degree of advance provided, the breadth of potential interest to researchers and timeliness.

Please be assured that this editorial decision does not represent a criticism of the quality of your work, nor are we questioning its value to others working in this area. We hope that you will rapidly receive a more favorable response elsewhere.

Although we cannot offer to publish your manuscript, I suggest that you consider Nature Communications as a suitable venue for this work. To transfer your manuscript, please use our manuscript transfer portal. You will not have to re-supply manuscript metadata and files, unless you wish to make modifications. For more information, please see our [manuscript transfer FAQ](http://www.nature.com/authors/author_resources/transfer_manuscripts.html?WT.mc_id=EMI_NPG_1511_AUTHORTRANSF&WT.ec_id=AUTHOR) page.

I am sorry that we cannot respond more positively on this occasion.

Sincerely,

Nick Bernard, PhD
Senior Editor
Nature Immunology

Version 1:

Decision Letter:

Dear Dr Idoyaga,

Thank you for your letter from (INSERT DATE) asking us to reconsider our decision on your manuscript, "CELL-FATE-SWITCHING OF PLASMACYTOID DENDRITIC CELLS TO CONVENTIONAL DENDRITIC CELLS".

Now that I have had a chance to discuss the matter carefully with my colleagues, I am happy to say that we would consider sending your manuscript to external review if you can add the proposed data. I'm sure, however, that you'll understand that we cannot predict the outcome of the review process.

Once you have made these revisions, please use the URL below to submit the revised manuscript with figures and a revised version of the life sciences reporting summary. It will be available to referees (and, potentially, statisticians) to aid in their

evaluation if the manuscript goes back for peer review. A revised checklist is essential for re-review of the paper.

The Reporting Summary can be found here:
<https://www.nature.com/documents/nr-reporting-summary.pdf>

The Editorial Policy Checklist can be found here: <https://www.nature.com/documents/nr-editorial-policy-checklist.pdf>

Link Redacted

Please let us know how you wish to proceed and when we can expect your revised manuscript.

With kind regards,

Nick Bernard, PhD
Senior Editor
Nature Immunology

Version 2:

Decision Letter:

25th Oct 2024

Dear Dr Idoyaga,

As you are aware, your Article, "CROSSTALK BETWEEN TNF AND IFN- γ DETERMINES PLASMACYTOID DENDRITIC CELL FATE AND FUNCTION" has now been seen by the original 3 referees and while they find your work of considerable potential interest, they have raised substantial concerns that must be addressed. In light of these comments, we cannot accept the manuscript for publication, but would be very interested in considering a revised version that addresses these serious concerns.

We have also looked over your Author Response to these comments (thanks for emailing that to me) and I am happy to say we are very pleased by your plans for revision. My only comment is that you might have not responded to this comment: "What is the authors explanation at the molecular level for the antagonism between IFN and TNF signaling pathways?"

If you choose to revise your manuscript as per your plan, please highlight all changes in the manuscript text file in Microsoft Word format.

* If you have not done so already please begin to revise your manuscript so that it conforms to our Article format instructions at <http://www.nature.com/ni/authors/index.html>. Refer also to any guidelines provided in this letter.

The Reporting Summary can be found here:
<https://www.nature.com/documents/nr-reporting-summary.pdf>

Extended Data figures and tables are online-only (appearing in the online PDF and full-text HTML version of the paper), peer-reviewed display items that provide essential background to the Article but are not included in the printed version of the paper due to space constraints or being of interest only to a few specialists. A maximum of ten Extended Data display items (figures and tables) is typically permitted. When re-submitting your manuscript, please ensure that any supplementary

figures and tables that are more critical to the manuscript's conclusions are converted to Extended data to increase these data's visibility.

Link Redacted

If you wish to submit a suitably revised manuscript we would hope to receive it within 6 months. If you cannot send it within this time, please let us know. We will be happy to consider your revision so long as nothing similar has been accepted for publication at Nature Immunology or published elsewhere.

Nature Immunology is committed to improving transparency in authorship. As part of our efforts in this direction, we are now requesting that all authors identified as 'corresponding author' on published papers create and link their Open Researcher and Contributor Identifier (ORCID) with their account on the Manuscript Tracking System (MTS), prior to acceptance. ORCID helps the scientific community achieve unambiguous attribution of all scholarly contributions. You can create and link your ORCID from the home page of the MTS by clicking on 'Modify my Springer Nature account'. For more information please visit www.springernature.com/orcid.

Thank you for the opportunity to review your work.

Sincerely,

Nick Bernard, PhD
Senior Editor
Nature Immunology

Reviewers' Comments:

Reviewer #1 (Remarks to the Author):

In this work, the authors investigate the fate-switching process of human pDC into antigen-presenting cDC2, and identify the environmental signals that regulate this process.

This work is highly relevant to the field and addresses a much-debated question. There are however some concerns that would need to be resolved before the manuscript is suitable for publication (see below).

Results reported in the manuscript are in line with previously published work, in particular Alculumbre et al (Nat Immunol 2017, 10.1038/s41590-017-0012-z). Although culture conditions were different in this previous study, conclusions are rather similar to the ones from the present manuscript, in particular cell state stability, differential functional properties (IFN-I production versus cDC features including antigen presentation and CCR7-directed migration) and the role of paracrine TNF in promoting antigen-presenting cells. The novelty of the present manuscript lies primarily in the antagonistic cross-talk between IFN α and TNF signaling pathways and the fate-switching in pDC from elderly people. These aspects need to be strengthened for the manuscript to go significantly beyond the existing literature.

Main issues

1. DC3 are absent from the PanDC Atlas generated in Figure 1 and Figure S1. Presumably, DC3 are removed during the enrichment step. Is there a proportion of cDC2 that could be contaminating DC3? Did the authors check this, and how? This needs to be addressed for clarity, even though it is not a main focus of the manuscript.
2. The in vivo validation of results from Figure 1 is not convincing enough. In Figure 1 k-n, it is unclear how the authors performed the analysis of clinical datasets. The text is too concise and too vague to allow a critical assessment of these results. Legends are unclear and it is difficult to understand what is shown in the panels. What are the UMAPs shown in l-n? Are these merged cells from all samples including healthy? What is the origin of the datasets (generated by the authors or downloaded from a public database)? Patients characteristics? What are the 'other skin lesions'? Methods used for cell clustering and identification, signature scoring, etc? The authors conclude that these clinical samples contain cells expressing C1 and C3 signatures. An alternative interpretation would be that these samples contain pDC and cDC2. The authors need to provide evidence that the C1 and C3 signatures are distinct from pDC and cDC2 signatures, and for comparison analyze the cDC2, tDC and C2 signatures as well in these datasets. The authors claim that these datasets provide an opportunity to analyze in vivo fate conversion, therefore one might expect some proportion of 'transitional' C2 cells as well. If the transitional cells are not identified, the authors need to propose some explanation for this.

3. The interpretation of results shown in Figure 2 is confusing. On one hand, the authors state that "pDCs, itDCs and icDC2s are stable states", i.e. they do not convert into other cell states. On the other hand, the authors conclude that "activated pDCs convert into icDC2s passing through an intermediate proliferative itDC cell state". These statements are apparently contradictory and the description of these results is difficult to follow. This needs to be clarified. In addition, sorting cells at day 4 to assess their conversion potential could be too late. To strengthen this part, the authors should assess the conversion potential of cells at an earlier stage, for instance day 2. The authors also need to exclude differential cell death in the culture over time for the different subpopulations.

4. The cross-talk between TNF and IFN α for fate-switching is a major point in the manuscript. However, it should be strengthened by additional experimental evidence. The authors were unable to measure type I interferon in the culture, which is known to be difficult to detect. However, they should analyze what happens when IFNAR signaling is blocked. This could be achieved for instance with B18R, a potent inhibitor for human IFNAR that is commercially available. What is the trigger for IFN α and TNF production in the culture? Is it CD40-L? What is the kinetics of TNF expression? Is it expressed from the start of the culture or at a later time point? The authors could also analyze what happens when blocking TNF in the culture, for instance using an anti-TNF blocking antibody, which is also commercially available.

What is the authors' explanation for the fact that only 50% of the pDC switch to icDC2? Is there heterogeneity in IFN α and TNF production? Did the authors try to perform intracellular flow cytometry for these cytokines? Is there heterogeneity in receptor expression for IFN α and TNF? Alternatively, results from figure 3m suggest that, in CD40L cultures, cell state transitions could be pushed by cytokines. How is fate-switching affected by different concentrations of TNF or IFN α ? Can all pDC be pushed to become icDC2?

Is there a link between the lineage-determining transcription factors analyzed in figure 4 b-i and TNF or IFN α ? What is the authors' explanation at the molecular level for the antagonism between IFN and TNF signaling pathways? These points should be at least discussed based on literature.

5. The authors suggest that pDC in elderly people are not "real" pDC anymore but switched to cDC2. In figure 5, the authors performed bulk RNA-seq. How did they verify the potential heterogeneity or contamination of their pDC population from elderly people? This is a key point. In addition, the authors should provide additional evidence that pDC in elderly people have converted to cDC2, for instance by analyzing some of their functional features as in figure 3.

Other points

- In figure S1, it is not entirely clear how the reference datasets were used. This should be better explained in the text and legend.
- Related to Figure 1, it is not clear why cluster C4 was not explored further (for instance in terms of gene expression and signature scores). This should be explained in the text. What is the interpretation of the authors regarding the position of C4 cluster in trajectory analyses?
- Related to Figure 3, it is unclear how the gene ontology analysis was performed. On which transcriptomic datasets?
- In Fig3e, it is unclear what the axis represents: "IFN α /cell"?
- In Fig3f, the results would be strengthened by showing the fluorescence-minus-one control stainings for each population, as background fluorescence might be different.
- Related to figure 3, is the chromatin accessibility of MARCH1 locus different in pDC, itDC and icDC2?
- In figure S3, it would be more informative to show, instead of heatmaps depicting mean value, graphs with values for each individual donor, which would allow assessing the variability between donors.

Reviewer #2 (Remarks to the Author):

The study of Hornero and colleagues addresses a long-standing question in DC biology, whether pDCs have the capacity once activated to differentiate into a conventional DC-like phenotype. As the authors note there has been multiple claims in this space and the validity of the pDC-cDC differentiation pathway has been disputed.

The current study uses human pDCs and multiple single cell omics approaches and some functional assays to tackle this problem. The authors conclude that pDC can indeed undergo a terminal differentiation into a cDC2 like state. They present evidence that this occurs in vitro and in vivo in inflammatory situations and possibly in ageing and suggest that TNF is a key driver of the process.

Overall, this is an interesting and innovative paper that addresses a problem that needs to be sorted out. I find the authors' interpretation of the data generally convincing, and the results raise very interesting questions for the interpretation of pDC function in inflammatory situations. I do have some comments that should be considered, particularly around the in vivo interpretation of the single cell transcriptomic data.

Specific comments

1. Figure 1K-n shows that pDCs are recruited to the skin in inflammatory diseases. That part is expected from the literature. The issue I have is with using the invitro generated C3 signature to show that many DCs also express the signature. The authors are careful with their conclusion (lines 114-155), but the inference is that these cells are potentially pDC derived. I feel this can easily mislead a reader as figure 1f clearly shows that the C3 signature scores highly for both cDC1 and cDC2 and is thus not helpful in distinguishing the origin of the cells. As the data on pDC diversity in figure 5 is much more compelling, I suggest moving the data in figure 1k-n to be associated with figure 5.

2. Figure 3h-k. the functional studies compare pDCs with icDC2 and shows some canonical cDC2 functionality. Bona fide cDC2 need to be included in this analysis, so the reader can get an idea to what extent the icDC2 functionally resemble mature cDC2. At present the evidence for icDC2 functionality looks relatively modest.

3. Figure 5h-j. The data on the proposed switch to an icDC2 phenotype in elderly individuals needs more validation to be convincing, particularly as the age difference 20-40 vs >65 is not huge, and the data is coming from bulk sorts. I suggest some high dimensional FACS is needed to validate this finding. The authors have identified several markers of the pDC to icDC2 differentiation process that could be used in the FACS panels to better characterise what is happening.
4. Figure 5j. This should be reduced to a single pan NFkB boxplot, as all genes listed bind the same DNA motif and thus the different plots are not meaningful.
5. It appears that an NFkB mediated signal is required for this process, and this signal can come from either CD40L or TNF. The expression pattern of CD40 is not clear to me. Ref 9 suggests that all pDC express CD40, but the authors RNA and protein data suggest that CD40 expression in pDC is very low. Perhaps plots with the kinetics of CD40 expression could be shown in FACS plots, instead of the normalised MFI shown in the extended data figure 2 which is not very informative. Is there evidence of heterogeneity of CD40 in the starting pDC population? Finally does IL3+CD40L result in TNF expression, thus explaining the requirement for CD40 signalling.

Reviewer #3 (Remarks to the Author):

In this manuscript, Arroyo Hornero et al. reveal an intriguing differentiation trajectory whereby IFN-producing pDCs can convert into cDC-like cells through a tDC intermediate stage. At the moment the DC field is chaotic given the vast heterogeneity, especially within the cDC2 family, that has been revealed by profiling these cells using new technologies like scRNAseq and advanced flow cytometry. This has left open questions about whether these new populations represent distinct cellular states or true cellular lineages. This manuscript makes a significant contribution to our understanding of this complexity, revealing an unexpected ability of pDCs to function as progenitors beyond their known role in IFN production in certain inflammatory settings. The study is conceptually novel, methodologically robust, and supported by solid data. However, we have a few suggestions that could strengthen the conclusions and improve the clarity of the manuscript.

Major

- Previously, the authors have shown that tDCs can give rise to cDC2A-like cells in the mouse¹. This aspect is not addressed in the current manuscript. Is there a preferential fate for pDCs in the human experiments, namely cDC2A or cDC2B? The authors could use human cDC2A vs cDC2B signatures that have been published before² to probe their existing data.
- Acknowledging the considerable difficulty of such experiments, we wonder if the authors attempted or considered performing their differentiation assays in a single-cell setup. This would provide definitive evidence of the potential of pDCs, excluding the possibility of small contaminating populations, to differentiate into cDC-like cells. Given that cells proliferate throughout the trans-differentiation trajectory, this is an important point. Alternatively, could barcoding be employed at the experiment's start, with clonality assessed at the end?
- Building on the previous suggestion, it would be useful to map a pre-cDC signature onto the panDC/culture UMAP in Figure 1 to verify that sorted pDCs do not contain contaminant pre-cDCs.
- Regarding human in vivo data (Fig. 1L and 5) – how do the authors confirm that the C3 signature superimposed onto UMAPS does not correspond to bona fide cDC2s (not pDC derived) that can also be recruited during inflammation? Given that transcriptionally, iDC2 and cDC2 appear difficult to distinguish, it is unclear why the authors assume that these correspond to icDC2. For instance, in Figure 5A, would be important to also compare the CD123-lo population with a cDC2 signature. Given that most likely these would also align transcriptionally, the authors should be careful with the interpretations of in vivo experiments as it is challenging to discriminate icDC2 from bona fide cDC2 in this context.
- In Figure 2I, to prove the stability of the icDC2 population, it would be important to perform this experiment in conditions where sorted icDC2 are re-cultured in the absence of CD40L to ensure these do not revert to prior pDC state. Also, the term stability is counterintuitive considering that the main finding is that pDCs convert into tDC and then into cDC2-like, perhaps saying “differentiation directionality” would make the purpose of the experiment more intuitively understood.

Minor

- In Figure 3, the authors perform various functional experiments clearly demonstrating superior APC functions of icDC2 compared to pDC or tDC and claim that “pDC convert into functional cDCs”. However, the functional capacities of icDC2 are never directly compared to bona fide cDC2. It would therefore be more appropriate to re-phrase the figure title/interpretation of these experiments as “pDC acquire cDC-like functions”.
- In Figure 4M, the graph legend “none” is confusing – does this mean the pDC were completely untreated or treated with CD40L but no additional cytokine?

Version 3:

Decision Letter:

Our ref: NI-A38122C

2nd Jun 2025

Dear Dr. Idoyaga,

Thank you for submitting your revised manuscript "CROSSTALK BETWEEN TNF AND IFN-I DETERMINES PLASMACYTOID DENDRITIC CELL FATE AND FUNCTION" (NI-A38122C). It has now been seen by the original referees

and their comments are below. The reviewers find that the paper has improved in revision, and therefore we'll be happy in principle to publish it in Nature Immunology, pending minor revisions to comply with our editorial and formatting guidelines.

We will now perform detailed checks on your paper and will send you a checklist detailing our editorial and formatting requirements in about a week. Please do not upload the final materials and make any revisions until you receive this additional information from us.

If you had not uploaded a Word file for the current version of the manuscript, we will need one before beginning the editing process; please email that to immunology@us.nature.com at your earliest convenience.

Thank you again for your interest in Nature Immunology Please do not hesitate to contact me if you have any questions.

Sincerely,

Nick Bernard, PhD
Senior Editor
Nature Immunology

Reviewer #1 (Remarks to the Author):

In the revised version of their manuscript, the authors have comprehensively and convincingly addressed all my comments and questions. The manuscript has been improved substantially during the revision process, and now provides compelling evidence to support the authors' conclusions.

Reviewer #2 (Remarks to the Author):

The authors have done an excellence job of revising this manuscript. The newer work on TNF/IFN cross talk and the improved analysis of changes in pDC during aging have improved the work and I have no further concerns.

Stephen Nutt

Reviewer #3 (Remarks to the Author):

I have carefully reviewed the revised version of the manuscript and the accompanying point-by-point rebuttal. I am pleased to say that the authors have addressed all of my previous comments to my full satisfaction. I want to commend them for the enormous amount of additional work they have undertaken to strengthen the study. The revised manuscript is clear, rigorous, and significantly improved. The final outcome is truly outstanding.

Carlos Minutti

POINT-BY-POINT RESPONSE

REVIEWER #1

In this work, the authors investigate the fate-switching process of human pDC into antigen-presenting cDC2, and identify the environmental signals that regulate this process. This work is highly relevant to the field and addresses a much-debated question. There are however some concerns that would need to be resolved before the manuscript is suitable for publication (see below). Results reported in the manuscript are in line with previously published work, in particular Alculumbre et al (Nat Immunol 2017, 10.1038/s41590-017-0012-z). Although culture conditions were different in this previous study, conclusions are rather similar to the ones from the present manuscript, in particular cell state stability, differential functional properties (IFN-I production versus cDC features including antigen presentation and CCR7-directed migration) and the role of paracrine TNF in promoting antigen-presenting cells. The novelty of the present manuscript lies primarily in the antagonistic cross-talk between IFN α and TNF signaling pathways and the fate-switching in pDC from elderly people. These aspects need to be strengthened for the manuscript to go significantly beyond the existing literature.

We thank the reviewer for highlighting the relevance of our work. In the revised manuscript, we have further emphasized the novel aspects of our study—specifically, the antagonistic crosstalk between TNF and IFN-I signaling, and the fate-switching capacity of pDCs from elderly donors—by incorporating additional experiments and refined analyses (see below).

To clarify the role of TNF/IFN-I crosstalk, we systematically evaluated pDC fate-switching under varying TNF and IFN-I concentrations and performed blockade experiments to demonstrate that TNF promotes conversion, while IFN-I restricts this transition (see below). Moreover, as suggested by the reviewer, we analyzed the mechanism by which TNF promotes fate-switching and found that it triggers downregulation of the pDC-defining transcription factor TCF4 (see below). These findings provide direct mechanistic evidence of how environmental cues regulate pDC plasticity.

We also expanded our analysis of aging by comparing the phenotype and function of pDCs from adult and elderly donors. Our results indicate that elderly pDCs show a progressive loss of lineage identity and convert more efficiently into icDC2s following CD40L stimulation. This underscores the potential role of pDC fate-switching in age-associated immune dysfunction.

JULIANA IDOYAGA, PHD

CHANCELLOR'S ASSOCIATE PROFESSOR OF PHARMACOLOGY & MOLECULAR BIOLOGY

UC San Diego Schools of Medicine and Biological Sciences

9500 Gilman Drive • Leichtag Family Foundation Biomedical Research Building, Room 284 • La Jolla, CA 92093

T: 858-822-0491 • jjidoyaga@health.ucsd.edu

In relation to Alculumbre et al. (*Nat. Immunol* 2017), we appreciate the reviewer's thoughtful comparison and agree that both studies investigate the broader theme of pDC plasticity. However, we respectfully emphasize that our experimental design, analytical methods, and conclusions are fundamentally distinct:

1. While Alculumbre et al. examined early pDC responses (24-48 hours) to viral stimuli, our study analyzed pDC differentiation over 2–6 days using CD40L stimulation. We found that pDC fate-switching is largely undetectable at 24 hours (**Extended Data Fig. 4f**), and commonly used activation markers (PDL1 and CD80) do not correspond with the transcriptional states we define in our study—pDCs, itDCs, and icDC2s (**Extended Data Fig. 8k**).
2. The transcriptional changes described by Alculumbre et al. primarily reflected short-term activation programs, including ISGs and NF- κ B target genes, which are expected in response to viruses. While we also observe NF- κ B activation in our system, our study captures broader transcriptional and epigenetic remodeling associated with a loss of pDC identity and acquisition of cDC2 features—including downregulation of TCF4 and upregulation of ID2—supporting an identity switch (**Fig.4a-i** and **Extended Data Fig.7**).
3. Alculumbre et al. proposed pDCs “diversification” into three stable states. In contrast, our data support a unidirectional differentiation trajectory, with pDCs transitioning through a proliferative itDC intermediate before acquiring a stable icDC2 identity. itDCs do not revert to pDCs, even upon IFN-I exposure, consistent with a committed rather than diversified response (**Fig.2m** and **Fig.4t**).
4. Alculumbre et al. did not directly address the possibility of tDC or pre-cDC contamination in their pDC preparations—a point that has since become a recognized confounding factor. Our study was specifically designed to avoid this issue: we used stringent sorting and validated our results with single-cell clonal differentiation assays (see below), demonstrating that pDC fate-switching is intrinsic and not due to contamination.
5. While Alculumbre et al. concluded that diversification is independent of IFN-I, we show that IFN-I actively inhibits pDC fate-switching by downregulating TNF receptor expression and maintaining high levels of TCF4 (**Fig.4u-v**). This provides new mechanistic insight into how IFN-I signaling preserves pDC identity and raises the question of whether fate-switching can occur in IFN-I-rich environments such as viral infections.

JULIANA IDOYAGA, PHD

CHANCELLOR'S ASSOCIATE PROFESSOR OF PHARMACOLOGY & MOLECULAR BIOLOGY

UC San Diego Schools of Medicine and Biological Sciences

9500 Gilman Drive • Leichtag Family Foundation Biomedical Research Building, Room 284 • La Jolla, CA 92093

T: 858-822-0491 • jjidoyaga@health.ucsd.edu

6. Prior studies assessed antigen presentation using mixed leukocyte reactions (MLR), which test MHCII expression but not antigen handling. In contrast, we directly evaluated the capacity of icDC2s to capture, process, and present antigen to autologous naïve antigen-specific T cells—providing functional validation of their conversion into cDC2-like cells (**Fig.3i-j**).
7. Finally, our study integrated single-cell transcriptomic, chromatin accessibility data and phenotypic analysis to define the trajectory from pDCs to icDC2s. We provide new insights into the mechanisms governing pDC fate-switching, including cell-to-cell variability in the expression of the TNFR that correlates with conversion potential (**Fig.4s**).

Together, these conceptual and technical differences distinguish our work from that of Alculumbre et al. and support a model in which pDC fate-switching is a unidirectional, regulated, cytokine-driven process that culminates in the acquisition of cDC2-like identity—rather than a stable diversification into multiple populations. We have now clarified these points in the Discussion section of the revised manuscript.

Main issues

1. DC3 are absent from the PanDC Atlas generated in Figure 1 and Figure S1. Presumably, DC3 are removed during the enrichment step. Is there a proportion of cDC2 that could be contaminating DC3? Did the authors check this, and how? This needs to be addressed for clarity, even though it is not a main focus of the manuscript.

We appreciate this comment. Given that DC3 share greater similarity with monocytes and originate from MDP progenitors (Cytlak et al., *Immunity* 2020; Bourdely et al., *Immunity* 2020), we intentionally excluded them from our PanDC Atlas. To achieve this, we purified DCs lacking CD14 expression (see **Extended Data Fig. 1b**), as DC3 are known to express CD14 in humans (Villani et al., *Science* 2017; Dutertre et al., *Immunity* 2019; see also **Fig.6d-f**).

To further address the reviewer's comment, we also analyzed the cDC2 transcriptome from our PanDC Atlas and aligned it with published DC2 and DC3 transcriptional signatures. As shown in the

JULIANA IDOYAGA, PHD

CHANCELLOR'S ASSOCIATE PROFESSOR OF PHARMACOLOGY & MOLECULAR BIOLOGY

UC San Diego Schools of Medicine and Biological Sciences

9500 Gilman Drive • Leichtag Family Foundation Biomedical Research Building, Room 284 • La Jolla, CA 92093

T: 858-822-0491 • jjidoyaga@health.ucsd.edu

revised manuscript (**Extended Data Fig. 1h**), the PanDC cDC2 transcriptome aligns with DC2 signatures but not DC3 signatures from Dutertre et al. *Immunity* 2019.

Together, these results show that our PanDC Atlas captures mainly DC2 with minimal or no DC3 contamination. These points have now been explicitly addressed in the revised manuscript.

2. The *in vivo* validation of results from Figure 1 is not convincing enough. In Figure 1 k-n, it is unclear how the authors performed the analysis of clinical datasets. The text is too concise and too vague to allow a critical assessment of these results. Legends are unclear and it is difficult to understand what is shown in the panels. What are the UMAPs shown in l-n? Are these merged cells from all samples including healthy? What is the origin of the datasets (generated by the authors or downloaded from a public database)? Patients characteristics? What are the 'other skin lesions'? Methods used for cell clustering and identification, signature scoring, etc? The authors conclude that these clinical samples contain cells expressing C1 and C3 signatures. An alternative interpretation would be that these samples contain pDC and cDC2. The authors need to provide evidence that the C1 and C3 signatures are distinct from pDC and cDC2 signatures, and for comparison analyze the cDC2, tDC and C2 signatures as well in these datasets. The authors claim that these datasets provide an opportunity to analyze *in vivo* fate conversion, therefore one might expect some proportion of 'transitional' C2 cells as well. If the transitional cells are not identified, the authors need to propose some explanation for this.

We thank the reviewer for these important points and apologize for the lack of clarity in our initial submission. Please note that following reviewer #2's suggestion, we have moved these data to **Extended Data Fig. 9** to improve clarity and focus of the main figures. We have now included detailed descriptions of the dataset and analyses in the revised manuscript (see figure legend of **Extended Data Fig.9** and material and methods, section 'Transcriptomic datasets of skin inflammation and skin wounding').

Briefly, in **Extended Data of Fig. 9**, we used a publicly available CITE-seq dataset (Liu et al., *Sci Immunol* 2022) to assess whether our *in vitro* signatures are also observed in tissues—or whether these signatures are unique to *in vitro* conditions and not representative of physiological states. To address this, we selected the skin as a model, given the absence of pDCs in healthy skin and their infiltration during skin inflammation. The original study (Liu et al., *Sci. Immunol* 2022) defined

JULIANA IDOYAGA, PHD

CHANCELLOR'S ASSOCIATE PROFESSOR OF PHARMACOLOGY & MOLECULAR BIOLOGY

UC San Diego Schools of Medicine and Biological Sciences

9500 Gilman Drive • Leichtag Family Foundation Biomedical Research Building, Room 284 • La Jolla, CA 92093

T: 858-822-0491 • jjidoyaga@health.ucsd.edu

metadata on donor skin conditions, including samples from healthy controls, as well as from patients with atopic dermatitis, psoriasis vulgaris, bullous pemphigoid, lichen planus, and clinicopathologically indeterminate rash. Myeloid cells were subclustered from pre-processed data based on *HLA-DRA*⁺ and *MS4A1*⁺ expression across all the donors (UMAP shown in **Extended Data Fig. 9a**). To assess pDC infiltration in inflamed skin, we focused on the UMAP visualization generated from healthy donors, all inflamed lesions, and from patients with psoriasis vulgaris or atopic dermatitis (**Extended Data Fig. 9d**). Transcriptomic and protein-level evidence of cellular identity is also shown (**Extended Data Fig. 9b-c**). A full description of the analysis has now been added to the Materials and Methods (section “*Transcriptomic datasets of skin inflammation and skin wounding*”).

Importantly, we have also clarified the interpretation of our alignment results. It was not our intention to imply that C1 (pDC) and C3 (icDC2) signatures are fundamentally distinct from canonical pDC and cDC2 signatures. On the contrary, our interpretation is that C1 corresponds to pDCs, and C3 overlaps with cDC2s. To emphasize this, and as suggested by the reviewer, we performed additional comparative analyses. Specifically, we aligned the Liu et al. dataset (*Sci. Immunol* 2022) to both our day-2 *in vitro*-activated pDCs (SMART-seq2 dataset) and our PanDC reference map (**Extended Data Fig. 9e**). As expected, both the PanDC-pDC and day2-pDC signatures aligned with the skin pDC cluster, while the PanDC-cDC2 and day2-icDC2 signatures overlapped with the skin cDC2s and activated cDC2s. The itDC signature primarily aligned with the cDC2 cluster (**Extended Data Fig. 9e**), which is consistent with our earlier observations that itDCs begin expressing cDC2 genes (**Figure 1g**). We did not detect a PanDC-tDC signature in Liu et al. dataset, likely due to the rapid conversion of tDCs into cDC2s (Sulczewski et al., *Nat. Immunol* 2024).

Together, these results show that the cell states identified *in vitro*—pDCs and icDC2s—have a transcriptomic profile that are also detectable *in vivo*. In other words, these are not an *in vitro* artifact, but transcriptional programs identifiable in human skin during inflammatory disease. However, as the reviewer noted, it is not possible to distinguish bona fide cDC2s from pDC-derived cDC2s in the Liu et al. dataset, since it includes a heterogeneous mixture of DC subsets. To address this limitation, we analyzed a publicly available datasets of sorted pDCs isolated from skin blisters (Chen et al., *JEM* 2019; **Fig. 5**), as well as our newly generated dataset of peripheral blood pDCs from elderly donors (**Fig. 6**). As discussed below, these datasets suggest that pDC undergo fate-switching *in vivo*.

JULIANA IDOYAGA, PHD

CHANCELLOR'S ASSOCIATE PROFESSOR OF PHARMACOLOGY & MOLECULAR BIOLOGY

UC San Diego Schools of Medicine and Biological Sciences

9500 Gilman Drive • Leichtag Family Foundation Biomedical Research Building, Room 284 • La Jolla, CA 92093

T: 858-822-0491 • jjidoyaga@health.ucsd.edu

3. The interpretation of results shown in Figure 2 is confusing. On one hand, the authors state that "pDCs, itDCs and icDC2s are stable states", i.e. they do not convert into other cell states. On the other hand, the authors conclude that "activated pDCs convert into icDC2s passing through an intermediate proliferative itDC cell state". These statements are apparently contradictory and the description of these results is difficult to follow. This needs to be clarified. In addition, sorting cells at day 4 to assess their conversion potential could be too late. To strengthen this part, the authors should assess the conversion potential of cells at an earlier stage, for instance day 2. The authors also need to exclude differential cell death in the culture over time for the different subpopulations.

We appreciate the reviewer's insightful comment and acknowledge that our description of the results in Figure 2 may have been unclear. To improve clarity, we have revised the text to adopt the terminology suggested by Reviewer #3—using "differentiation directionality" rather than "stable states".

To address the reviewer's concern regarding conversion potential at an earlier stage, we performed additional experiments in which cells were re-sorted at day 2 and re-cultured for either 2 or 4 more days (see **Extended Data Fig.4a-b**). These results show that icDC2s purified at day 2 retain their phenotype after prolonged culture, confirming their stability. Additionally, the majority of the pDCs also retain their phenotype. In contrast, 50-60% of day-2 itDCs differentiate into icDC2s, but not into pDCs. This directionality is further supported by experiments showing that itDCs do not revert to a pDC phenotype when cultured without CD40L (**Fig.2m**) or in the presence of IFN-I (**Fig.4t**), consistent with a unidirectional progression from pDCs through itDCs to icDC2s

To address concerns about differential cell death, we performed ApoTracker-based quantification of apoptosis over time (see **Extended Data Fig.4f**). Day-1 icDC2s showed slightly elevated—but modest—level of apoptosis, likely reflecting the technical challenge of assessing apoptosis by flow cytometry in very small cell numbers at this early time point (6% of the cells are icDC2s at day-1; **Extended Data Fig.4f**). In contrast, day-2 and day-4 icDC2s showed no significant increase in apoptosis, supporting the stability of this population. Similarly, pDCs and itDCs did not exhibit increased apoptosis throughout the culture period.

These additional experiments and clarifications have been incorporated into the revised manuscript to improve clarity.

4. The cross-talk between TNF and IFN α for fate-switching is a major point in the manuscript. However, it should be strengthened by additional experimental evidence. The authors were unable to measure type I interferon in the culture, which is known to be difficult to detect. However, they should analyze what happens when IFNAR signaling is blocked. This could be achieved for instance with B18R, a potent inhibitor for human IFNAR that is commercially available. What is the trigger for IFN α and TNF production in the culture? Is it CD40-L? What is the kinetics of TNF expression? Is it expressed from the start of the culture or at a later time point? The authors could also analyze what happens when blocking TNF in the culture, for instance using an anti-TNF blocking antibody, which is also commercially available.

We have now strengthened our conclusion that TNF-IFN-I crosstalk is critical for pDC fate-switching by performing the following experiments suggested by the reviewer:

1. **Determining the trigger and kinetics of TNF expression:** We now show that TNF secretion in the culture is induced by CD40L stimulation (see *Figure 4n*). Additionally, TNF levels peak early (day 1) and gradually decline thereafter (see *Extended Data Figure 8b*), indicating a rapid but transient TNF response.
2. **Blocking TNF signaling:** To further test the role of TNF, we blocked TNFR1/2. We found that TNF signaling blockade prevents CD40L-mediated differentiation of pDCs into icDC2s, confirming that TNF is the key cytokine driving fate-switching (see *Figure 4o* and *Extended Data Fig. 8c*).
3. **Blocking IFN-I signaling:** We assessed the effects of blocking IFN-I signaling using B18R. Our results show that blocking IFN-I does not enhance icDC2 conversion (see *Extended Data Figure 8h*), suggesting that IFN-I is not secreted by CD40L-stimulated pDCs as hinted by our ELISA measurements.

Altogether, our data support a model in which CD40L induces TNF secretion, which in turn promotes pDC differentiation into icDC2s via TNFR1/2 signaling. Blocking IFN-I signaling does not alter

pDC fate switching, further suggesting that IFN-I is not induced by CD40L, consistent with our inability to detect it in the supernatant.

These new data and analyses have been incorporated into the revised manuscript.

What is the authors' explanation for the fact that only 50% of the pDC switch to icDC2? Is there heterogeneity in IFN α and TNF production? Did the authors try to perform intracellular flow cytometry for these cytokines? Is there heterogeneity in receptor expression for IFN α and TNF? Alternatively, results from figure 3m suggest that, in CD40L cultures, cell states transitions could be pushed by cytokines. How is fate-switching affected by different concentrations of TNF or IFN α ? Can all pDC be pushed to become icDC2?

Our findings support a model in which differential sensing of microenvironmental cues—specifically TNF—is the primary determinant of pDC fate-switching. As the reviewer may know, intracellular cytokine staining requires the addition of Brefeldin A, which disrupts the feedback loops regulating cytokine expression. Therefore, we did not use this approach to assess heterogeneity in cytokine production at the single-cell level. However, to address the reviewer's question and investigate potential heterogeneity within the starting pDC population, we performed the following experiments:

1. **Effect of TNF on fate-switching:** To determine whether TNF levels influence fate-switching, we tested a range of TNF concentrations (2–2000 ng/ml) in both day-2 and day-4 cultures. Surprisingly, pDC-to-icDC2 conversion rates remained consistent across all TNF doses and time points tested (see **Figure 4q-r**). Additionally, to test whether TNF depletion over time might impact conversion, we added TNF on day 2 to refresh the medium, but this did not further enhance pDC-to-icDC2 differentiation further (see **Figure 4r**). These findings indicate that TNF levels alone are not a limiting factor for fate-switching and suggest that additional regulatory mechanisms are involved.
2. **Heterogeneity in receptor expression:** To further investigate why only 20–40% of pDCs undergo fate-switching—independent of TNF levels—we analyzed TNFR1 and TNFR2

JULIANA IDOYAGA, PHD

CHANCELLOR'S ASSOCIATE PROFESSOR OF PHARMACOLOGY & MOLECULAR BIOLOGY

UC San Diego Schools of Medicine and Biological Sciences

9500 Gilman Drive • Leichtag Family Foundation Biomedical Research Building, Room 284 • La Jolla, CA 92093

T: 858-822-0491 • jjidoyaga@health.ucsd.edu

expression in both freshly purified and CD40L-stimulated pDCs. We found that only a small fraction of freshly isolated pDCs express TNFR1 (10-30%), while TNFR2 is expressed in 60-80% of pDCs (see **Extended Data Figure 8g**). Importantly, blocking only TNFR1 (but not TNFR2) results in a significant decrease in icDC2s (see **Figure 4o**). Moreover, donor-specific expression of TNFR1 (but not TNFR2) correlates with the percentage of pDCs undergoing fate-switching across donors (see **Figure 4s**), supporting the hypothesis that cell-to-cell variation in TNFR1 expression determines fate-switching.

3. **Effect of IFN-I on fate-switching:** We evaluated whether different doses of IFN- α alter the frequency of pDCs versus icDC2s in culture. Higher concentrations of IFN- α progressively reduced icDC2 conversion, suggesting that IFN-I levels play a key inhibitory role in fate-switching (see **Extended Data Figure 8a**).
4. **Pre-existing IFN-I imprinting in pDCs:** Although we did not detect IFN-I in culture supernatants and blocking IFN-I signaling with B18R did not alter icDC2 conversion, we observed that a subset of freshly isolated pDCs express interferon-stimulated genes (ISG) (see **Extended Data Figure 8i**). This suggests that some pDCs may have prior *in vivo* IFN-I exposure, potentially pre-conditioning them against fate-switching. While this hypothesis is now discussed in the revised manuscript, validating this mechanism would require further experimentation beyond the scope of this study.

Altogether, these results suggest that pDC fate-switching is regulated by cytokine signaling and intrinsic heterogeneity in TNFR expression. In addition, our findings support the idea that prior IFN-I exposure may pre-condition certain pDCs against fate-switching. These insights have now been incorporated into the revised manuscript.

Is there a link between the lineage-determining transcription factors analyzed in figure 4 b-i and TNF or IFN α ? What is the authors explanation at the molecular level for the antagonism between IFN and TNF signaling pathways? These points should be at least discussed based on literature.

We thank the reviewer for this insightful comment, which prompted us to performed new and important experiments. We now demonstrate that TNF promotes downregulation of TCF4, a key

JULIANA IDOYAGA, PHD

CHANCELLOR'S ASSOCIATE PROFESSOR OF PHARMACOLOGY & MOLECULAR BIOLOGY

UC San Diego Schools of Medicine and Biological Sciences

9500 Gilman Drive • Leichtag Family Foundation Biomedical Research Building, Room 284 • La Jolla, CA 92093

T: 858-822-0491 • jjidoyaga@health.ucsd.edu

transcription factor required for maintaining pDC identity. Blocking TNFR1/2 prevents TCF4 downregulation (see **Figure 4p**), and this correlates with a significant reduction in icDC2 differentiation (see **Figure 4o**). These results are consistent with previous findings that monocyte-derived TNF downregulates TCF4 (Dewald et al., *Viruses* 2020).

Furthermore, we show that IFN-I downregulates TNFR1/2 expression (see **Figure 4u**) and maintains high levels of TCF4 (see **Figure 4v**), providing a molecular explanation for the antagonism between TNF and IFN-I signaling in regulating pDC fate.

These mechanistic insights are now presented in the revised manuscript and discussed in the context of the existing literature.

5. The authors suggest that pDC in elderly people are not "real" pDC anymore but switched to cDC2. In figure 5, the authors performed bulk RNA-seq. How did they verify the potential heterogeneity or contamination of their pDC population from elderly people? This is a key point.

We appreciate the reviewer's concern regarding the potential contamination of pDC populations from elderly donors in our bulk-seq analysis (**Fig.6**). While single-cell RNA-seq is a powerful tool for assessing cellular heterogeneity, its sequencing depth is often insufficient to capture transcriptional differences between closely related populations, such as pDCs from elderly versus adult donors. For this reason, we opted for bulk RNA-seq, which provides greater sequencing depth and enables a more comprehensive comparison of transcriptional changes between these populations.

To verify our observations and exclude the possibility of contamination, we implemented the following approaches and performed the following new experiments:

1. **Stringent pDC purification:** pDCs from elderly and adult donors were stringently purified through enrichment followed by cell sorting (FACS), ensuring the exclusion of contaminating DC subsets or precursors. Importantly, we applied the same gating strategy across all samples, using markers that distinguish *bona fide* pDCs from tDCs. To enhance transparency, we now include the sorting strategy and post-sort purity for each donor analyzed in bulk-seq (see **Extended Data Figure 10a**).

2. **Single-cell analysis of pDC identity markers:** To further validate our findings, we performed single-cell multidimensional phenotypic analysis by CyTOF and flow cytometry. We found that pDCs from elderly and adult donors clustered together and remain distinct from other DC subsets (**Figure 6g**). However, comparative analysis revealed that elderly pDCs exhibit relatively reduced expression of pDC-defining proteins and increased expression of cDC-defining proteins (see **Figure 6g-h**). We confirmed these observations in two different cohorts of elderly and adult donors using flow cytometry, showing that although still clustering together, elderly pDCs exhibit significantly reduced expression of the transcription factors TCF4 and IRF8 (**Fig. 6i and Extended Data Fig.10d**) (the latter transcription factor being a known target of TCF4 (Leylek et al., *Cell Rep* 2020; Cisse et al., *Cell* 2008).

Together, these results confirm that our sorted pDCs from elderly donors are not contaminated with cDC2s but instead exhibit an intrinsic transcriptional and phenotypic shift toward a cDC2-like state.

In addition, the authors should provide additional evidence that pDC in elderly people have converted to cDC2, for instance by analyzing some of their functional features as in figure 3.

In the revised manuscript, we now show that a higher frequency of pDCs from elderly donors undergo fate-switching into icDC2s compared to adult pDCs (see **Fig.6k**). This increased conversion after culturing with CD40L correlates with greater phagocytic capacity for particulate antigen, while uptake of soluble DQ-OVA remains unchanged (see **Fig.6l**).

Notably, as shown in **Figure 6j**, elderly donors exhibit significantly lower numbers of pDCs. Due to these constraints, we were unable to directly compare the antigen-presenting capacity of elderly versus adult pDCs in this study.

Altogether, these findings support the idea that elderly pDCs undergo functional reprogramming, reinforcing their shift toward a cDC2-like state. This discussion has now been explicitly incorporated into the revised manuscript.

Other points

JULIANA IDOYAGA, PHD

CHANCELLOR'S ASSOCIATE PROFESSOR OF PHARMACOLOGY & MOLECULAR BIOLOGY

UC San Diego Schools of Medicine and Biological Sciences

9500 Gilman Drive • Leichtag Family Foundation Biomedical Research Building, Room 284 • La Jolla, CA 92093

T: 858-822-0491 • jjidoyaga@health.ucsd.edu

- In figure S1, it is not entirely clear how the reference datasets were used. This should be better explained in the text and legend.

We have added further details to the main text, methods (section “*Differentially Expressed Gene Analysis*”, and figure legend (*Extended Data Fig.1g-i*) in the revised version of the manuscript.

- Related to Figure 1, it is not clear why cluster C4 was not explored further (for instance in terms of gene expression and signature scores). This should be explained in the text. What is the interpretation of the authors regarding the position of C4 cluster in trajectory analyses?

Cluster C4 is characterized by a strong proliferation signature, as shown in *Extended Data Figure 2e*. C4 positions close to C2 (itDC), correlating with the higher proliferative capacity of itDCs (*Fig.2n*). Consequently, since it represents a transcriptional state associated with cell cycling rather than a distinct functional subset, we did not perform separate analysis of its gene expression profile or signature scores. This rationale is now explicitly clarified in the revised manuscript.

- Related to Figure 3, it is unclear how the gene ontology analysis was performed. On which transcriptomic datasets?

We apologize for the lack of clarity in the original submission. In the revised manuscript, we clarify that the gene ontology (GO) analysis in *Figure 3* was performed using the SMART-seq2 single-cell transcriptomic dataset derived from day-2 cultured cells. Additionally, we have explicitly described the GO analysis workflow in the Materials and Methods (section “*Gene Set Enrichment Analysis*”) to ensure full transparency and reproducibility.

- In Fig3e, it is unclear what the axis represents: "IFNa/cell"?

We appreciate the reviewer's comment. "IFN- α /cell" refers to the amount of IFN- α secreted, normalized to the number of cells cultured in each condition. This clarification has now been explicitly added to the revised manuscript and figure legend.

- In Fig3f, the results would be strengthened by showing the fluorescence-minus-one control staining for each population, as background fluorescence might be different.

As requested by the reviewer, we have now included fluorescence-minus-one (FMO) control staining for each marker to account for background fluorescence (**Fig.3f**).

- Related to figure 3, is the chromatin accessibility of MARCH1 locus different in pDC, itDC and icDC2?

We thank the reviewer for raising this point. As shown in **Figure 3d**, MARCH1 gene expression is downregulated in icDC2s. In the revised manuscript, we now extend this analysis by examining chromatin accessibility at the MARCH1 locus in pDCs, itDCs, and icDC2s using our scATAC-seq data (**Extended Data Fig.6b**).

- In figure S3, it would be more informative to show, instead of heatmaps depicting mean value, graphs with values for each individual donors, which would allow assessing the variability between donors.

As requested, we have now included bar graphs with values for each individual donor measuring phagocytic and antigen processing capacity (**Extended Data Fig.5c-f**).

REVIEWER #2

(Remarks to the Author)

The study of Hornero and colleagues addresses a long-standing question in DC biology, whether pDCs have the capacity once activated to differentiate into a conventional DC-like phenotype. As the authors

JULIANA IDOYAGA, PHD

CHANCELLOR'S ASSOCIATE PROFESSOR OF PHARMACOLOGY & MOLECULAR BIOLOGY

UC San Diego Schools of Medicine and Biological Sciences

9500 Gilman Drive • Leichtag Family Foundation Biomedical Research Building, Room 284 • La Jolla, CA 92093

T: 858-822-0491 • jjidoyaga@health.ucsd.edu

note there has been multiple claims in this space and the validity of the pDC-cDC differentiation pathway has been disputed.

The current study uses human pDCs and multiple single cell omics approaches and some functional assays to tackle this problem. The authors conclude that pDC can indeed undergo a terminal differentiation into a cDC2 like state. They present evidence that this occurs in vitro and in vivo in inflammatory situations and possibly in ageing and suggest that TNF is a key driver of the process.

Overall, this is an interesting and innovative paper that addresses a problem that needs to be sorted out. I find the authors' interpretation of the data generally convincing, and the results raise very interesting questions for the interpretation of pDC function in inflammatory situations. I do have some comments that should be considered, particularly around the in vivo interpretation of the single cell transcriptomic data.

We thank the reviewer for their comments highlighting the relevance, novelty and strength of our work.

Specific comments

1. Figure 1K-n shows that pDCs are recruited to the skin in inflammatory diseases. That part is expected from the literature. The issue I have is with using the invitro generated C3 signature to show that many DCs also express the signature. The authors are careful with their conclusion (lines 114-155), but the inference is that these cells are potentially pDC derived. I feel this can easily mislead a reader as figure 1f clearly shows that the C3 signature scores highly for both cDC1 and cDC2 and is thus not helpful in distinguishing the origin of the cells. As the data on pDC diversity in figure 5 is much more compelling, I suggest moving the data in figure 1k-n to be associated with figure 5.

We agree with the reviewer's comment and appreciate the suggestion to relocate Figure 1k-n to the extended data of Figure 5, which has been implemented in the revised manuscript (**Extended Data Fig.9**). Additionally, we have clarified in the text that C3 signature is shared across multiple DC subsets and does not specifically distinguish pDC-derived cells in inflamed skin.

JULIANA IDOYAGA, PHD

CHANCELLOR'S ASSOCIATE PROFESSOR OF PHARMACOLOGY & MOLECULAR BIOLOGY

UC San Diego Schools of Medicine and Biological Sciences

9500 Gilman Drive • Leichtag Family Foundation Biomedical Research Building, Room 284 • La Jolla, CA 92093

T: 858-822-0491 • jidoyaga@health.ucsd.edu

2. Figure 3h-k. the functional studies compare pDCs with icDC2 and shows some canonical cDC2 functionality. Bona fide cDC2 need to be included in this analysis, so the reader can get an idea to what extent the icDC2 functionally resemble mature cDC2. At present the evidence for icDC2 functionality looks relatively modest.

Thank you for this comment. In the revised manuscript, we have included data comparing freshly isolated cDC2s and pDCs from the same donor in their ability to capture antigen, process it, and present it to T cells (see *Fig. 3h* and *Extended Data Fig. 6d*).

Notably, we were unable to directly compare freshly isolated and culture-induced cells from the same donor side by side due to the number of cells required for these experiments and the timing differences in sample collection—freshly isolated cells had to be analyzed on day 0 immediately after sorting from fresh blood, whereas cultured cells were analyzed on day 4. Nevertheless, we believe that our analyses provide a relevant comparison to the functional assays performed on cultured cells, as experimental conditions were standardized (even though the cells were obtained from different donors).

3. Figure 5h-j. The data on the proposed switch to an icDC2 phenotype in elderly individuals needs more validation to be convincing, particularly as the age difference 20-40 vs >65 is not huge, and the data is coming from bulk sorts. I suggest some high dimensional FACS is needed to validate this finding. The authors have identified several markers of the pDC to icDC2 differentiation process that could be used in the FACS panels to better characterise what is happening.

We thank the reviewer for this insightful comment. Based on their suggestion, we performed CyTOF analysis in a different cohort of 6 adult and 6 elderly donors to further characterize age-related differences in pDCs. Our analysis shows that while adult and elderly pDCs cluster together and remain distinct from other DC subsets, elderly pDCs exhibit relatively lower expression of canonical pDC markers compared to adult pDCs (see *Fig. 6g-h*). To further validate these findings, we measured protein levels of two key transcription factors, TCF4 and IRF8, in two independent cohorts of donors.

JULIANA IDOYAGA, PHD

CHANCELLOR'S ASSOCIATE PROFESSOR OF PHARMACOLOGY & MOLECULAR BIOLOGY

UC San Diego Schools of Medicine and Biological Sciences

9500 Gilman Drive • Leichtag Family Foundation Biomedical Research Building, Room 284 • La Jolla, CA 92093

T: 858-822-0491 • jjidoyaga@health.ucsd.edu

Consistently, we found that elderly pDCs express these transcription factors at lower levels than adult pDCs (see **Fig. 6i** and **Extended Data Fig. 10d**).

Additionally, we quantified pDC numbers in blood, as now reported in the revised manuscript. Consistent with previous findings, elderly individuals have significantly fewer circulating pDCs than adults (see **Fig. 6j**).

Finally, as suggested by Reviewer #1, we compared the fate-switching capacity of elderly vs. adult pDCs following CD40L stimulation. Our results demonstrate that elderly pDCs differentiate into icDC2s more efficiently than adult pDCs (see **Fig. 6k**). Consequently, cultured elderly pDCs exhibit a superior capacity to capture particulate antigen compared to cultured adult pDCs (see **Fig. 6l**).

These new findings further support our conclusion that pDCs in elderly individuals gradually lose their canonical identity.

4. Figure 5j. This should be reduced to a single pan NFkB boxplot, as all genes listed bind the same DNA motif and thus the different plots are not meaningful.

In the revised manuscript, we have shown a single plot in the main figure, as suggested (see **Fig.6c**). We have moved other gene boxplots to the **Extended Data Figure 10b**.

5. It appears that an NFkB mediated signal is required for this process, and this signal can come from either CD40L or TNF. The expression pattern of CD40 is not clear to me. Ref 9 suggests that all pDC express CD40, but the authors RNA and protein data suggest that CD40 expression in pDC is very low. Perhaps plots with the kinetics of CD40 expression could be shown in FACS plots, instead of the normalised MFI shown in the extended data figure 2 which is not very informative. Is there evidence of heterogeneity of CD40 in the starting pDC population? Finally does IL3+CD40L result in TNF expression, thus explaining the requirement for CD40 signaling.

We thank the reviewer for this insightful question, which prompted us to perform additional experiments to clarify the role of CD40 and TNF signaling in pDC fate-switching.

1. CD40 expression dynamics:

JULIANA IDOYAGA, PHD

CHANCELLOR'S ASSOCIATE PROFESSOR OF PHARMACOLOGY & MOLECULAR BIOLOGY

UC San Diego Schools of Medicine and Biological Sciences

9500 Gilman Drive • Leichtag Family Foundation Biomedical Research Building, Room 284 • La Jolla, CA 92093

T: 858-822-0491 • jjidoyaga@health.ucsd.edu

- In the revised manuscript, we now show that only ~3–5% of freshly isolated pDCs express CD40 (see **Extended Data Fig. 8d**).
- However, CD40 expression is rapidly upregulated within one day of culture with IL-3 + CD40L and ~60–80% of all the population becomes CD40⁺ (see **Extended Data Fig. 8d**).
- To better illustrate this point, we now show % of CD40⁺ cells over time (**Extended Data Fig. 8d**).

2. TNF secretion following CD40L stimulation:

- As suggested by the reviewer, we now show that CD40L promotes secretion of TNF (see **Figure 4n**).
- TNF secretion peaks at day 1, indicating rapid production following CD40L stimulation (see **Extended Data Figure 8b**).
- TNF secretion is paralleled by IL-8, a TNF-stimulated cytokine (**Figure 4n**) (Decalf et al., *JEM* 2007).

3. TNF receptor expression and its role in fate-switching:

- In freshly isolated pDCs, TNFR1 is expressed in ~10–30% of cells, while TNFR2 is expressed by 60–80% of the population (see **Extended Data Fig. 8g**). Notably, TNFR1 expression correlates with the frequency of icDC2s in the culture across donors (see **Fig. 4s**).
- Blocking TNFR1/2 significantly reduces pDC-to-icDC2 conversion, confirming that TNF signaling is essential for fate-switching (see **Figure 4o** and **Extended Data Figure 8c**).

Altogether, our findings support a model in which CD40L triggers TNF secretion, and TNF signaling through TNFR1/2 is necessary for pDC fate conversion (**Fig. 4w**). The widespread upregulation of CD40 in culture appears to be, at least in part, a downstream consequence of TNF signaling, as TNFR1/2 blockade significantly reduced CD40 expression (**Extended Data Fig. 8e**). Notably, we also show that TNF promotes TCF4 downregulation (**Fig. 4p**), providing a molecular mechanism for the loss of pDC identity.

These results have now been incorporated into the revised manuscript.

JULIANA IDOYAGA, PHD

CHANCELLOR'S ASSOCIATE PROFESSOR OF PHARMACOLOGY & MOLECULAR BIOLOGY

UC San Diego Schools of Medicine and Biological Sciences

9500 Gilman Drive • Leichtag Family Foundation Biomedical Research Building, Room 284 • La Jolla, CA 92093

T: 858-822-0491 • jidoyaga@health.ucsd.edu

REVIEWER #3

(Remarks to the Author)

In this manuscript, Arroyo Hornero et al. reveal an intriguing differentiation trajectory whereby IFN-producing pDCs can convert into cDC-like cells through a tDC intermediate stage. At the moment the DC field is chaotic given the vast heterogeneity, especially within the cDC2 family, that has been revealed by profiling these cells using new technologies like scRNAseq and advanced flow cytometry. This has left open questions about whether these new populations represent distinct cellular states or true cellular lineages. This manuscript makes a significant contribution to our understanding of this complexity, revealing an unexpected ability of pDCs to function as progenitors beyond their known role in IFN production in certain inflammatory settings. The study is conceptually novel, methodologically robust, and supported by solid data. However, we have a few suggestions that could strengthen the conclusions and improve the clarity of the manuscript.

We thank the reviewer for their thoughtful comments highlighting the relevance, novelty, and strength of our work.

Major

- Previously, the authors have shown that tDCs can give rise to cDC2A-like cells in the mouse. This aspect is not addressed in the current manuscript. Is there a preferential fate for pDCs in the human experiments, namely cDC2A or cDC2B? The authors could use human cDC2A vs cDC2B signatures that have been published before to probe their existing data.

We thank the reviewer for this insightful point. To address this, we aligned pDC-derived icDC2s with published signature scores for DC2 (also known as cDC2A in mice) and DC3 (cDC2B in mice). Our analysis shows that icDC2s exhibit stronger alignment with the DC2 signature, suggesting a preferential differentiation toward this subset (see *Extended Data Figure 4h*). Additionally, we

examined the expression of key markers distinguishing DC2 from DC3. icDC2s express CD5, a known DC2 surface marker, and lack expression of CD14 and CD163, which are characteristic of DC3 (see **Extended Data Figure 4i**).

Altogether, these findings suggest that, like tDCs in mice and human (Sulczewski et al., *Nat Immunol* 2024), human pDCs preferentially differentiate into DC2- rather than DC3-like cells. These results have now been incorporated into the revised manuscript.

• Acknowledging the considerable difficulty of such experiments, we wonder if the authors attempted or considered performing their differentiation assays in a single-cell setup. This would provide definitive evidence of the potential of pDCs, excluding the possibility of small contaminating populations, to differentiate into cDC-like cells. Given that cells proliferate throughout the trans-differentiation trajectory, this is an important point. Alternatively, could barcoding be employed at the experiment's start, with clonality assessed at the end?

We acknowledge that this was an extremely challenging experiment to perform—but we are thrilled by the results, and we thank the reviewer for the challenge!

To address this question, we designed a single-cell differentiation assay using CellTrace Violet (CTV) and CFSE labeling. Specifically, we:

- Labeled half of the pDCs with CFSE to serve as the culture bed (filler cells), providing all necessary signals for fate conversion, including TNF.
- Labeled the other half with CTV and seeded them at varying densities (1, 10, 100, 1000, or 5000 cells, the latter matching the number of bed cells).
- Repeated this setup across four different blood donors in 4 independent experiments.

Key Findings (see **Figure 2p** and **Extended Data Figure 4g**):

- A single pDC can indeed undergo fate-switching and differentiate into itDCs and icDC2s, providing definitive evidence that pDCs have intrinsic differentiation potential, and excluding the possibility of small contaminating populations.
- The recovery of these experiments was low, consistent with our previous findings that these cells do not proliferate significantly (a point that is now explicitly explain in **Figure 2o**).

JULIANA IDOYAGA, PHD

CHANCELLOR'S ASSOCIATE PROFESSOR OF PHARMACOLOGY & MOLECULAR BIOLOGY

UC San Diego Schools of Medicine and Biological Sciences

9500 Gilman Drive • Leichtag Family Foundation Biomedical Research Building, Room 284 • La Jolla, CA 92093

T: 858-822-0491 • jjidoyaga@health.ucsd.edu

- Not all single pDCs undergo fate-switching, suggesting intrinsic heterogeneity within the population (a point explained by the cell-to-cell variability in expression of TNFR1, as show in **Fig. 4s**).
- As the number of pDCs in culture increases, the frequency of fate-switching events begins to resemble “filler cell” condition, further supporting the idea that the ability of pDCs to differentiate is, at least in part, pre-determined by their expression of TNFR1 (**Fig. 4s**).

These results provide compelling evidence that pDCs can differentiate into cDC-like cells while highlighting important cell-intrinsic and environmental regulatory factors (further explored in **Figure 4**).

- Building on the previous suggestion, it would be useful to map a pre-cDC signature onto the panDC/culture UMAP in Figure 1 to verify that sorted pDCs do not contain contaminant pre-cDCs.

As suggested by the reviewer, we mapped two human pre-cDC signatures onto our PanDC dataset to assess the presence of potential contaminating pre-cDCs:

- The Regev signature CD100^{hi} progenitor signature (*Villani et al., Science 2017*) mapped to a small cluster of cells, which we have now labeled "CD100^{hi} progenitors" in the revised manuscript (see **Extended Data Figure 1i**). The progenitor cells were originally identified within the HLA-DR⁺CD11c⁻CD123⁻ population at the protein expression. We have now show that this population is present in the PanDC dataset, but absent from sorted pDCs (**Extended Data Figure 1c and 2b**), further supporting the purity of our preparations.
- The Regev ASDCs/AXL⁺ DCs signature (*Villani et al., Science 2017*) mapped to tDCs as previously described (**Extended Data Figure 1i**) (*Leylek et al., Cell Rep 2019*).
- The Ginhoux pre-DC signature (See et al., *Science 2017*) mapped to tDCs (see **Extended Data Figure 1i**), consistent with our previous findings (*Leylek et al., Cell Rep 2019*). While Ginhoux group has proposed that tDCs/ASDCs represent pre-DCs, this remains a topic of ongoing debate, and we and others have suggested that this is not the case (*Villani et al., Science 2017*; *Leylek et al., Cell Rep 2019*; *Sulczewski et al., Nat Immunol 2024*).

Together, these results confirm that our sorted pDC population does not contain contaminating pre-cDCs. These findings have now been incorporated into the revised manuscript.

- Regarding human *in vivo* data (Fig. 1L and 5) – how do the authors confirm that the C3 signature superimposed onto UMAPS does not correspond to bona fide cDC2s (not pDC derived) that can also be recruited during inflammation? Given that transcriptionally, iDC2 and cDC2 appear difficult to distinguish, it is unclear why the authors assume that these correspond to icDC2. For instance, in Figure 5A, would be important to also compare the CD123-lo population with a cDC2 signature. Given that most likely these would also align transcriptionally, the authors should be careful with the interpretations of *in vivo* experiments as it is challenging to discriminate icDC2 from bona fide cDC2 in this context.

Thank you for this important comment, which prompted us to clarify several aspects of our *in vivo* analyses in the revised manuscript. The main changes are summarized below:

- As the reviewer correctly pointed out, the Liu et al. (2022) dataset contains a mixture of DC subsets, making it impossible to definitively distinguish bona fide cDC2s from pDC-derived icDC2s. Our intention with this analysis was not to claim that all cDC2-like cells in inflamed skin are pDC-derived, but rather to demonstrate that the transcriptional signatures identified *in vitro*—particularly the icDC2 signature—are also detectable in inflamed human tissues, supporting their physiological relevance and arguing against an *in vitro* artifact. To strengthen this point, we focused on skin—a setting where pDCs are absent in healthy skin and only appear upon recruitment during inflammation. We have clarified these points in the revised manuscript and moved this original Figure 1I to **Extended Data Fig. 9**, as suggested by Reviewer #2.
- Given that C3 cluster (icDC2s) has a similar transcriptional signature with freshly isolated cDC2 (**Figure 1**), we expect that C3 and cDC2s will align in the UMAP of Liu et al. (2022) dataset. Indeed, we now show that pDC, cDC2 and activated cDC2 clusters from Liu et al. dataset aligned with pDCs and icDC2s from our *in vitro*-culture SMART-seq2 dataset, as well as with freshly isolated pDCs and cDC2s from the PanDC reference map (**Extended Data Fig. 9a, e**). These results confirm that the transcriptional reprogramming observed *in vitro* is recapitulated *in vivo*. However, as the reviewer points out, we have not identified a unique transcriptional

JULIANA IDOYAGA, PHD

CHANCELLOR'S ASSOCIATE PROFESSOR OF PHARMACOLOGY & MOLECULAR BIOLOGY

UC San Diego Schools of Medicine and Biological Sciences

9500 Gilman Drive • Leichter Family Foundation Biomedical Research Building, Room 284 • La Jolla, CA 92093

T: 858-822-0491 • jjidoyaga@health.ucsd.edu

signature that distinguishes pDC-derived icDC2s from bona fide cDC2s, suggesting that both populations converge on a shared functional identity—likely driven by overlapping transcriptional regulators. This point is now discussed in the revised text.

- To more directly address the question of pDC-to-icDC2 conversion *in vivo* in human, we focused on datasets of FACS-purified pDCs (**Figure 5**). Specifically, we re-analyzed public dataset of BDCA2⁺ CD123⁺ pDCs isolated from skin blisters (Chen et al., *JEM* 2019). In the revised manuscript, we made several clarifications and added further analyses to validate our conclusions:
 - In the original Chen dataset, two pDC populations were sorted based on BDCA2⁺ CD123⁺ expression, but differed slightly in CD123 protein levels. In our initial manuscript, we referred to one population as "CD123^{low}", which we now recognize may be misleading. We now refer to this population as "CD123^{int}" to reflect moderate CD123 expression and emphasize that these cells were sorted as pDCs using canonical markers, including BDCA2 (Chen et al., *JEM* 2019).
 - As expected based on our original analysis, we now show that the CD123^{int} BDCA2⁺ population aligns closely with cDC2s from our PanDC reference map (**Extended Data Figure 9h**). While this raises the possibility that these cells represent infiltrating cDC2s rather than pDC-derived icDC2s, we note that they were originally sorted as BDCA2⁺ CD123⁺ cells—markers typically used to define pDCs. This suggests that at least some of these cells may arise from pDCs that have downregulated CD123 during inflammation. We now acknowledge this ambiguity in the revised manuscript and address it more directly in subsequent analyses of the CD123^{hi} BDCA2⁺ population (see next point).
 - To further explore the capacity of pDCs to differentiate into icDC2s *in vivo*, we then focused on the CD123^{hi} BDCA2⁺ population, which was FACS-purified as a homogenous population of pDCs by Chen et al. (2019). Surprisingly, even within this population, we observed transcriptional heterogeneity. Upon challenge with house dust mite, cells could be stratified by their pDC signature score into "high," "intermediate," and "low" pDC identity subclusters (**Extended Data Figure 9i and Fig. 5e**). Notably, the "low" pDC signature group, despite being CD123^{hi} BDCA2⁺ by protein expression, exhibited near-complete loss of pDC-defining genes and upregulation of cDC2 genes,

JULIANA IDOYAGA, PHD

CHANCELLOR'S ASSOCIATE PROFESSOR OF PHARMACOLOGY & MOLECULAR BIOLOGY

UC San Diego Schools of Medicine and Biological Sciences

9500 Gilman Drive • Leichtag Family Foundation Biomedical Research Building, Room 284 • La Jolla, CA 92093

T: 858-822-0491 • jjidoyaga@health.ucsd.edu

closely resembling the CD123^{int} cluster. These cells also expressed high levels of cDC activation markers (e.g., HLA-DR, CD86, CD83), further supporting a fate switch (**Figure 5f**). Finally, these cells displayed higher scores for TCF4-repressed genes and SPI1-activated genes, highlighting shared transcriptional network with *in vitro* pDC-derived icDC2s (**Figure 5g**).

While we cannot fully exclude the contribution of *bona fide* cDC2s in the skin blister, our new analyses of CD123^{hi} BDCA2⁺ pDC population support the conclusion that pDC-to-cDC2 conversion can occur in human inflammation. We have revised the text to reflect this interpretation more carefully and transparently.

- In Figure 2l, to prove the stability of the icDC2 population, it would be important to perform this experiment in conditions where sorted icDC2 are re-cultured in the absence of CD40L to ensure these do not revert to prior pDC state. Also, the term stability is counterintuitive considering that the main finding is that pDCs convert into tDC and then into cDC2-like, perhaps saying “differentiation directionality” would make the purpose of the experiment more intuitively understood.

As suggested by the reviewer, we evaluated the potential for itDCs and icDC2s to revert to the pDC state by re-culturing them either in the absence of CD40L or in the presence of IFN-I (see **Figure 2m and 4t**).

- icDC2s remained stable and did not revert to itDCs or pDCs under either condition.
- itDCs also did not revert to pDCs; however, in the presence of IFN-I, a higher proportion of cells remained in the itDC state, suggesting that IFN-I impedes full differentiation of itDCs into icDC2s.

Altogether, these results confirm that pDC-to-cDC2 conversion is a unidirectional and irreversible differentiation process rather than a transient activation state.

Additionally, we appreciate the reviewer’s suggestion to adopt the term “**differentiation directionality**” instead of “**stability**” to more intuitively describe this process. This terminology has been implemented throughout the revised manuscript.

Minor

- In Figure 3, the authors perform various functional experiments clearly demonstrating superior APC functions of icDC2 compared to pDC or tDC and claim that “pDC convert into functional cDCs”. However, the functional capacities of icDC2 are never directly compared to bona fide cDC2. It would therefore be more appropriate to re-phrase the figure title/interpretation of these experiments as “pDC acquire cDC-like functions”.

Thank you for this comment.

As suggested by the reviewer, we have now compared freshly isolated pDCs and cDC2s in their capacity to capture particulate antigen and present it to naïve autologous T cells. Our results show that pDC-derived icDC2s and freshly isolated cDC2s exhibit comparable functional capacities. Specifically, while pDCs are unable to capture particulate antigen, process it, and present it to autologous naïve T cells, both pDC-derived icDC2 and cDC2s efficiently perform these functions (see **Fig. 3h**, **Fig. 3k** and **Extended Data Fig. 6d**). These additional comparisons strengthen our conclusions that pDCs can acquire functional features of cDC2s.

- In Figure 4M, the graph legend “none” is confusing – does this mean the pDC were completely untreated or treated with CD40L but no additional cytokine?

The term “None” refers to cells cultured in medium containing only IL-3, without additional stimuli such as CD40L or cytokines. We now refer to this condition as “medium” and have clarified in the Methods that all culture media contained the survival cytokine IL-3.

POINT-BY-POINT RESPONSE

REVIEWER #1:

In the revised version of their manuscript, the authors have comprehensively and convincingly addressed all my comments and questions. The manuscript has been improved substantially during the revision process, and now provides compelling evidence to support the authors' conclusions.

We thank the reviewer for their words and are pleased that the revised manuscript meets their expectations.

REVIEWER #2:

The authors have done an excellence job of revising this manuscript. The newer work on TNF/IFN cross talk and the improved analysis of changes in pDC during aging have improved the work and I have no further concerns.

Stephen Nutt

We thank Dr. Nutt for his encouraging feedback. We are glad that the additional data and analyses strengthened the manuscript.

REVIEWER #3:

I have carefully reviewed the revised version of the manuscript and the accompanying point-by-point rebuttal. I am pleased to say that the authors have addressed all of my previous comments to my full satisfaction. I want to commend them for the enormous amount of additional work they have undertaken to strengthen the study. The revised manuscript is clear, rigorous, and significantly improved. The final outcome is truly outstanding.

JULIANA IDOYAGA, PHD

CHANCELLOR'S ASSOCIATE PROFESSOR OF PHARMACOLOGY & MOLECULAR BIOLOGY

UC San Diego Schools of Medicine and Biological Sciences

9500 Gilman Drive • Leichtag Family Foundation Biomedical Research Building, Room 284 • La Jolla, CA 92093

T: 858-822-0491 • jidoyaga@health.ucsd.edu

UC San Diego

SCHOOL OF MEDICINE

Department of Pharmacology

UC San Diego

SCHOOL OF BIOLOGICAL SCIENCES

Department of Molecular Biology

Carlos Minutti

We greatly appreciate Dr. Minutti's comments and are thankful for the recognition of the additional efforts made during revision.

JULIANA IDOYAGA, PHD

CHANCELLOR'S ASSOCIATE PROFESSOR OF PHARMACOLOGY & MOLECULAR BIOLOGY

UC San Diego Schools of Medicine and Biological Sciences

9500 Gilman Drive • Leichtag Family Foundation Biomedical Research Building, Room 284 • La Jolla, CA 92093

T: 858-822-0491 • jjidoyaga@health.ucsd.edu